# SPFL: Sequential Updates with Parallel Aggregation for Enhanced Federated Learning Under Category and Domain Shifts

**Haoyuan Liang**[1]*, **Shilei Cao**[1]* , **Guowen Li**[1] , **Zhiyu Ye**[3] , **Haohuan Fu**[2,3] , **Juepeng Zheng**[1,2]†

[1] School of Artificial Intelligence, Sun Yat-sen University, China
[2] National Supercomputing Center in Shenzhen, China
[3] Tsinghua Shenzhen International Graduate School, China
{lianghy68 , caoshlei , ligw8}@mail2.sysu.edu.cn
zhengjp8@mail.sysu.edu.cn, yezy25@mails.tsinghua.edu.cn

## Abstract

Federated Learning (FL) has recently emerged as the primary approach to overcoming data silos, enabling collaborative model training without sharing sensitive or proprietary data. Parallel Federated Learning (PFL) aggregates models trained independently on each client's local data, which could prevent the model from converging to the optimal solution due to limited data exposure. In contrast, Sequential Federated Learning (SFL) allows models to traverse client datasets sequentially, enhancing data utilization. However, SFL effectiveness is limited in real-world Non-IID scenarios characterized by category shift (inconsistent class distributions) and domain shift (distribution discrepancies). These shifts cause two critical issues: *update order sensitivity*, where model performance varies significantly with the sequence of client updates; and *catastrophic forgetting*, where the model forgets previously learned features when trained on new client data. Therefore, based on SFL, we propose a novel updating framework, **SPFL** (**S**equential updates with **P**arallel aggregation **F**ederated **L**earning), that can be integrated into existing PFL methods. It integrates sequential updates with parallel aggregation to enhance data utilization and ease update order sensitivity. Meanwhile, we give the convergence analysis of SPFL under strong convex, general convex, and non-convex conditions, proving that this update scheme is significantly better than PFL and SFL. Additionally, we introduce the GLAM (**G**lobal-**L**ocal **A**lignment **M**odule) to mitigate catastrophic forgetting by aligning the predictions of the local model with those of previous models and the global model during training. Our extensive experiments demonstrate that integrating the SPFL framework into existing PFL methods significantly improves performance under category and domain shifts.

## 1 Introduction

Deep learning models have demonstrated immense potential across various vision tasks, typically depending on large-scale data training [1, 2, 3]. As privacy concerns regarding the centralized collection of extensive data continue to escalate, traditional centralized training methods increasingly fail to address clients' privacy needs. In response, Federated Learning (FL) [4, 5, 6] has emerged as a decentralized solution, designed specifically to prioritize data privacy by allowing models to be trained collaboratively without centralizing sensitive information. Aligning with the foundational

---

*Equal contributions.
†Corresponding author.

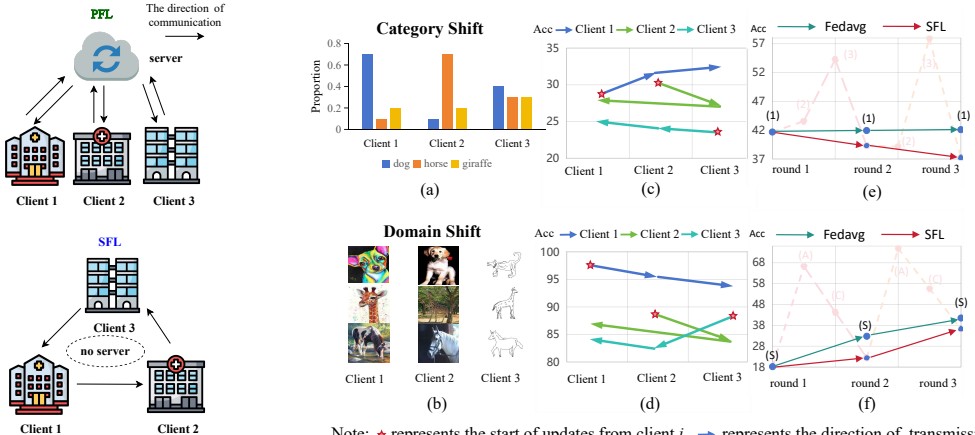

Figure 1: **Top**: Parallel Federated Learning (**PFL**): The server aggregates the local models trained independently on each client. **Bottom**: Sequential Federated Learning (**SFL**): The local model will be trained on the clients sequentially without a centralized sever.

Figure 2: (a) and (b) present cases of category shift and domain shift. (c) and (d) illustrate the performance of three starting points for client training (*i.e.*, *Client 1→Client 2→Client 3*, *Client 2→Client 3→Client 1*, and *Client 3→Client 1→Client 2*) on the three clients' data under category and domain shift, respectively. The solid lines in (e) and (f) compare the performance of *Client 1*'s data between FedAvg, which repeatedly learns solely on *Client 1*'s data, and SFL, which iteratively learns sequentially across different client datasets under category and domain shift, respectively. The dashed lines represent the sequential training process of SFL.

method FedAvg [7], existing FL methods [8, 9] commonly employ Parallel Federated Learning (PFL), which aggregates models trained independently on each client at a central server, as shown in Fig. 1 (top). While PFL effectively enhances the global model's performance through iterative aggregation, it does not fully exploit all available data since each client's model is trained solely on its local data, potentially leading to poor model generalization. This raises the question: **what strategies can be explored to leverage client data better and improve the model generalization?**

One promising strategy is to allow the model to traverse each client's dataset sequentially, as demonstrated in Fig. 1 (bottom), leading to Sequential Federated Learning (SFL) [10, 11, 12]. Although current SFL methods perform effectively in **I**ndependently and **I**dentically **D**istributed (IID) settings across clients to enhance generalization and leverage more data, their performance remains constrained in real-world Non-IID scenarios. The challenges arise because the Non-IID nature across clients mainly manifests in two forms in image tasks: **category shift** and **domain shift**. The category shift refers to inconsistent class distributions across clients. For instance, dog samples constitute 70% of the *Client 1*'s data but only 10% of the *Client 2*'s, as shown in Fig. 2 (a). The domain shift involves distribution discrepancies arising from different domains, such as images sampled from paintings in one client and real photos in another, as depicted in Fig. 2 (b).

To analyze the impact of two types of shifts in SFL, we test SFL on two datasets characterized by the category shift [13, 14] (*i.e.* CIFAR-10) and the domain shift [15, 16] (*i.e.* PACS) following previous work, respectively. The results presented in Fig. 2 reveal two critical issues impacting SFL under Non-IID conditions: (1) **Update Order Sensitivity**: The model becomes highly sensitive to the order in which clients are updated. Different update sequences can lead to inconsistent results in the same client dataset, as illustrated in Fig. 2 (c) and (d). For example, Fig. 2 (c) demonstrates that starting the training process from *Client 2* (*Client 2→Client 3→Client 1*) leads to better performance in the *Client 1* dataset than starting from *Client 3* (*Client 3→Client 1→Client 2*) under category shift. This may be because the latter update order causes the model to become trapped in a saddle point, resulting in consistently poor performance. In addition, we present two examples to theoretically demonstrate the existence of this phenomenon in Appendix Section H. However, solving this by exhaustively exploring all possible update sequences to determine the optimal order is impractical due to the high computational and time costs, especially in real-world applications involving a large

number of clients. (2) **Catastrophic Forgetting**: As the model updates sequentially across clients with different data distributions, it tends to forget previously learned features when trained on new client data, a phenomenon known as catastrophic forgetting in continual learning [17, 15, 18]. This issue is evident in Fig. 2 (e) and (f), where SFL performs worse in *Client 1*'s data after repeated iterations under category or domain shift, compared to SFL, which learns repeatedly on a single client's dataset.

To effectively leverage the available client data and overcome these challenges under category shift and domain shift, we propose a novel FL framework named Sequential updates with Parallel aggregation Federated Learning (SPFL) and a Global-Local Alignment Module (GLAM).

Specifically, SPFL harnesses the strengths of both parallel and sequential learning. The server initially distributes the global model to all clients. Each client trains the model on its local data before passing the updated model to the next client for further training. After completing the sequential update round starting from different clients, the server aggregates these models in parallel to form a new global model for the next iteration. **This hybrid approach maximizes data utilization and mitigates the risk of suboptimal update sequences, improving model robustness within a single computation round.** Moreover, SPFL eliminates the need to share dataset sizes for weighting during aggregation, which increases privacy in applications where dataset sizes may leak sensitive information. Additionally, SPFL can be integrated into existing FL optimization approaches as a new updating strategy to address category shift, such as Scaffold [19] and FedDyn [20]. Furthermore, our proposed GLAM tackles catastrophic forgetting by continuously aligning the predictions of the global model with the local model during training. It also aligns the predictions of the local model with those from the previous client, effectively preserving learned knowledge among clients. The GLAM demonstrates significant effectiveness in improving the performance of SPFL under domain shift. **We also provide a convergence analysis of SPFL under strong convex, general convex, and non-convex conditions, demonstrating that this update scheme outperforms both PFL and SFL in Appendix Section G.** In summary, the contributions of our paper are summarized as follows:

• We identify and analyze the limitations of existing SFL methods in Non-IID scenarios, highlighting the issues of update order sensitivity and catastrophic forgetting under category and domain shifts.

• We propose SPFL, a novel federated learning framework that combines sequential updates with parallel aggregation. SPFL improves data utilization and model robustness by aggregating models with different starting clients to traverse each client's dataset sequentially. We also design GLAM to align predictions of the local model, both with global model and previous local model to mitigate catastrophic forgetting, ensuring that the model retains knowledge across different client updates.

• We provide a convergence analysis under strong convex, general convex, and non-convex conditions. Extensive experiments demonstrate the effectiveness of SPFL under Non-IID conditions (domain shift and category shift), as well as their compatibility with traditional PFL methods.

## 2 Related Work

### 2.1 Parallel Federated Learning

As privacy concerns grow, companies become increasingly hesitant to upload their data to the cloud for centralized model training. To effectively utilize these distributed data, FedAvg [7] pioneer federated learning community that serves as the foundation of Parallel Federated Learning (PFL) [21]. Building upon this work, most federated optimization algorithms have adopted the PFL paradigm, improving FedAvg through various improvements [22, 23, 19, 24, 25]. For example, SCAFFOLD [19] mitigates client drift in federated learning through control variates. FedSAM [25] addresses the decline in global model generalization caused by inconsistent data distributions across clients by employing the Sharpness Aware Minimization optimizer. Although these FL methods significantly enhance the handling of traditional Non-IID issues, they are less effective on complex image tasks involving Non-IID data, such as domain shift [26] and category shift [13]. Therefore, to address domain shift, FedCSA [27] addresses the inconsistency in the global model due to Non-IID data, using a model bias-based client data clustering method. Furthermore, to address the category shift, FedDisco [28] tackles the poor convergence of the global model under the category shift. However, existing PFL methods fall short in fully exploiting the available client data, since each local model is

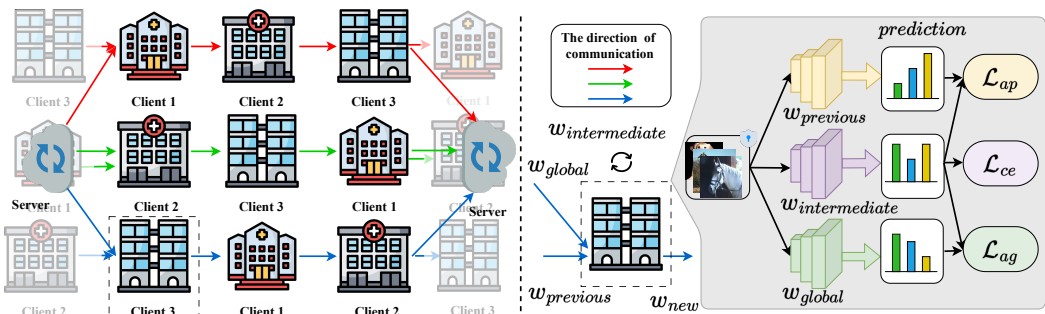

Figure 3: (**Left**) The process of our update method Sequential Updates with Parallel Aggregation (SPFL). The serial numbers indicate the steps to be executed sequentially. (**Right**) The module that solves the forgetting problem when the domain differences of each client are too large, that is, the module that Global-Local Alignment Module (GLAM). Lines of different colors represent the update order of different starts, and the black represents aggregation to the central server.

limited access to its corresponding client data. This limitation may result in suboptimal convergence of model training due to insufficient data exposure, which motivates our proposed SPFL.

## 2.2 Sequential Federated Learning

Nowadays, studies on Sequential Federated Learning (SFL) remain relatively limited. Some research addresses the issue of information leakage in federated learning by performing membership inference attacks in the SFL setting [29]. Although other research uses sequential updates for privacy protection [29], they do not address the issue of poor model performance on Non-IID data. FedGSP [30] seeks to address the decline in model performance caused by Non-IID data in federated learning through a dynamic collaborative training approach that transitions from sequential to grouped parallel updates. However, this approach deviates from the standard SFL setting. Similarly, FedSeq [10] mitigates the issues of slow convergence and performance degradation caused by computational heterogeneity in federated learning by sequentially training on heterogeneous client groups, referred to as super-clients. Although some experimental results demonstrate the advantages of FedSeq on Non-IID data, and some scholars theoretically prove the convergence of SFL under Non-IID conditions [31], these models still perform poorly on complex tasks involving domain shift and category shift.

## 3 Method

In this section, we first introduce the problem formulation. Then, we provide the convergence analysis of SPFL, comparing it with PFL and SFL. Next, to tackle the challenges associated with domain shift for SPFL, we propose a Global-Local Alignment Module (GLAM). The workflow of SPFL is illustrated in Fig. 3. Limited by space, notations are summarized in Appendix D. Meanwhile, the general assumptions for convergence are provided in Appendix E.

### 3.1 Preliminaries

In our paper, we adopt a similar setting to the traditional FL and assume that each client $m$ has its dataset $D_k = \{(x_i^m, y_i^m)\}_{i=1}^{q_m}$ where represents the amount of data from client $k$ and performs model updates in this way where $x \in \mathcal{X}$ is the input and $y \in \mathcal{Y}$ its corresponding label. For each client $k$, we have separate models $f(\cdot; \boldsymbol{w}_k)$. The optimization objective of PFL can be written as (1):

$$\min_w \sum_{m=1}^{M} \sum_{k=1}^{K} \frac{q_m}{\sum_{j'=1}^{M} q_{j'}} \sum_{i=1}^{q_m} \mathcal{L}_m\left(f(x_i^m; \boldsymbol{w}_m), y_i^m\right) \tag{1}$$

Unlike PFL, the server is no longer needed in SFL, and our optimization objective is written as (2):

$$\min_w \sum_{m=1}^{M} \sum_{k=1}^{K} \sum_{i=1}^{q_m} \mathcal{L}_m\left(f(x_i^m; \boldsymbol{w}_m), y_i^m\right) \tag{2}$$

By comparing (1) and (2), we observe that the gradient of $\mathcal{L}_m$ is solely dependent on the dataset of client $m$, whereas the gradient of $\mathcal{L}_m$ in (2) is influenced by all the datasets. Meanwhile, we can see that SFL does not need to introduce the dataset size in the optimization. However, as we analyzed in Fig. 2, inconsistent starting points have a great impact on model performance. To solve this problem, we proposed Sequential updates with Parallel aggregation (SPFL), which can avoid the worst solution of the model, but of course, this does not represent the best model performance. To distinguish the different models at each starting point, we add conditional probability to the optimization target to fully consider this issue. Meanwhile, because SPFL goes through each client, the dataset that undergoes gradient descent is the same size, and we no longer need to consider the impact of the dataset size on the model, which could be proved in Fig. 4c.

## 3.2 Sequential updates with Parallel aggregation

Our paper proposes a new framework of updating Sequential updates with Parallel aggregation (SPFL), similar to the client setting in PFL. Unlike PFL, our updating framework can continuously learn from *different client datasets*. From the perspective of global optimization, we are different from PFL because the aggregation process is different from the SFL aggregation process, as shown:

$$\min_{w} \frac{1}{M} \sum_{n=1}^{M} \sum_{m=1}^{M} \sum_{k=1}^{K} \sum_{i=1}^{q_{\pi_m^n,k}} \nabla \mathcal{L}_{\pi_m^n,k} \left( f(x_i^{\pi_m^n,k}; \boldsymbol{w}_{\pi_m^n,k}), y_i^{\pi_m^n,k} | \sum_{j=1}^{m-1} \nabla \mathcal{L}_{\pi_j^n,k} \right) \tag{3}$$

However, to compare the subsequent convergence analysis, according to Assumption 1, this deviation can be incorporated into the $\sigma$ in Assumption 3. For client $m$, the update method is simplified as (4). See (45) for more details.

$$\boldsymbol{w}^{(r+1)} = \boldsymbol{w}^{(r)} - \eta \frac{1}{M} \sum_{n=1}^{M} \sum_{m=1}^{M} \sum_{k=0}^{K-1} \mathbf{g}_{\pi_m^n,k}^{(r)} \tag{4}$$

To help readers better understand SPFL, we provide the pseudo-code in Appendix Algorithm 1.

## 3.3 Convergence analysis of SPFL

**Theorem 1.** *For SPFL, there exists a constant effective learning rate $\tilde{\eta} := MK\eta$ and weights $\theta_r$, such that the weighted average of the global parameters $\bar{\boldsymbol{w}}^{(R)} = \frac{1}{W_R} \sum_{r=0}^{R} \theta_r \boldsymbol{w}^{(r)}$ (where $W_R = \sum_{r=0}^{R} \theta_r$) satisfies the following upper bounds:*

***Strongly convex:*** *Under Assumptions 2, 3, and 5, there exists a constant effective learning rate $\frac{1}{\mu R} \leq \tilde{\eta} \leq \frac{1}{6L}$ and weights $\theta_r = \left(1 - \frac{\mu \tilde{\eta}}{2}\right)^{-(r+1)}$, such that the following holds:*

$$\mathbb{E}\left[\mathcal{L}(\bar{\boldsymbol{w}}^{(R)}) - \mathcal{L}(\boldsymbol{w}^*)\right] \leq \frac{9}{2} \mu \mathcal{A}^2 \exp\left(-\frac{1}{2}\mu \tilde{\eta} R\right) + \frac{12\tilde{\eta}\sigma^2}{M^2 K} + \frac{18L\tilde{\eta}^2\sigma^2}{MK} + \frac{18L\tilde{\eta}^2\zeta_*^2}{MK} \tag{5}$$

***General convex:*** *Under Assumptions 2, 3, and 5, there exists a constant effective learning rate $\tilde{\eta} \leq \frac{1}{6L}$ and weights $\theta_r = 1$, such that the following holds:*

$$\mathbb{E}\left[\mathcal{L}(\bar{\boldsymbol{w}}^{(R)}) - \mathcal{L}(\boldsymbol{w}^*)\right] \leq \frac{3\mathcal{A}^2}{\tilde{\eta}R} + \frac{12\tilde{\eta}\sigma^2}{M^2 K} + \frac{18L\tilde{\eta}^2\sigma^2}{MK} + \frac{18L\tilde{\eta}^2\zeta_*^2}{MK} \tag{6}$$

***Non-convex:*** *Under Assumptions 2, 3, and 4, there exists a constant effective learning rate $\tilde{\eta} \leq \frac{1}{6L(\beta+1)}$ and weights $\theta_r = 1$, such that the following holds:*

$$\min_{0 \leq r \leq R} \mathbb{E}\left[\left\|\nabla \mathcal{L}(\boldsymbol{w}^{(r)})\right\|^2\right] \leq \frac{3\mathcal{B}}{\tilde{\eta}R} + \frac{3L\tilde{\eta}\sigma^2}{M^2 K} + \frac{27L^2\tilde{\eta}^2\sigma^2}{8MK} + \frac{27L^2\tilde{\eta}^2\zeta^2}{8M} \tag{7}$$

$\mathcal{A} := \|\boldsymbol{w}^{(0)} - \boldsymbol{w}^*\|$ *for the convex cases and* $\mathcal{B} := \mathcal{L}(\boldsymbol{w}^{(0)}) - \mathcal{L}^*$ *for the non-convex case.*

We include Corollary 1 in the appendix. Additionally, we compare SPFL, SFL, and PFL. As shown in Tab. 1, SPFL achieves a smaller convergence upper bound than both SFL and PFL, indicating a faster convergence rate. Compared to PFL and SFL, our proposed SPFL achieves a tighter upper

Table 1: The upper bounds in the strongly convex case, with absolute constants and polylogarithmic factors omitted, are presented. The general convex and non-convex cases are included in Corollary 1.

| Method | Bound ($\mathcal{A} = \|\boldsymbol{w}^{(0)} - \boldsymbol{w}^*\|$) |
|---|---|
| PFL (FedAvg) [31] | $\frac{\sigma^2}{\mu MKR} + \frac{L\sigma^2}{\mu^2 KR^2} + \frac{L\zeta_*^2}{\mu^2 R^2} + \mu\mathcal{A}^2 \exp\left(-\frac{\mu R}{L}\right)$ |
| PFL (Scaffold) [19] | $\frac{\sigma^2}{\mu MKR} + \frac{L\sigma^2}{\mu^2 KR^2} + \frac{L\zeta_*^2}{\mu^2 R^2} + \mu\mathcal{A}^2 \exp\left(-\frac{\mu R}{L}\right)$ |
| SFL [31] | $\frac{\sigma^2}{\mu MKR} + \frac{L\sigma^2}{\mu^2 MKR^2} + \frac{L\zeta_*^2}{\mu^2 MR^2} + \mu\mathcal{A}^2 \exp\left(-\frac{\mu R}{L}\right)$ |
| SPFL (ours) | $\frac{\sigma^2}{\mu M^2 KR} + \frac{L\sigma^2}{\mu^2 MKR^2} + \frac{L\zeta_*^2}{\mu^2 MR^2} + \mu\mathcal{A}^2 \exp\left(-\frac{\mu R}{L}\right)$ |

bound in the strongly convex case, notably improving the variance term from $\tilde{\mathcal{O}}(1/(MKR))$ to $\tilde{\mathcal{O}}(1/(M^2 KR))$. Therefore, increasing the number of *Clients* can enhance the convergence speed of the model compared to other methods.

As shown in Fig. 3, we set an update order for the startup of each client and perform aggregation after each step, enabling more effective integration with traditional personalized federated learning (PFL). This sequential federated learning (SFL) strategy allows the model to traverse a larger volume of data, leading to a 3% improvement in handling category shift. However, SFL alone remains insufficient for addressing domain shift due to the inherently uneven data distributions across clients. While SPFL inherits the PFL-style weighted aggregation, which helps maintain competitive performance under domain shift, it still faces limitations. To mitigate these issues, we introduce a global-local alignment module (GLAM) specifically designed to enhance robustness against domain discrepancies.

### 3.4 Global-Local Alignment Module (GLAM)

In SFL with client data, if the model does not learn enough in the current domain and quickly transitions to the next domain, it may cause the model to forget the features learned in the previous round when moving to the next. For example, in the PACS task, when the model transitions from learning comics to the photo domain, we aim to offset the style features during the learning process. However, it seems that some content information is also lost, making the SFL method significantly inferior to the PFL method in domain transfer. Parameter are summarized in Appendix D.

In real-world federated learning, the data distribution of each client is unknown, so the model needs to be robust enough to perform well across a variety of potential scenarios. Traditional solutions to the forgetting problem typically store part of the source domain data, which is not permissible in federated learning. **In SPFL, each client maintains not only its own trained model but also a global model and a model passed from the previous client—an advantage that neither SFL nor PFL possesses.** To save resources as much as possible and fully leverage the advantages of SPFL, $f(\boldsymbol{w}_g; \cdot)$ and $f(\boldsymbol{w}_p; \cdot)$ are used solely for inference prediction, while they assist model $(\boldsymbol{w}_i; \cdot)$ in performing gradient descent. We fully leverage this advantage and develop a global-local alignment module (GLAM) based on these three models. To prevent the local model from excessively forgetting previous information, we designed two additional *loss* terms.

To ensure the convergence of the training process, we first use the cross-entropy loss function as the basis for gradient descent. The specific loss function used during the training is as follows:

$$\mathcal{L}_{ce} = -\frac{1}{B} \sum_{i=1}^{B} \boldsymbol{y}_i^T \log(f(\boldsymbol{w}_i; \boldsymbol{x}_i^{\pi_m^n, k})) \tag{8}$$

To mitigate catastrophic forgetting during training, we incorporate the predictions of the global model with those of the model under training by using a mutual cross-entropy loss, as shown in (9). This loss $\mathcal{L}_{ag}$ is named Approach to the global model, ensures consistency at the output level, and facilitates alignment between local and global models.

$$\mathcal{L}_{ag} = -\frac{1}{B} \sum_{i=1}^{B} f(\boldsymbol{w}_g; x_i^{\pi_m^n, k})^T \log(f(\boldsymbol{w}_i; x_i^{\pi_m^n, k})) \tag{9}$$

For a client, we can also use the model passed from the previous client to align, thereby reducing forgetting of the previous client. Unlike before, we use KL divergence to make the distributions of

the two outputs consistent. We refer to this loss as the Local Alignment with the Previous Output $\mathcal{L}_{ap}$, which is defined as follows:

$$\mathcal{L}_{ap} = -\frac{1}{2B}\sum_{i=1}^{B} KL(P||Q) + KL(Q||P), s.t. \ P = f(\boldsymbol{w}_i; x_i^{\pi_m^n, k}), Q = f(\boldsymbol{w}_p; x_i^{\pi_m^n, k}) \quad (10)$$

Table 2: Comparison of Classic and State-of-the-Art Algorithms in Federated Learning with and without SPFL on Cifar-10, Cifar-100, CINIC-10 and Fmnist in Category Shift

| Method | Cifar-10 resnet18 (num=10) | Cifar-100 resnet18 (num=10) | CINIC-10 simple-cnn (num=10) | Fmnist resnet18 (num=10) | Avg |
|---|---|---|---|---|---|
| FedAvg [7] | 75.00 | 57.55 | 39.74 | 81.13 | 63.35 |
| + SPFL | 78.88 (**+3.88**) | 64.33 (**+6.78**) | 41.01 (**+1.27**) | 84.75(**+3.62**) | 67.24 (**+3.89**) |
| FedDc [32] | 80.52 | 64.44 | 40.92 | 83.44 | 67.33 |
| + SPFL | 85.90(**+5.38**) | 69.88 (**+5.44**) | 44.14 (**+3.23**) | 87.67(**+4.23**) | 71.90 (**+4.57**) |
| FedDyn [20] | 77.91 | 64.11 | 40.91 | 81.35 | 66.03 |
| + SPFL | 80.90(**+2.99**) | 64.24 (**+0.13**) | 43.06 (**+2.15**) | 83.23(**+1.88**) | 67.86 (**+1.83**) |
| FedNova [33] | 76.02 | 57.64 | 39.82 | 81.38 | 63.46 |
| + SPFL | 79.31(**+3.29**) | 64.47 (**+6.83**) | 41.32 (**+1.50**) | 84.85(**+3.47**) | 67.49(**+4.03**) |
| FedProx [34] | 76.04 | 57.56 | 39.65 | 81.23 | 63.62 |
| + SPFL | 77.25(**+1.21**) | 60.46 (**+2.90**) | 40.27 (**+0.62**) | 83.89(**+2.66**) | 65.47 (**+1.85**) |
| MOON [35] | 78.70 | 58.44 | 40.11 | 81.29 | 63.68 |
| + SPFL | 79.98 (**+1.28**) | 63.54 (**+5.20**) | 41.11 (**+1.00**) | 84.83(**+3.54**) | 67.37(**+3.69**) |
| SCAFFOLD [19] | 76.38 | 56.15 | 36.00 | 80.71 | 62.31 |
| + SPFL | 78.73 (**+2.35**) | 65.05 (**+8.90**) | 39.52 (**+3.52**) | 85.80(**+5.09**) | 67.26 (**+4.95**) |
| FedDisco [28] | 76.32 | 57.50 | 39.67 | 81.24 | 63.68 |
| + SPFL | 78.58(**+2.26**) | 64.44 (**+6.94**) | 40.79 (**+1.12**) | 84.27(**+3.02**) | 67.13(**+3.45**) |

Table 3: Comparison of Classic Algorithms in FL with and without SPFL on Cifar-10 in IID

| Method | Fedavg [7] | Feddc [32] | FedDyn [20] | FedNova [33] | FedProx [34] | MOON [35] | SCAFFOLD [19] | FedDisco [28] |
|---|---|---|---|---|---|---|---|---|
| PFL | 83.81 | 81.75 | 80.66 | 83.39 | 83.94 | 84.03 | 82.62 | 84.03 |
| SPFL | 89.85 (**+6.54**) | 93.08 (**+11.33**) | 81.77 (**+1.11**) | 90.00 (**+6.61**) | 86.12(**+2.18**) | 90.16 (**+6.13**) | 91.97 (**+9.53**) | 89.92 (**+5.89**) |

This approach helps to maintain a consistent output distribution across clients, thus facilitating a smoother transition and reducing catastrophic forgetting. Combining (9) , (10) , and (8), we can derive the final loss function $\mathcal{L}$ for our model, as shown below:

$$\mathcal{L} = \tau\mathcal{L}_{ap} + \rho\mathcal{L}_{ag} + \mathcal{L}_{ce} \quad (11)$$

where $\tau$ and $\rho$ are hyperparameters used to control the influence of the previous client's model and the global model on the current gradient descent, which affects the convergence of the model.

In this paper, we enhance SFL to develop SPFL and introduce the GLAM module, making SPFL a more effective update method than PFL. SPFL achieves significant improvements under category shift and delivers domain shift performance comparable to that of PFL. These insights are empirically validated through extensive experiments.

## 4 Experiments

### 4.1 Set up

In this section, we introduce the setup. Unless otherwise specified, all experiments will be conducted under the following conditions. We include the dataset descriptions and important experimental details in Appendix Sections I.1 and I.2. Here, we present only the comparison methods:

**Comparing Methods.** We compared our proposed method, SPFL, with the most classic algorithms in federated learning, both with and without integration. This comparison aimed to evaluate the performance enhancement that the integration of SPFL brings to the federated learning process. Including: (1) Fedavg [7] is the first algorithm for FL and PFL; (2) FedProx [34], SCAFFOLD [19], FedDyn [20], MOON [35], and FedDC[32] focus on dynamically adjusting the client models; (3) FedNova [33] adjusts the iteration counts from a global perspective to optimize the federated learning process; (4) FedDisco [28] is the first paper that focuses on addressing category shift in FL.

To comprehensively evaluate the impact of the SPFL update framework under domain shift, we compare not only with traditional PFL methods but also with domain generalization approaches (e.g.,

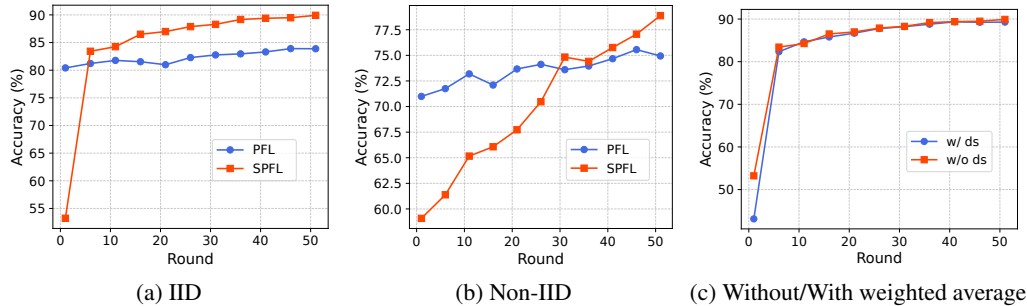

| (a) IID | (b) Non-IID | (c) Without/With weighted average |

Figure 4: (a) Comparison of the performance of SPFL and PFL on IID data in the cifar 10; (b) Comparison of the performance of SPFL and PFL on Non-IID in the cifar 10; (c) indicates that the Weighted Average shows no significant difference from the Arithmetic Average in SPFL.

Table 4: Comparison between PFL method and *GLAM* under the updated framework of SPFL

| Method | is PFL? | with SPFL | *Photo* | *Art* | *Cartoon* | *Sketch* | Avg |
|---|---|---|---|---|---|---|---|
| Ditto [40] | ✓ | ✓ | 92.16 | 74.22 | 69.03 | 63.09 | 74.63 |
| Fedavg [7] | ✓ | ✓ | 90.96 | 76.17 | 72.31 | 65.84 | 76.32 |
| Fedprox [34] | ✓ | ✓ | 91.62 | 76.07 | 71.20 | 67.68 | 76.64 |
| Scaffold [19] | ✓ | ✓ | 91.32 | 76.61 | 67.45 | 67.12 | 75.62 |
| AM [36] | ✓ | ✓ | 92.16 | 81.01 | 67.15 | 68.01 | 77.08 |
| RSC [37] | ✓ | ✓ | 89.82 | 75.05 | 72.40 | 66.94 | 76.05 |
| CCNet [39] | ✓ | ✓ | 91.92 | 75.10 | 65.10 | **70.78** | 75.73 |
| Fedseq [41] | ✗ | ✗ | 70.54 | 59.01 | 58.53 | 52.28 | 60.04 |
| FedDG-GA [38] | ✓ | ✓ | 90.78 | 69.63 | 69.58 | 63.76 | 73.44 |
| SPFL-GLAM (ours) | ✗ | ✓ | **92.28** | **77.44** | **74.42** | 67.32 | **77.87** |

AM [36], RSC [37]) and federated domain generalization methods (e.g., FedDG-GA [38], CCNet [39]). Ditto [40] is a personalized FL method that incorporates model distillation. Since our approach also involves distillation components, we include it in our comparisons as well. As shown in Tab. 1, our proposed GLAM effectively mitigates the impact of domain shift within the SPFL framework. Currently, the performance of most federated sequential learning methods heavily depends on data distribution and hierarchical aggregation strategies. For comparison, we include FedSeq [41], which demonstrates limited effectiveness under domain shift scenarios.

## 4.2 Results

**The performance on IID.** From Tab. 3, it can be observed that for client data under IID, SPFL can fully leverage its capability to encounter more data. SPFL can perfectly adapt to each method, achieving an average improvement of over 6%. Additionally, SPFL demonstrates a fast convergence rate; the model can reach from 40% accuracy to 70% in just one round, as shown in Fig. 4a. However, most real-world data is Non-IID. So we focus more on the performance of SPFL in the domain and category shift. In Fig. 4b, SPFL also shows strong performance in the category shift.

**Performance under Category Shift.** We evaluate SPFL on four datasets (CIFAR-10 [42], CIFAR-100 [42], CINIC-10 [43], and FMNIST [44]), as shown in Tab. 2. SPFL can be effectively integrated with existing federated learning methods, yielding an average performance gain of approximately 3%. This integration highlights the compatibility and potential of SPFL in enhancing federated learning algorithms across diverse data distributions under category shift.

**Performance under Domain Shift.** In Tab. 5, the proposed SPFL framework alone does not perform well under domain shift. To address this limitation, we introduce the Global-Local Alignment Module (GLAM), which significantly narrows the performance gap. With the integration of GLAM, SPFL's performance improved from 2.05% below the expected baseline to just 0.5% below, demonstrating the module's effectiveness in enhancing SPFL's robustness to domain shift. Additionally, most PFL methods integrated into the SPFL framework show suboptimal performance, as illustrated in Tab. 4.

## 4.3 Ablation Studies

**Impact of Hyperparameters $\tau$ and $\rho$ in the model.** As shown in Fig. 5a and 5b, we observe that setting $\tau$ and $\rho$ too high leads the model to overly align with the global models and previous

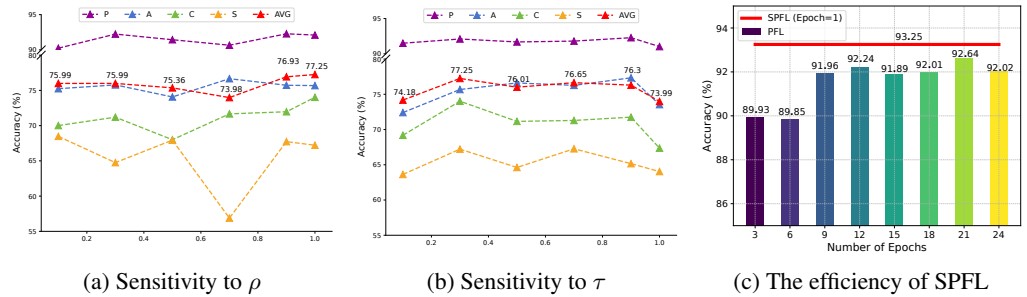

|       (a) Sensitivity to $\rho$       |       (b) Sensitivity to $\tau$       |       (c) The efficiency of SPFL       |

Figure 5: (a) and (b) illustrate the sensitivity to GLAM hyperparameters; (c) demonstrates increasing the number of PFL's local Epoch under category shift does not lead to significant improvement.

Table 5: The combination experiments of the two different loss functions $\mathcal{L}_{ap}$ and $\mathcal{L}_{ag}$ were conducted on the PACS dataset using a pre-trained ResNet-18 (Best in **bold**)

| Method | $\mathcal{L}_{ag}$ | $\mathcal{L}_{ap}$ | *Photo* | *Art* | *Cartoon* | *Sketch* | Avg |
|---|---|---|---|---|---|---|---|
| PFL (Fedavg) | × | × | 90.48 | 76.95 | 74.87 | **71.21** | **78.37** |
| SFL (Fedseq) | × | × | 70.54 | 59.01 | 58.53 | 52.28 | 60.04 |
| SPFL | × | × | 90.96 | 76.17 | 72.31 | 65.84 | 76.32 |
| SPFL | ✓ | × | 91.50 | 73.73 | 72.40 | 66.58 | 76.05 |
| SPFL | × | ✓ | 92.22 | 76.12 | 71.46 | 64.34 | 76.04 |
| SPFL | ✓ | ✓ | 92.28 | **77.44** | **74.42** | 67.32 | 77.87 |

rounds' models, which in turn degrades local training performance. Through our sensitivity analysis, we observe that the combination $\tau = 0.3$ and $\rho = 1$ yields the best model performance.

**Impact of the GLAM Module.** As shown in Tab. 5, SPFL alone is insufficient to effectively address domain shift. However, with the introduction of the proposed GLAM module, the SPFL update method becomes competitive under domain shift scenarios. Each component of GLAM ($\mathcal{L}_{ap}$ and $\mathcal{L}_{ag}$) contributes to the overall performance improvement. In addition, we added in the appendix Tab. 10 that if PFL is used first to improve the overall ability of the model, and then SPFL is used, the generalization of the model can be further improved.

**The Impact of Uploaded Dataset Size on the Model.** In PFL, the contribution of each client is weighted based on the size of its dataset. In contrast, SPFL iteratively updates the model over all client datasets, effectively treating them as having equal size. **As a result, SPFL does not require uploading client data, offering stronger privacy protection compared to PFL.** As shown in Fig. 4c, our experimental results further confirm that the presence or absence of client data upload has no significant impact on model performance in SPFL.

**The impact of Epoch on the SPFL framework.** In our setting, we trained for 1 epoch with 10 clients, resulting in a computational cost that is 10 times higher than PFL. However, as shown in Fig. 5c, when the number of epochs is set to 10, the resource consumption of PFL and SPFL becomes comparable. In Fig. 5c, we conduct 200 rounds of training to approximate the theoretical upper bound. It can be observed that SPFL improves the upper limit of model convergence when addressing category shift, which is consistent with the results reported in Tab. 1. **Notably, even when the number of PFL epochs is increased to 21—doubling the computational cost compared to SPFL—the model performance of PFL remains 0.6% lower than that of SPFL!**

## 5    Communication cost analysis

When directly applying SPFL to approximate the performance of PFL, it often incurs significant communication and computational overhead. To demonstrate the advantage of SPFL in achieving a lower convergence bound, the pre-trained model reported in Tab. 6 and Tab. 7 is trained on CIFAR-10 with FedAvg for 200 rounds across 10 clients. Furthermore, to show that SPFL can attain a superior convergence upper bound, we use SPFL to match the effect of PFL within only 10 rounds, surpassing the performance of PFL trained for thousands of rounds, while simultaneously

Table 6: Efficiency Comparison on CIFAR-10 (Dirichlet $\alpha$=0.1, Number of clients=10)

| Method | Accuracy(%) | Rounds | Comm. Cost (GB) | Total Time(s) | Compute Cost(GB) |
|--------|-------------|--------|-----------------|---------------|------------------|
| PFL | 79.15 | 500 | 116.90 | 3419.91 | 175.35 |
| SPFL(Ours) | 84.17 | 15 | 21.042 | 678.6 | 87.675 |

Table 7: Efficiency Comparison on CIFAR-10 (Dirichlet $\alpha$=0.1, Number of clients=100)

| Method | Accuracy(%) | Rounds | Comm. Cost (TB) | Total Time(s) | Compute Cost(TB) |
|--------|-------------|--------|-----------------|---------------|------------------|
| PFL | 81.38 | 2000 | 18.30 | 30259.82 | 27.46 |
| SPFL(Ours) | 86.68 | 10 | 4.68 | 14713.13 | 22.85 |

reducing both communication and computational costs. This provides a new direction for the practical application of SPFL.

## 6 Conclusion

In this paper, we first identify and analyze the limitations of existing Sequential Federated Learning (SFL) methods under non-independent and identically distributed (Non-IID) scenarios, with a focus on addressing category and domain shifts caused by update order sensitivity and catastrophic forgetting. To tackle these challenges, we propose SPFL, a novel framework that integrates seamlessly with existing federated learning (FL) methods. SPFL combines sequential updates with parallel aggregation to enhance data utilization and reduce sensitivity to update order. In this framework, the model is distributed to all clients, where each client performs local training and sequentially passes the model. The server then aggregates the updated models from each starting client to generate a new global model. We provide a convergence analysis of the SPFL update scheme, showing that it achieves faster convergence than both PFL and SFL across strongly convex, general convex, and non-convex settings. Additionally, we introduce the Global-Local Alignment Module (GLAM) to address catastrophic forgetting by aligning the predictions of the global model with those of the local and previous models during training. Extensive experiments demonstrate the effectiveness of SPFL and GLAM under Non-IID conditions—specifically domain and category shifts—and confirm their compatibility with traditional Parallel Federated Learning (PFL) methods.

## 7 Acknowledgement

This work was supported in part by the National Natural Science Foundation of China under Grant T2125006 and Grant 42401415; in part by Shenzhen Science and Technology Program under Grant KCXFZ20240903093759004 and Grant KJZD20230923115106012; in part by the Fundamental Research Funds for the Central Universities, Sun Yat-sen University, under Project 24xkjc002; and in part by Jiangsu Innovation Capacity Building Program under Project BM2022028.

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

# SPFL: Sequential Updates with Parallel Aggregation for Enhanced Federated Learning Under Category and Domain Shifts (Supplementary material)

## Table of Contents in Appendix

# A Pseudocode

Here we show our algorithm process.

---

**Algorithm 1** Sequential updates With Parallel aggregation Federated learning (SPFL)

---

**Require:** Initial global model $w = w_0$, the number of clients is $k$ , the client dataset $D = \{D_1, D_2, ..., D_k\}$, (Hyperparameters:local epoch $T$,total aggregation round $R$)
**Ensure:** Final global model $w_R$

1: **Server:**
2:     Deploy the global model $w_0$ to each client to obtain each client model:$w_0^k = w_0$
3:     **for** each round $i = 1, 2, ..., R$ **do**
4:         **for** each client $e = 1, 2, ..., k$ **in parallel do**
5:           **for** each client $a = e, ..., k \& 1, ..., e - 1$ **do**
6:             **if** $e + 1 \leq k$ :
7:               $w_i^{e+1} \leftarrow$ **Client** $(e, w_i)$
8:             **else**:
9:               $w_{i+1}^e \leftarrow$ **Client** $(e, w_i)$
10:         $w_{i+1} \leftarrow \frac{1}{k} \sum_{e=1}^k w_{i+1}^e$
11:         Deploy $w_{i+1}$ to all clients.
12: **Client:**
13:     **for** each local epoch $t = 1, 2, ..., T$ **do**
14:         **for** Batch $b = 1, 2, ..., B$ **do**
15:           $\nabla \mathcal{L}(w_{i+(t)}; x_{B_q}^{D_k}) = \nabla \mathcal{L}_{ce} + \tau \nabla \mathcal{L}_{ag} + \rho \nabla \mathcal{L}_{ap}$
16:           $w_{i+(t+1)} \leftarrow w_{i+(t)} - \eta \nabla \mathcal{L}(w_{i+(t)}; x_{B_q}^{D_k})$
17:     **if** $k \% R$ :
18:         return $w_{i+1} = w_{i+(T)}$ to **Next Client**
19:     **else** :
20:         return $w_{i+1} = w_{i+(T)}$ to **Server**

---

# B More Related Work

Domain generalization is also an important criterion for evaluating the model. Examining the generalization ability of the global model is crucial, as it better reflects the overall performance of the model rather than its tendency to overfit. Therefore, we introduce related work on Federated Domain Generalization in this section.

## B.1 Federated Domain Generalization

In recent years, Domain Generalization (DG) has developed rapidly, aiming to learn models that generalize well to unseen domains. However, in practice, multi-source data often exhibit significant domain shifts—for example, between general-purpose datasets and meteorological datasets [45]. Federated Domain Generalization (FedDG) [46, 47] is an emerging field where each domain trains a local model, which is then aggregated, considering the generalization ability of the global model on target clients (domains). Domain shift is one of the most important issues in FedDG, which is consistent with this paper. Despite recent progress, research in this area is still limited. For example, ELCFS [48] generalizes the model by sharing spectra across domains, yet this approach risks privacy leaks. [49] further proposes a flatness-aware optimization method for better generalization on local updates. CCST [50] could lead to more uniform distributions of source clients and make each local model learn to fit the image styles of all the clients to avoid the different model biases. An adaptive weighting method, named GA [38], is proposed to achieve a tighter generalization bound through explicit re-weighted aggregation. However, the abovementioned studies only focus on the Federated Aggregation. To this end, AutoFedGP [51] tackles the domain adaptation (DA) problem by projecting local gradients and assigning adaptive weights to align source and target domains. For domain generalization (DG), FedOMG [52] enhances robustness to unseen domains by aligning gradient directions across clients without access to target data. However, both approaches focus on model-level alignment, which may be insufficient when domain shifts induce severe forgetting during sequential

client updates. In contrast, our proposed SPFL explicitly targets this issue by introducing GLAM. This global-local alignment mechanism concurrently aligns each client with both its predecessor and the global model, effectively mitigating the forgetting problem caused by domain shift.

## C   Corollary 1

Due to space limitations, we introduce Corollary 1 here.

**Corollary 1.** *Applying the results of Theorem 1, and by selecting an appropriate learning rate (see the proof of Theorem 1 in the Appendix Section G), we derive the following convergence bounds for SPFL:*

***Strongly convex:*** *Under Assumptions 2, 3, and 5, there exists a constant effective learning rate* $\frac{1}{\mu R} \leq \tilde{\eta} \leq \frac{1}{6L}$ *and weights* $\theta_r = \left(1 - \frac{\mu\tilde{\eta}}{2}\right)^{-(r+1)}$, *such that the following holds:*

$$\mathbb{E}\left[\mathcal{L}(\bar{\boldsymbol{w}}^{(R)}) - \mathcal{L}(\boldsymbol{w}^*)\right] = \tilde{\mathcal{O}}\left(\frac{\sigma^2}{\mu M^2 K R} + \frac{L\sigma^2}{\mu^2 M K R^2} + \frac{L\zeta_*^2}{\mu^2 M R^2} + \mu \mathcal{A}^2 \exp\left(-\frac{\mu R}{12L}\right)\right) \quad (12)$$

***General convex:*** *Under Assumptions 2, 3, and 5, there exists a constant effective learning rate* $\tilde{\eta} \leq \frac{1}{6L}$ *and weights* $\theta_r = 1$, *such that the following holds:*

$$\mathbb{E}\left[\mathcal{L}(\bar{\boldsymbol{w}}^{(R)}) - \mathcal{L}(\boldsymbol{w}^*)\right] = \mathcal{O}\left(\frac{\sigma\mathcal{A}}{\sqrt{M^2 K R}} + \frac{\left(L\sigma^2\mathcal{A}^4\right)^{1/3}}{(MK)^{1/3}R^{2/3}} + \frac{\left(L\zeta_*^2\mathcal{A}^4\right)^{1/3}}{M^{1/3}R^{2/3}} + \frac{L\mathcal{A}^2}{R}\right) \quad (13)$$

***Non-convex:*** *Under Assumptions 2, 3, and 4, there exists a constant effective learning rate* $\tilde{\eta} \leq \frac{1}{6L(\beta+1)}$ *and weights* $\theta_r = 1$, *such that the following holds:*

$$\min_{0 \leq r \leq R} \mathbb{E}\left[\|\nabla\mathcal{L}(\boldsymbol{w}^{(r)})\|^2\right] = \mathcal{O}\left(\frac{\left(L\sigma^2\mathcal{B}\right)^{1/2}}{\sqrt{M^2 K R}} + \frac{\left(L^2\sigma^2\mathcal{B}^2\right)^{1/3}}{(MK)^{1/3}R^{2/3}} + \frac{\left(L^2\zeta_*^2\mathcal{B}^2\right)^{1/3}}{M^{1/3}R^{2/3}} + \frac{L\beta\mathcal{B}}{R}\right) \quad (14)$$

where $\mathcal{O}$ omits absolute constants, $\tilde{\mathcal{O}}$ omits absolute constants and polylogarithmic factors, $\mathcal{A} := \|\boldsymbol{w}^{(0)} - \boldsymbol{w}^*\|$ for the convex cases, and $\mathcal{B} := \mathcal{L}(\boldsymbol{w}^{(0)}) - \mathcal{L}^*$ for the non-convex case.

## D   Notations

Tab.8 summarizes the notations appearing in this paper. We follow the same settings as in [31] to compare the convergence bounds better.

## E   Assumption

We assume that:

- $\mathcal{L}$ is lower bounded by $\mathcal{L}^*$ for all cases and there exists a minimizer $\boldsymbol{w}^*$ such that $\mathcal{L}(\boldsymbol{w}^*) = \mathcal{L}^*$ for strongly and generally convex cases;
- each local objective function is *L-smooth*(Assumption.2);
- Furthermore, we need to make assumptions about the diversities: the assumptions on the stochasticity bounding the diversity of $\{\mathcal{L}_m(\cdot; \xi_m^i) : i \in [|\mathcal{D}_m|]\}$ with respect to $i$ inside each client (Assumption 3);
- the assumptions on the heterogeneity bounding the diversity of local objectives $\{\mathcal{L}_m : m \in [M]\}$ with respect to $m$ across clients (Assumptions 4, 5).

**Assumption 1** (Gradient Boundedness)**.** *There exists a constant $G > 0$ such that for all clients $m$, any parameter $x \in \mathbb{R}^d$, and any sample $\xi$, the following holds:*

$$\|\nabla\mathcal{L}_m(\boldsymbol{w}; \xi)\| \leq G. \quad (15)$$

Table 8: Summary of key notations.

| Symbol | Description |
|---|---|
| $R, r$ | number, index of training rounds |
| $M, m$ | number, index of clients |
| $K, k$ | number, index of local update steps |
| $q^m$ | Number of datasets for client $m$ |
| $B$ | Number of batch size |
| $N$ | number of participating clients |
| $n$ | Client $n$ is the starting $Client$ |
| $S$ | S represents S clients selected from M clients. When all clients participate,$M = S$ |
| $\pi$ | $\{\pi_1, \pi_2, \ldots, \pi_M\}$ is a permutation of $\{1, 2, \ldots, M\}$ |
| $\pi^{(r+1)}$ | $\pi^{(r+1)} = \mathcal{R}\left(\pi^{(r)}\right) := \left[\pi_2^{(r)}, \pi_3^{(r)}, \ldots, \pi_M^{(r)}, \pi_1^{(r)}\right]$ |
| $\eta$ | learning rate (or stepsize) |
| $\tilde{\eta}$ | effective learning rate ($\tilde{\eta} := MK\eta$ in SFL and SPFL and $\tilde{\eta} := K\eta$ in PFL) |
| $\mu$ | $\mu$-strong convexity constant |
| $L$ | $L$-smoothness constant (Asm. 2) |
| $\sigma$ | upper bound on variance of stochastic gradients at each client (Asm. 3) |
| $\beta, \zeta$ | constants in Asm. 3a to bound heterogeneity everywhere |
| $\zeta_*$ | constants in Asm. 3b to bound heterogeneity at the optima |
| $\mathcal{L}/\mathcal{L}_m$ | global objective/local objective of client $m$ |
| $\boldsymbol{w}^{(r)}$ | global model parameters in the $r$-th round |
| $\boldsymbol{w}_g$ | the global model $f(\cdot; w_g)$ where $w_g$ represents the global model parameters. |
| $\boldsymbol{w}_i$ | Represents the parameters of the model during training,and $\boldsymbol{w}_i = \boldsymbol{w}_{intermediate}$ |
| $\boldsymbol{w}_p$ | Indicates the parameters of the model passed from the previous client training, and $\boldsymbol{w}_p = \boldsymbol{w}_{previous}$ |
| $\boldsymbol{w}_{m,k,n}^{(r)}$ | When client $n$ is the starting client, local model parameters of the $m$-th client after $k$ local steps in the $r$-th round |
| $\mathbf{g}_{\pi_m^n, k}^{(r)}$ | When client $n$ is the starting client , $\mathbf{g}_{\pi_m^n, k}^{(r)} := \nabla f_{\pi_m^n}(\boldsymbol{w}_{m,k,n}^{(r)}; \xi)$ denotes the stochastic gradients of $\mathcal{L}_{\pi_m^n}$ regarding $\boldsymbol{w}_{m,k,n}^{(r)}$ |

**Assumption 2** (L-Smoothness). *Each local objective function $\mathcal{L}_m$ is L-smooth,$m \in \{1, 2, \ldots, M\}$, i.e. , there exists a constant $L > 0$ such that $\|\nabla\mathcal{L}_m(\boldsymbol{x}) - \nabla\mathcal{L}_m(\mathbf{y})\| \leq L\|\boldsymbol{x} - \mathbf{y}\|$,for all $\boldsymbol{x}, \boldsymbol{y} \in \mathbb{R}$.*

**Assumption 3.** *The variance of the stochastic gradient at each client is bounded:*

$$\mathbb{E}_{\xi \sim \mathcal{D}_{\pi_m^n}}\left[\left\|\nabla\mathcal{L}_{\pi_m^n}(\boldsymbol{w}; \xi) - \nabla\mathcal{L}_{\pi_m^n}(\boldsymbol{w})\right\|^2 |\boldsymbol{w}\right] \leq \sigma^2, \quad \forall m \in \{1, 2, \ldots, M\} \tag{16}$$

**Assumption 4.** *For strongly convex and generally convex functions, there exist constants $\beta^2$ and $\zeta^2$ such that*

$$\frac{1}{M}\sum_{m=1}^{M}\|\nabla\mathcal{L}_m(\boldsymbol{w}) - \nabla\mathcal{L}(\boldsymbol{w})\|^2 \leq \beta^2\|\nabla\mathcal{L}(\boldsymbol{w})\|^2 + \zeta^2 \tag{17}$$

**Assumption 5.** *For non-convex functions, there exists one constant $\zeta_*^2$ such that*

$$\frac{1}{M}\sum_{m=1}^{M}\|\nabla\mathcal{L}_m(\boldsymbol{w}^*)\|^2 = \zeta_*^2 \tag{18}$$

# F   Technical Lemmas

## F.1   Basic Identities and Inequalities

These identities and inequalities are mostly from [53],[54],[55],[19],[56] and [31].Thanks for the analytical ideas provided by these works.

**E-norm identity**. (1) For any random variable **x**, letting the variance can be decomposed as

$$\mathbb{E}\left[\|\boldsymbol{w} - \mathbb{E}[\boldsymbol{w}]\|^2\right] = \mathbb{E}\left[\|\boldsymbol{w}\|^2\right] - \|\mathbb{E}[\boldsymbol{w}]\|^2 \tag{19}$$

(2) In particular, its version for vectors with finite number of values gives

$$\frac{1}{n}\sum_{i=1}^{n}\|\boldsymbol{x}_i - \bar{\boldsymbol{x}}\|^2 = \frac{1}{n}\sum_{i=1}^{n}\|\boldsymbol{x}_i\|^2 - \left\|\frac{1}{n}\sum_{i=1}^{n}\boldsymbol{x}_i\right\|^2 \tag{20}$$

where vectors $\boldsymbol{w}_1, \ldots, \boldsymbol{w}_n \in \mathbb{R}^d$ are the values of **w** and their average is $\bar{\boldsymbol{w}} = \frac{1}{n}\sum_{i=1}^{n}\boldsymbol{w}_i$.

**Lemma 1.** *Under standard discrete summation rules, the following closed-form expressions hold for any integer $K \geq 1$:*

$$\sum_{k=1}^{K-1} k = \frac{(K-1)K}{2}, \quad \sum_{k=1}^{K-1} k^2 = \frac{(K-1)K(2K-1)}{6}, \quad \sum_{k=1}^{K-1} k^3 = \left(\frac{(K-1)K}{2}\right)^2 \tag{21}$$

**Lemma 2** (Subadditivity of Concave Power Functions). *Let $0 < n < 1$ and $a, b > 0$. The power function $f(x) = x^n$ is concave, and the following inequality holds:*

$$(a+b)^n \leq a^n + b^n$$

**Jensen's inequality.** For any convex function $h$ and any vectors $\boldsymbol{x}_1, \ldots, \boldsymbol{x}_n$ we have

$$h\left(\frac{1}{n}\sum_{i=1}^{n}\boldsymbol{x}_i\right) \leq \frac{1}{n}\sum_{i=1}^{n}h(\boldsymbol{x}_i) \tag{22}$$

As a special case with $h(\boldsymbol{x}) = \|\boldsymbol{x}\|^2$, we obtain

$$\left\|\frac{1}{n}\sum_{i=1}^{n}x_i\right\|^2 \leq \frac{1}{n}\sum_{i=1}^{n}\|x_i\|^2 \tag{23}$$

**Smoothness and general convexity, strong convexity.** There are some useful inequalities concerning L-smoothness (Assumption 2), convexity, and $\mu - strong\ convexity$. Their proofs can be found in [53] and [56].

*Bregman Divergence* associated with function $h$ and arbitrary $\boldsymbol{x}, \boldsymbol{y}$ is denoted as

$$D_h(\boldsymbol{x}, \boldsymbol{y}) := h(\boldsymbol{x}) - h(\boldsymbol{y}) - \langle \nabla h(\boldsymbol{y}), \boldsymbol{x} - \boldsymbol{y}\rangle \tag{24}$$

The divergence is strictly non-negative when the function $h$ is convex. A more formal definition can be found in [57]. One corollary[58] called the three-point identity, is

$$D_h(\boldsymbol{z}, \boldsymbol{x}) + D_h(\boldsymbol{x}, \boldsymbol{y}) - D_h(\boldsymbol{z}, \boldsymbol{y}) = \langle \nabla h(\boldsymbol{y}) - \nabla h(\boldsymbol{x}), \boldsymbol{z} - \boldsymbol{x}\rangle \tag{25}$$

where $\boldsymbol{x}, \boldsymbol{y}, \boldsymbol{z}$ are three points in the set.

When $h$ is *L-smooth*, with the definition of *Bregman divergence*, a consequence of *L-smoothness* is

$$D_h(\boldsymbol{x}, \boldsymbol{y}) = h(\boldsymbol{x}) - h(\boldsymbol{y}) - \langle \nabla h(\boldsymbol{y}), \boldsymbol{x} - \boldsymbol{y}\rangle \leq \frac{L}{2}\|\boldsymbol{x} - \boldsymbol{y}\|^2 \tag{26}$$

Further, if $h$ is *L-smooth* and lower bounded by $h_*$, then

$$\|\nabla h(\boldsymbol{x})\|^2 \leq 2L\left(h(\boldsymbol{x}) - h_*\right) \tag{27}$$

If $h$ is smooth and convex in $L$ (the definition of convexity can be found in [59]), then

$$D_h(\boldsymbol{x}, \boldsymbol{y}) \geq \frac{1}{2L}\|\nabla h(\boldsymbol{x}) - \nabla h(\boldsymbol{y})\|^2 \tag{28}$$

The function $h : \mathbb{R}^d \to \mathbb{R}$ is $\mu - strongly$ convex if and only if there exists a convex function $g : \mathbb{R}^d \to \mathbb{R}$ such that $h(\boldsymbol{x}) = g(\boldsymbol{x}) + \frac{\mu}{2}\|\boldsymbol{x}\|^2$.

If $h$ is $\mu - strongly$ convex, it holds that

$$\frac{\mu}{2}\|x - \boldsymbol{y}\|^2 \leq D_h(\boldsymbol{x}, \boldsymbol{y}) \tag{29}$$

## F.2 Technical Lemmas

We obtain a recurrence relation suitable for SPFL based on previous work[31, 19], as shown below: Lemma .3 ,Lemma .4 and lemma .5. Two nonnegative sequences $\{r_t\}_{r\geq 0}$ , $\{s_t\}_{r\geq 0}$ , which satisfy the relation

$$r_{t+1} \leq (1 - a\gamma_t)r_t - b\gamma_t s_t + c_1\gamma_t^2 + c_2\gamma_t^3, \tag{30}$$

for all $t \geq 0$ and for parameters $b > 0$, $a, c_1, c_2 \geq 0$ and non-negative learning rates $\{\gamma_t\}_{r\geq 0}$ with $\gamma_t < \frac{1}{d}$, $\forall t \geq 0$, for a parameter $d \geq a$, $d > 0$.

**Lemma 3** (linear convergence rate and Constant Step sizes). $\{r_t\}_{r\geq 0}$ , $\{r_t\}_{r\geq 0}$ as in (30) and $a > 0$. Then there exists a constant step size $\gamma_t = \gamma \leq \frac{1}{d}$ such that for weights $\theta_t := (1 - a\gamma)^{-(t+1)}$ and $W_t := \sum_{t=0}^T w_t$ it holds:

$$\Psi_T = \frac{b}{W_T} \sum_{t=0}^T s_t\theta_t \leq 3ar_0(1 - a\gamma)^{(T+1)} + c_1\gamma + c_2\gamma^2 \tag{31}$$

$$\leq 3ar_0 \exp\left[-a\gamma(T+1)\right] + c_1\gamma + c_2\gamma^2$$

$$\frac{b}{W_T} \sum_{t=0}^T s_t\theta_t + ar_{T+1} = \tilde{\mathcal{O}}\left(dr_0 \exp\left[-\frac{aT}{d}\right] + \frac{c_1}{aT} + \frac{c_2}{a^2T^2}\right) \tag{32}$$

*Proof.* We start by rearranging (30) and multiplying both sides with $\theta_t$:

$$bs_t\theta_t \leq \frac{\theta_t(1-a\gamma)r_t}{\gamma} - \frac{\theta_t r_{t+1}}{\gamma} + c_1\gamma\theta_t + c_2\gamma^2\theta_t = \frac{w_{t-1}r_t}{\gamma} - \frac{\theta_t r_{t+1}}{\gamma} + c_1\gamma\theta_t + c_2\gamma^2\theta_t$$

By summing from $t = 0$ to $t = T$ , we obtain a telescoping sum:

$$\frac{b}{W_T} \sum_{t=0}^T s_t\theta_t \leq \frac{1}{\gamma W_T}\left(w_0(1-a\gamma)r_0 - w_T r_{T+1}\right) + c_1\gamma + c_2\gamma^2$$

and hence

$$\Psi_T = \frac{b}{W_T} \sum_{t=0}^T s_t\theta_t \leq \frac{b}{W_T} \sum_{t=0}^T s_t\theta_t + \frac{w_T r_{T+1}}{\gamma W_T} \leq \frac{r_0}{\gamma W_T} + c_1\gamma + c_2\gamma^2$$

With the estimates

- $W_T = (1 - a\gamma)^{-(T+1)} \sum_{t=0}^T (1 - a\gamma)^t \leq \frac{w_T}{a\gamma}$ (here we leverage $a\gamma \leq \frac{a}{d} \leq 1$)
- and $W_T \geq w_T = (1 - a\gamma)^{-(T+1)}$

We can further simplify the left and right-hand sides:

$$\frac{b}{W_T} \sum_{t=0}^T s_t w_t + ar_{T+1} \leq (1 - a\gamma)^{(T+1)}\frac{r_0}{\gamma} + c_1\gamma + c_2\gamma^2 \leq \frac{r_0}{\gamma} \exp\left[-a\gamma(T+1)\right] + c_1\gamma + c_2\gamma^2$$

Now the lemma follows by carefully tuning $\gamma$. $\gamma = \frac{\ln(\max\{2, a^2 r_0 T^2/c_1\})}{aT}$ and $\gamma = \frac{\ln(\max\{2, a^3 r_0 T^3/c_2\})}{aT}$ ,yielding two choices of $\gamma$.Consider the three cases:

- If $\frac{1}{d} \geq \frac{\ln(\max\{2, a^2 r_0 T^2/c_1\})}{aT}$ then we choose $\gamma = \frac{\ln(\max\{2, a^2 r_0 T^2/c_1\})}{aT}$ and get that Eq.32:

$$\tilde{\mathcal{O}}\left(ar_0 T \exp[-\ln(\max\{2, a^2 r_0 T^2/c_1\})]\right) + \tilde{\mathcal{O}}\left(\frac{c_1}{aT} + \frac{c_2}{a^2T^2}\right) = \tilde{\mathcal{O}}\left(\frac{c_1}{aT} + \frac{c_2}{a^2T^2}\right)$$

  as in case $2 \geq a^2 r_0 T^2/c_1$ it holds $ar_0 T \leq \frac{2c_1}{aT}$.

- If $\frac{1}{d} \geq \frac{\ln(\max\{2, a^3 r_0 T^3/c_2\})}{aT}$ then we choose $\gamma = \frac{\ln(\max\{2, a^3 r_0 T^3/c_2\})}{aT}$ and get that Eq.32:

$$\tilde{\mathcal{O}}\left(ar_0 T \exp[-\ln(\max\{2, a^3 r_0 T^3/c_2\})]\right) + \tilde{\mathcal{O}}\left(\frac{c_1}{aT} + \frac{c_2}{a^2T^2}\right) = \tilde{\mathcal{O}}\left(\frac{c_1}{aT} + \frac{c_2}{a^2T^2}\right)$$

  as in case $2 \geq a^3 r_0 T^3/c_2$ it holds $ar_0 T \leq \frac{2c_2}{a^2T^2}$.

- If otherwise $\frac{1}{d} < \frac{\ln(\max\{2,a^2 r_0 T^2/c_1, a^3 r_0 T^3/c_2\})}{aT}$ then we pick $\gamma = \frac{1}{d}$ and get that Eq.32:

$$dr_0 \exp\left[-\frac{aT}{d}\right] + \frac{c_1}{d} + \frac{c_2}{d^2}$$

$$\leq dr_0 \exp\left[-\frac{aT}{d}\right] + \frac{c_1 \ln(\max\{2, a^2 r_0 T^2/c_1, a^3 r_0 T^3/c_2\})}{aT}$$

$$+ \frac{c_2 \ln^2(\max\{2, a^2 r_0 T^2/c_1, a^3 r_0 T^3/c_2\})}{a^2 T^2}$$

$$= \tilde{\mathcal{O}}\left(dr_0 \exp\left[-\frac{aT}{d}\right] + \frac{c_1}{aT} + \frac{c_2}{a^2 T^2}\right)$$

**Lemma 4** (linear convergence rate and Decreasing Step sizes). $\{r_t\}_{r\geq 0}$, $\{s_t\}_{r\geq 0}$ as in (30) and $a > 0$. Then there exist step sizes $\gamma_t = \gamma \leq \frac{1}{d}$ and weights $\theta_t \geq 0$, $W_t := \sum_{t=0}^{T} w_t$, such that:

$$\frac{b}{W_T}\sum_{t=0}^{T} s_t\theta_t + ar_{T+1} \leq 32dr_0 \exp\left[-\frac{aT}{2d}\right] + \frac{36c_1}{aT} + \frac{36c_2}{adT} \tag{33}$$

*Proof.* Let $\{r_t\}_{r\geq 0}$, $\{s_t\}_{r\geq 0}$ be as in (30) for $a > 0$ and for constant stepsizes $\gamma_t := \gamma = \frac{1}{d}$, $\forall t > 0$. Then, it holds for all $T \geq 0$. We have

$$r_T \leq (1-a\gamma)r_{T-1} + c_1\gamma^2 + c_2\gamma^3 \leq (1-a\gamma)^T r_0 + c_1\gamma^2 \sum_{t=0}^{T-1}(1-a\gamma)^t + +c_2\gamma^3 \sum_{t=0}^{T-1}(1-a\gamma)^t$$

$$\leq (1-a\gamma)^T r_0 + \frac{c_1\gamma}{a} + +\frac{c_2\gamma^2}{a}$$

$$\leq r_0 \exp\left[-\frac{aT}{d}\right] + \frac{c_1}{ad} + +\frac{c_2}{ad^2} \tag{34}$$

Let $\{r_t\}_{r\geq 0}$, $\{s_t\}_{r\geq 0}$ as in (30) for $a > 0$ and for decreasing steps $\gamma_t := \frac{2}{a(\kappa+t)}$, $\forall t > 0$, with parameter $\kappa := \frac{2d}{a}$, weights $w_t := (\kappa+t)$ and $W_T := \sum_{t=0}^{T}\theta_t$. Then

$$bs_t w_t \leq \frac{w_t(1-a\gamma_t)r_t}{\gamma_t} - \frac{w_t r_{t+1}}{\gamma_t} + c_1\gamma_t w_t + +c_2\gamma_t^2 w_t$$

$$= a(\kappa+t)(\kappa+t-2)r_t - a(\kappa+t)^2 r_{t+1} + \frac{c_1}{a} + \frac{c_2}{a}\gamma_t \tag{35}$$

$$\leq a(\kappa+t-1)^2 r_t - a(\kappa+t)^2 r_{t+1} + \frac{c_1}{a} + \frac{c_2}{ad}$$

where the equality follows from the definition of $\gamma_t$ and $\theta_t$, and the inequality from $(\kappa+t)(\kappa+t-2) = (\kappa+t-1)^2 - 1 \leq (\kappa+t-1)^2$. Again, we have a telescoping sum:

$$\frac{b}{W_T}\sum_{t=0}^{T} s_t w_t + \frac{a(\kappa+T)^2 r_{T+1}}{W_T} \leq \frac{a\kappa^2 r_0}{W_T} + \frac{c_1(T+1)}{aW_T} + \frac{c_2(T+1)}{adW_T}$$

$$\leq \frac{2a\kappa^2 r_0}{T^2} + \frac{2c_1}{aT} + +\frac{2c_2}{adT} \tag{36}$$

with

- $W_T = \sum_{t=0}^{T}\theta_t = \sum_{t=0}^{T}(\kappa+t) = \frac{(2\kappa+T)(T+1)}{2} \geq \frac{T(T+1)}{2} \geq \frac{T^2}{2}$,
- and $W_T = \frac{(2\kappa+T)(T+1)}{2} \leq \frac{2(\kappa+T)(1+T)}{2} \leq (\kappa+T)^2$ for $\kappa = \frac{2d}{a} \geq 1$

By applying (34) and (36), we conclude the proof.

For the integer $T \geq 0$, we choose the step sizes and weights as follows:

$$if \quad T \leq \frac{d}{a}, \qquad \gamma_t = \frac{1}{d}, \qquad \theta_t = (1-a\gamma_t)^{-(t+1)} = \left(1-\frac{a}{d}\right)^{-(t+1)},$$

$$\text{if} \quad T > \tfrac{d}{a} \text{and} t < t_0, \quad \gamma_t = \tfrac{1}{d}, \quad \theta_t = 0,$$

$$\text{if} \quad T > \tfrac{d}{a} \text{and } t \geq t_0, \quad \gamma_t = \tfrac{2}{a(\kappa+t-t_0)}, \quad \theta_t = (\kappa + t - t_0)^2,$$

for $\kappa = \tfrac{2d}{a}$ and $t_0 = \lceil \tfrac{T}{2} \rceil$. We will now show that these choices imply the claimed result.

We start with the case $T \leq \tfrac{d}{a}$. This case is similar to the proof of the Lemma. 32 and it suffices to consider Eq .35 for the choice $\gamma_t = \tfrac{1}{d}$. We observe that Eq .35 simplifies to

$$dr_0 \exp\left[-\frac{aT}{d}\right] + \frac{c_1}{d} + \frac{c_2}{d^2} \leq dr_0 \exp\left[-\frac{aT}{d}\right] + \frac{c_1}{aT} + \frac{c_2}{a^2 T^2}$$

If $T > \tfrac{d}{a}$, then from Eq.34 we obtain the following :

$$r_{t_0} \leq r_0 \exp\left[-\frac{aT}{2d}\right] + \frac{c_1}{ad} + + \frac{c_2}{ad^2}$$

From Eq .36 we have for the second half of the iterates:

$$\frac{b}{W_T} \sum_{t=0}^{T} s_t \theta_t + ar_{T+1} = \frac{b}{W_T} \sum_{t=t_0}^{T} s_t \theta_t + ar_{T+1} \leq \frac{8a\kappa^2 r_{t_0}}{T^2} + \frac{4c_1}{aT} + \frac{4c_2}{adT}$$

Now we observe that the restart condition $r_{t_0}$ satisfies:

$$\frac{a\kappa^2 r_{t_0}}{T^2} = \frac{a\kappa^2 r_0 \exp\left(-\frac{aT}{2d}\right)}{T^2} + \frac{\kappa^2 c}{dT^2} + \frac{\kappa^2 c_2}{d^2 T^2} \leq 4ar_0 \exp\left[-\frac{aT}{2d}\right] + \frac{4c_1}{aT} + \frac{4c_2}{adT}$$

Because $T \geq \tfrac{d}{a}$. These inequalities show the claim:

$$\frac{b}{W_T} \sum_{t=0}^{T} s_t \theta_t + ar_{T+1} \leq 32dr_0 \exp\left[-\frac{aT}{2d}\right] + \frac{36c_1}{aT} + \frac{36c_2}{adT}$$

**Lemma 5** ( Sub-linear Convergence rate from [31]). $\{r_t\}_{r\geq 0}$ , $\{r_t\}_{r\geq 0}$ *as in (30) and $a = 0$. Then there exists the step size $\gamma_t \leq \tfrac{1}{d}$ such that for weights $\theta_t := 1$ and $W_t := \sum_{t=0}^{T} w_t$ it holds:*

*When our step size is a constant, that is, $\gamma_t = \gamma \leq \tfrac{1}{d}$, we have*

$$\Psi_T := \frac{b}{T+1} \sum_{t=0}^{T} s_t \leq \frac{r_0}{\gamma(T+1)} + c_1\gamma + c_2\gamma^2$$

*When we dynamically adjust the step size $\gamma_t$, we have*

$$\Psi_T \leq 2c_1^{\frac{1}{2}} \left(\frac{r_0}{T+1}\right)^{\frac{1}{2}} + 2c_2^{\frac{1}{3}} \left(\frac{r_0}{T+1}\right)^{\frac{2}{3}} + \frac{dr_0}{T+1} \tag{37}$$

*Proof.* For constant learning rates $\gamma_t = \gamma \leq \tfrac{1}{d}$ we can derive the estimate

$$\Psi_T = \frac{1}{\gamma(T+1)} \sum_{t=0}^{T} (r_t - r_{t+1}) + c_1\gamma + c_2\gamma^2 \leq \frac{r_0}{\gamma(T+1)} + c_1\gamma + c_2\gamma^2$$

which is the first result of this lemma. 5. Let $\frac{r_0}{\gamma(T+1)} = c_1\gamma$ and $\frac{r_0}{\gamma(T+1)} = c_2\gamma^2$ , yielding two choices of $\gamma$ ,$\gamma = \left(\frac{r_0}{c_1(T+1)}\right)^{\frac{1}{2}}$ and $\gamma = \left(\frac{r_0}{c_2(T+1)}\right)^{\frac{1}{3}}$ .Then choosing $\gamma = \min\left\{\left(\frac{r_0}{c_1(T+1)}\right)^{\frac{1}{2}}, \left(\frac{r_0}{c_2(T+1)}\right)^{\frac{1}{3}}, \frac{1}{d}\right\} \leq \frac{1}{d}$ , there are three cases:

If $\gamma = \frac{1}{d}$, which implies that $\gamma = \frac{1}{d} \le \left(\frac{r_0}{c_1(T+1)}\right)^{\frac{1}{2}}$ and $\gamma = \frac{1}{d} \le \left(\frac{r_0}{c_2(T+1)}\right)^{\frac{1}{3}}$, then:

$$\Psi_T \le \frac{dr_0}{T+1} + \frac{c_1}{d} + \frac{c_2}{d^2} \le \frac{dr_0}{T+1} + c_1^{\frac{1}{2}}\left(\frac{r_0}{T+1}\right)^{\frac{1}{2}} + c_2^{\frac{1}{3}}\left(\frac{r_0}{T+1}\right)^{\frac{2}{3}}$$

If $\gamma = \left(\frac{r_0}{c_1(T+1)}\right)^{\frac{1}{2}}$, which implies that $\gamma = \left(\frac{r_0}{c_1(T+1)}\right)^{\frac{1}{2}} \le \left(\frac{r_0}{c_2(T+1)}\right)^{\frac{1}{3}}$, then:

$$\Psi_T \le 2c_1\left(\frac{r_0}{c_1(T+1)}\right)^{\frac{1}{2}} + c_2\left(\frac{r_0}{c_1(T+1)}\right) \le 2c_1^{\frac{1}{2}}\left(\frac{r_0}{T+1}\right)^{\frac{1}{2}} + c_2^{\frac{1}{3}}\left(\frac{r_0}{T+1}\right)^{\frac{2}{3}}$$

If $\gamma = \left(\frac{r_0}{c_2(T+1)}\right)^{\frac{1}{3}}$, which implies that $\gamma = \left(\frac{r_0}{c_2(T+1)}\right)^{\frac{1}{3}} \le \left(\frac{r_0}{c_1(T+1)}\right)^{\frac{1}{2}}$, then:

$$\Psi_T \le c_1\left(\frac{r_0}{c_2(T+1)}\right)^{\frac{1}{3}} + 2c_2^{\frac{1}{3}}\left(\frac{r_0}{T+1}\right)^{\frac{2}{3}} \le c_1^{\frac{1}{2}}\left(\frac{r_0}{T+1}\right)^{\frac{1}{2}} + 2c_2^{\frac{1}{3}}\left(\frac{r_0}{T+1}\right)^{\frac{2}{3}}$$

Combining these three cases, we get the second result of this lemma :

$$\Psi_T \le 2c_1^{\frac{1}{2}}\left(\frac{r_0}{T+1}\right)^{\frac{1}{2}} + 2c_2^{\frac{1}{3}}\left(\frac{r_0}{T+1}\right)^{\frac{2}{3}} + \frac{dr_0}{T+1}$$

**Lemma 6** (Simple Random Sampling from [31]). *Let $\boldsymbol{w_1}, \boldsymbol{w_2}, \ldots, \boldsymbol{w_n}$ be fixed units (e.g., vectors). The population mean and population variance are given as*

$$\overline{w} := \frac{1}{n}\sum_{i=1}^{n} w_i \quad \zeta^2 := \frac{1}{n}\sum_{i=1}^{n} \|w_i - \overline{w}\|^2 \tag{38}$$

*Draw $s \in [n] = \{1, 2, \ldots, n\}$ random units $\boldsymbol{w}_{\pi_1}, \boldsymbol{w}_{\pi_2}, \ldots \boldsymbol{w}_{\pi_s}$ randomly from the population. There are two possible ways of simple random sampling, well known as "sampling with replacement (SWR)" and "sampling without replacement (SWOR)". For these two ways, the expectation and variance of the sample mean $\overline{\boldsymbol{w}}_\pi := \frac{1}{s}\sum_{p=1}^{s} \boldsymbol{w}_{\pi_p}$ satisfy*

$$SWR \quad : \quad \mathbb{E}[\overline{\boldsymbol{w}}_\pi] = \overline{\boldsymbol{w}} \qquad \mathbb{E}\left[\|\overline{\boldsymbol{w}}_\pi - \overline{\boldsymbol{w}}\|^2\right] = \frac{\zeta^2}{s} \tag{39}$$

$$SWOR: \quad \mathbb{E}[\overline{\boldsymbol{w}}_\pi] = \overline{\boldsymbol{w}} \qquad \mathbb{E}\left[\|\overline{\boldsymbol{w}}_\pi - \overline{\boldsymbol{w}}\|^2\right] = \frac{n-s}{s(n-1)}\zeta^2 \tag{40}$$

*Proof.* We can easily get the relationship between variance and expectation, as well as Eq.19 and Eq.20. If you want this proof, refer to [31].

**Lemma 7.** *Under the same conditions of Lemma 6, use the way "sampling without replacement" and let $b_{m,k}(i) = \begin{cases} K-1, & i \le m-1 \\ k-1, & i = m \end{cases}$ Then for $S \le M(M \ge 2)$, it holds that*

$$\frac{1}{S}\sum_{n=1}^{S}\sum_{m=1}^{S}\sum_{k=0}^{K-1}\mathbb{E}\left[\left\|\sum_{i=1}^{m}\sum_{j=0}^{b_{m,k}(i)}(\boldsymbol{w}_{\pi_i^n} - \overline{\boldsymbol{w}})\right\|^2\right] \le \frac{1}{2}S^2 K^3 \zeta^2 \tag{41}$$

*Proof.* As shown below, the idea refers to [31]:

$$\mathbb{E}\left[\left\|\sum_{i=1}^{m}\sum_{j=0}^{b_{m,k}(i)}(\boldsymbol{w}_{\pi_i^n} - \overline{\boldsymbol{w}})\right\|^2\right] = \mathbb{E}\left[\left\|K\sum_{i=1}^{m-1}(\boldsymbol{w}_{\pi_i^n} - \overline{\boldsymbol{w}}) + k(\boldsymbol{w}_{\pi_m^n} - \overline{\boldsymbol{w}})\right\|^2\right]$$

$$= K^2\mathbb{E}\left[\left\|\sum_{i=1}^{m-1}(\boldsymbol{w}_{\pi_i^n} - \overline{\boldsymbol{w}})\right\|^2\right] + k^2\mathbb{E}\left[\|\boldsymbol{w}_{\pi_m^n} - \overline{\boldsymbol{w}}\|^2\right] + 2Kk\mathbb{E}\left[\left\langle\sum_{i=1}^{m-1}(\boldsymbol{w}_{\pi_i^n} - \overline{\boldsymbol{w}}), (\boldsymbol{w}_{\pi_m^n} - \overline{\boldsymbol{w}})\right\rangle\right]$$

For the first term on the right-hand side in Eq.40, using Lemma.6, we have

$$K^2 \mathbb{E}\left[\left\|\sum_{i=1}^{m-1}\left(\boldsymbol{w}_{\pi_i^n}-\overline{\boldsymbol{w}}\right)\right\|^2\right] \stackrel{(6)}{=} \frac{(m-1)(M-(m-1))}{M-1}K^2\zeta^2$$

For the second term on the right-hand side in Eq.40, we have

$$k^2\mathbb{E}\left[\left\|\boldsymbol{w}_{\pi_m^n}-\overline{\boldsymbol{w}}\right\|^2\right] = k^2\mathbb{E}\left[\left\|\boldsymbol{w}_{\pi_m^n}-\overline{\boldsymbol{w}}\right\|^2\right] = k^2\zeta^2$$

For the third term on the right-hand side in Eq.40, we have

$$2Kk\mathbb{E}\left[\left\langle \sum_{i=1}^{m-1}\left(\boldsymbol{w}_{\pi_i^n}-\overline{\boldsymbol{w}}\right),\left(\boldsymbol{w}_{\pi_m^n}-\overline{\boldsymbol{w}}\right)\right\rangle\right] = 2Kk\sum_{i=1}^{m-1}\mathbb{E}\left[\left\langle \boldsymbol{w}_{\pi_i^n}-\overline{\boldsymbol{w}},\boldsymbol{w}_{\pi_m^n}-\overline{\boldsymbol{w}}\right\rangle\right] \stackrel{(6)}{=} -\frac{2(m-1)}{M-1}Kk\zeta^2$$

where we use Lemma.6 in the last equality, since $i \in \{1,2,\ldots,m-1\} \neq m$. With these three preceding equations, we get

$$\mathbb{E}\left[\left\|\sum_{i=1}^{m}\sum_{j=0}^{b_{m,k}(i)}\left(\boldsymbol{w}_{\pi_i^n}-\overline{\boldsymbol{w}}\right)\right\|^2\right] = \frac{(m-1)(M-(m-1))}{M-1}K^2\zeta^2 + k^2\zeta^2 - \frac{2(m-1)}{M-1}Kk\zeta^2$$

Then summing the preceding terms over $m$ and $k$, we can get

$$\frac{1}{S}\sum_{n=1}^{S}\sum_{m=1}^{S}\sum_{k=0}^{K-1}\mathbb{E}\left[\left\|\sum_{i=1}^{m}\sum_{j=0}^{b_{m,k}(i)}\left(\boldsymbol{w}_{\pi_i^n}-\overline{\boldsymbol{w}}\right)\right\|^2\right]$$

$$= \frac{MK^3\zeta^2}{S(M-1)}\sum_{n=1}^{S}\sum_{m=1}^{S}(m-1) - \frac{K^3\zeta^2}{S(M-1)}\sum_{n=1}^{S}\sum_{m=1}^{S}(m-1)^2 + S\zeta^2\sum_{k=0}^{K-1}k^2$$

$$- \frac{2K\zeta^2}{S(M-1)}\sum_{n=1}^{S}\sum_{m=1}^{S}(m-1)\sum_{k=0}^{K-1}k$$

Then, applying the lemma.1, we can simplify the preceding equation as follows:

$$\frac{1}{S}\sum_{n=1}^{S}\sum_{m=1}^{S}\sum_{k=0}^{K-1}\mathbb{E}\left\|\sum_{i=1}^{m}\sum_{j=0}^{b_{m,k}(i)}\left(\boldsymbol{w}_{\pi_i^n}-\overline{\boldsymbol{w}}\right)\right\|^2$$

$$= \frac{1}{2}SK^2(SK-1) - \frac{1}{6}SK(K^2-1) - \frac{1}{M-1}(S-1)S\left(\frac{1}{6}(2S-1)K - \frac{1}{2}\right) \leq \frac{1}{2}S^2K^3\zeta^2$$

**Lemma 8** (from [19]). *Let $\{\xi_i\}_{i=1}^n$ be a sequence of random variables. And the random sequence $\{\boldsymbol{w}_i\}_{i=1}^n$ satisfy that $\boldsymbol{w}_i \in \mathbb{R}^d$ is a function of $\xi_i, \xi_{i-1}, \ldots, \xi_1$ for all $i$. Suppose that the conditional expectation is $\mathbb{E}_{\xi_i}[\boldsymbol{w}_i|\xi_{i-1},\ldots,\xi_1] = \mathbf{e}_i$ (i.e., the vectors $\{\boldsymbol{w}_i - \mathbf{e}_i\}$ form a martingale difference sequence with respect to $\{\xi_i\}$), and the variance is bounded by $\mathbb{E}_{\xi_i}\left[\|\boldsymbol{w}_i - \mathbf{e}_i\|^2\Big|\xi_{i-1},\ldots,\xi_1\right] \leq \sigma^2$. Then it holds that*

$$\mathbb{E}\left[\left\|\sum_{i=1}^{n}(\boldsymbol{w}_i - \mathbf{e}_i)\right\|^2\right] = \sum_{i=1}^{n}\mathbb{E}\left[\|\boldsymbol{w}_i - \mathbf{e}_i\|^2\right] \leq n\sigma^2 \tag{42}$$

*Proof.* For details, please refer to [19]'s Lemma 4 and [31]'s Lemma 1.

**Lemma 9** (from [19]). *The following holds for any L-smooth and $\mu$-strongly convex function $h$, and any $\boldsymbol{x}, \boldsymbol{y}, \boldsymbol{z}$ in the domain of $h$:*

$$\langle \nabla h(\boldsymbol{x}), \boldsymbol{z} - \boldsymbol{y}\rangle \geq h(\boldsymbol{z}) - h(\mathbf{y}) + \frac{\mu}{4}\|\boldsymbol{y} - \boldsymbol{z}\|^2 - L\|\boldsymbol{z} - \boldsymbol{x}\|^2 \tag{43}$$

*Proof.* It can be easily obtained using the three-point identity (24) and Jensen's inequality (22). For details, please refer to [31]. Here, we only use relevant conclusions to assist our proof.

# G    Proofs of Theorem 1

In this section, we provide the proof of Theorem 1 for the strongly convex, general convex, and non-convex cases in G.1, G.2 and G.3, respectively.

In the following proof, we consider the partial client participation setting. So we assume that $\pi^n = \{\pi_1^n, \pi_2^n, \ldots, \pi_M^n\}$ is a permutation of $\{1, 2, \ldots, M\}$ in a certain training round and only the first $S$ selected clients $\{\pi_1^n, \pi_2^n, \ldots, \pi_S^n\}$ will participate in this round. Without otherwise stated, we use E[·] to represent the expectation concerning both types of randomness (i.e., sampling data samples $\zeta$ and sampling clients $\pi$). $\mathcal{R}$ represents rotation, that is, $\pi^{(n+1)} = \left[\pi_2^{(n+1)}, \pi_3^{(n+1)}, \ldots, \pi_M^{(n+1)}, \pi_1^{(n+1)}\right] = \mathcal{R}\left(\pi^{(n)}\right) := \left[\pi_2^{(n)}, \pi_3^{(n)}, \ldots, \pi_M^{(n)}, \pi_1^{(n)}\right]$, where $n$ represents the client starting with client $n$.

## G.1    Strongly Convex Case

### G.1.1    Finding the recursion

**Lemma 10.** *Let Assumptions 2, 3, and 5 hold, and assume that all the local objectives are $\mu$-strongly convex. If the learning rate satisfies $\eta \leq \frac{1}{6LSK}$, then it holds that*

$$
\begin{aligned}
\mathbb{E}\left[\left\|\boldsymbol{w}^{(r+1)} - \boldsymbol{w}^*\right\|^2\right] \leq & \left(1 - \frac{\mu SK\eta}{2}\right)\|\boldsymbol{w} - \boldsymbol{w}^*\|^2 + 4K\eta^2\sigma^2 + 4S^2K^2\eta^2\frac{M-S}{S(M-1)}\zeta_*^2 \\
& - \frac{2}{3}SK\eta D_{\mathcal{L}}(\boldsymbol{w}, \boldsymbol{w}^*) + \frac{8}{3}\frac{L\eta}{S}\sum_{n=1}^{S}\sum_{m=1}^{S}\sum_{k=0}^{K-1}\mathbb{E}\left[\|\boldsymbol{w}_{m,k,n} - \boldsymbol{w}\|^2\right]
\end{aligned}
\tag{44}
$$

*Proof.*    According to the pseudocode of our algorithm, the overall model updates of SFL after one complete training round are shown in (45).However, to compare the subsequent convergence analysis, according to Assumption 1, this deviation can be incorporated into the $\sigma$ in Assumption 3. For client $m$, the update method is simplified as (45):

$$
\begin{aligned}
\boldsymbol{w}^{(r+1)} &= \boldsymbol{w}^{(r)} - \eta\frac{1}{M}\sum_{n=1}^{M}\sum_{m=1}^{M}\sum_{k=1}^{K}\sum_{i=1}^{q_{\pi_m^n,k}}\nabla\mathcal{L}_{\pi_m^n,k}^{(r)}\left(f(x_i^{\pi_m^n,k};\boldsymbol{w}_{\pi_m^n,k}), y_i^{\pi_m^n,k}\mid\sum_{j=1}^{m-1}\nabla\mathcal{L}_{\pi_j^n,k}\right) \\
&= \boldsymbol{w}^{(r)} - \eta\frac{1}{M}\sum_{n=1}^{M}\sum_{m=1}^{M}\sum_{k=0}^{K-1}\mathbf{g}_{\pi_m^n,k}^{(r)}
\end{aligned}
\tag{45}
$$

More generally, when $S$ represents the number of clients selected to participate, we have

$$
\Delta\boldsymbol{w} = \boldsymbol{w}^{(r+1)} - \boldsymbol{w}^{(r)} = -\eta\frac{1}{S}\sum_{n=1}^{S}\sum_{m=1}^{S}\sum_{k=0}^{K-1}\mathbf{g}_{\pi_m^n,k}^{(r)}
$$

where $\mathbf{g}_{\pi_m^n,k}^{(r)} = \nabla f_{\pi_m^n}(\boldsymbol{w}_{m,k,n}^{(r)};\xi)$ is the stochastic gradient of $\mathcal{L}_{\pi_m^n}$ regarding the vector $\boldsymbol{w}_{m,k,n}^{(r)}$. Thus,

$$
\mathbb{E}\left[\Delta\boldsymbol{w}\right] = -\eta\frac{1}{S}\sum_{n=1}^{S}\sum_{m=1}^{S}\sum_{k=0}^{K-1}\mathbb{E}\left[\nabla\mathcal{L}_{\pi_m^n}(\boldsymbol{w}_{m,k,n})\right]
$$

In the following, we focus on the recurrence of adjacent training rounds, so we omit the superscript $r$ for a while, e.g., writing $\boldsymbol{w}_{m,k,n}^r$ as $\boldsymbol{w}_{m,k,n}$. In particular, we would like to use $\boldsymbol{w}$ to replace $\boldsymbol{w}_{1,0,1}$. Without otherwise stated, the expectation is conditioned on $\boldsymbol{w}^r$.

We start by substituting the overall updates:

$$\mathbb{E}\left[\|\boldsymbol{w} + \Delta\boldsymbol{w} - \boldsymbol{w}^*\|^2\right]$$

$$= \|\boldsymbol{w} - \boldsymbol{w}^*\|^2 + 2\mathbb{E}\left[\langle\boldsymbol{w} - \boldsymbol{w}^*, \Delta\boldsymbol{w}\rangle\right] + \mathbb{E}\left[\|\Delta\boldsymbol{w}\|^2\right]$$

$$= \|\boldsymbol{w} - \boldsymbol{w}^*\|^2 - 2\eta\frac{1}{S}\sum_{n=1}^{S}\sum_{m=1}^{S}\sum_{k=0}^{K-1}\mathbb{E}\left[\langle\nabla\mathcal{L}_{\pi_m^n}(\boldsymbol{w}_{m,k,n}), \boldsymbol{w} - \boldsymbol{w}^*\rangle\right] + \eta^2\mathbb{E}\left[\left\|\frac{1}{S}\sum_{n=1}^{S}\sum_{m=1}^{S}\sum_{k=0}^{K-1}\mathbf{g}_{\pi_m^n,k}\right\|^2\right]$$

$$(46)$$

We can apply Lemma.9 with $\boldsymbol{x} = \boldsymbol{w}_{m,k,n}$ , $\boldsymbol{y} = \boldsymbol{w}^*$ , $\boldsymbol{z} = \boldsymbol{w}$ and $h = \mathcal{L}_{\pi_m^n}$ for the second term on the right-hand side in (46):

$$- 2\eta\frac{1}{S}\sum_{n=1}^{S}\sum_{m=1}^{S}\sum_{k=0}^{K-1}\mathbb{E}\left[\langle\nabla\mathcal{L}_{\pi_m^n}(\boldsymbol{w}_{m,k,n}), \boldsymbol{w} - \boldsymbol{w}^*\rangle\right]$$

$$\leq -2\eta\frac{1}{S}\sum_{n=1}^{S}\sum_{m=1}^{S}\sum_{k=0}^{K-1}\mathbb{E}\left[\mathcal{L}_{\pi_m^n}(\boldsymbol{w}) - \mathcal{L}_{\pi_m^n}(\boldsymbol{w}^*) + \frac{\mu}{4}\|\boldsymbol{w} - \boldsymbol{w}^*\|^2 - L\|\boldsymbol{w}_{m,k,n} - \boldsymbol{w}\|^2\right] \quad (47)$$

$$\leq -2SK\eta D_{\mathcal{L}}(\boldsymbol{w}, \boldsymbol{w}^*) - \frac{1}{2}\mu SK\eta\|\boldsymbol{w} - \boldsymbol{w}^*\|^2 + 2L\eta\frac{1}{S}\sum_{n=1}^{S}\sum_{m=1}^{S}\sum_{k=0}^{K-1}\mathbb{E}\left[\|\boldsymbol{w}_{m,k,n} - \boldsymbol{w}\|^2\right]$$

For the third term on the right-hand side in (47), using Jensen's inequality, we have

$$\mathbb{E}\left[\left\|\frac{1}{S}\sum_{n=1}^{S}\sum_{m=1}^{S}\sum_{k=0}^{K-1}\mathbf{g}_{\pi_m^n,k}\right\|^2\right]$$

$$\leq 4\mathbb{E}\left[\left\|\frac{1}{S}\sum_{n=1}^{S}\sum_{m=1}^{S}\sum_{k=0}^{K-1}\left(\mathbf{g}_{\pi_m^n,k} - \nabla\mathcal{L}_{\pi_m^n}(\boldsymbol{w}_{m,k,n})\right)\right\|^2\right]$$

$$+ 4\mathbb{E}\left[\left\|\frac{1}{S}\sum_{n=1}^{S}\sum_{m=1}^{S}\sum_{k=0}^{K-1}\left(\nabla\mathcal{L}_{\pi_m^n}(\boldsymbol{w}_{m,k,n}) - \nabla\mathcal{L}_{\pi_m^n}(\boldsymbol{w})\right)\right\|^2\right]$$

$$+ 4\mathbb{E}\left[\left\|\frac{1}{S}\sum_{n=1}^{S}\sum_{m=1}^{S}\sum_{k=0}^{K-1}\left(\nabla\mathcal{L}_{\pi_m^n}(\boldsymbol{w}) - \nabla\mathcal{L}_{\pi_m^n}(\boldsymbol{w}^*)\right)\right\|^2\right] + 4\mathbb{E}\left[\left\|\frac{1}{S}\sum_{n=1}^{S}\sum_{m=1}^{S}\sum_{k=0}^{K-1}\nabla\mathcal{L}_{\pi_m^n}(\boldsymbol{w}^*)\right\|^2\right]$$

$$(48)$$

Seeing the data sample $\xi_{m,k,n}$, the stochastic gradient $\mathbf{g}_{\pi_m^n,k}$, the gradient $\nabla\mathcal{L}_{\pi_m^n}(\xi_{m,k,n})$ as $\xi_i, \boldsymbol{w}_i$ and $\mathbf{e}_i$ in Lemma.6 respectively and applying the result of Lemma6, the first term on the right-hand side in (48) can be bounded by $4K\sigma^2$:

$$\mathbb{E}\left[\left\|\frac{1}{S}\sum_{n=1}^{S}\sum_{m=1}^{S}\sum_{k=0}^{K-1}\left(\mathbf{g}_{\pi_m^n,k} - \nabla\mathcal{L}_{\pi_m^n}(\boldsymbol{w}_{m,k,n})\right)\right\|^2\right]$$

$$\overset{(23)}{\leq} 4\frac{1}{S^2}\sum_{n=1}^{S}\sum_{m=1}^{S}\sum_{k=0}^{K-1}\|\mathbb{E}\left(\mathbf{g}_{\pi_m^n,k} - \nabla\mathcal{L}_{\pi_m^n}(\boldsymbol{w}_{m,k,n})\right)\|^2$$

$$\leq 4K\sigma^2$$

$$(49)$$

For the second term on the right-hand side in (48), we have

$$4\mathbb{E}\left[\left\|\frac{1}{S}\sum_{n=1}^{S}\sum_{m=1}^{S}\sum_{k=0}^{K-1}\left(\nabla\mathcal{L}_{\pi_m^n}(\boldsymbol{w}_{m,k,n})-\nabla\mathcal{L}_{\pi_m^n}(\boldsymbol{w})\right)\right\|^2\right]$$

$$\overset{(23)}{\leq}4S^2K\frac{1}{S^2}\sum_{n=1}^{S}\sum_{m=1}^{S}\sum_{k=0}^{K-1}\mathbb{E}\left[\left\|\nabla\mathcal{L}_{\pi_m^n}(\boldsymbol{w}_{m,k,n})-\nabla\mathcal{L}_{\pi_m^n}(\boldsymbol{w})\right\|^2\right] \tag{50}$$

$$\overset{\text{Asm.2}}{\leq}4L^2K\sum_{n=1}^{S}\sum_{m=1}^{S}\sum_{k=0}^{K-1}\mathbb{E}\left[\left\|\boldsymbol{w}_{m,k,n}-\boldsymbol{w}\right\|^2\right]$$

For the third term on the right-hand side in 48, we have

$$4\mathbb{E}\left[\left\|\frac{1}{S}\sum_{n=1}^{S}\sum_{m=1}^{S}\sum_{k=0}^{K-1}\left(\nabla\mathcal{L}_{\pi_m^n}(\boldsymbol{w})-\nabla\mathcal{L}_{\pi_m^n}(\boldsymbol{w}^*)\right)\right\|^2\right]$$

$$\overset{(23)}{\leq}4S^2K\frac{1}{S^2}\sum_{n=1}^{S}\sum_{m=1}^{S}\sum_{k=0}^{K-1}\mathbb{E}\left[\left\|\nabla\mathcal{L}_{\pi_m^n}(\boldsymbol{w})-\nabla\mathcal{L}_{\pi_m^n}(\boldsymbol{w}^*)\right\|^2\right]$$

$$\overset{(29)}{\leq}8LK\sum_{n=1}^{S}\sum_{m=1}^{S}\sum_{k=0}^{K-1}\mathbb{E}\left[D_{\mathcal{L}_{\pi_m^n}}(\boldsymbol{w},\boldsymbol{w}^*)\right] \tag{51}$$

$$\overset{(26)}{=}8LS^2K^2D_{\mathcal{L}}(\boldsymbol{w},\boldsymbol{w}^*),$$

We explain it as follows, because $D_{\mathcal{L}}(\boldsymbol{w},\boldsymbol{w}^*)$ is linear concerning $\mathcal{L}(\boldsymbol{w})$, so it satisfies the formula:

$$\mathbb{E}\left[D_{\mathcal{L}_{\pi_m}}(\boldsymbol{w},\boldsymbol{w}^*)\right]=D_{\mathcal{L}}(\boldsymbol{w},\boldsymbol{w}^*)$$

The fourth term on the right hand side in 48 can be bounded by Lemma.6 as follows:

$$4\mathbb{E}\left[\left\|\frac{1}{S}\sum_{n=1}^{S}\sum_{m=1}^{S}\sum_{k=0}^{K-1}\nabla\mathcal{L}_{\pi_m^n}(\boldsymbol{w}^*)\right\|^2\right]\overset{(Lem.6)}{\leq}4S^2K^2\frac{M-S}{S(M-1)}\zeta_*^2 \tag{52}$$

With the preceding four inequalities, we can bound the third term on the right hand side in (48):

$$\mathbb{E}\left[\left\|\frac{1}{S}\sum_{n=1}^{S}\sum_{m=1}^{S}\sum_{k=0}^{K-1}\mathbf{g}_{\pi_m^n,k}\right\|^2\right]$$

$$\leq 4K\sigma^2+4L^2K\sum_{n=1}^{S}\sum_{m=1}^{S}\sum_{k=0}^{K-1}\mathbb{E}\left[\left\|\boldsymbol{w}_{m,k,n}-\boldsymbol{w}\right\|^2\right] \tag{53}$$

$$+8LS^2K^2D_{\mathcal{L}}(\boldsymbol{w},\boldsymbol{w}^*)+4S^2K^2\frac{M-S}{S(M-1)}\zeta_*^2$$

Then substituting (47) and (53) into (46), we have

$$\mathbb{E}\left[\left\|\boldsymbol{w}+\Delta\boldsymbol{w}-\boldsymbol{w}^*\right\|^2\right]\leq\left(1-\frac{\mu SK\eta}{2}\right)\left\|\boldsymbol{w}-\boldsymbol{w}^*\right\|^2+4K\eta^2\sigma^2+4S^2K^2\eta^2\frac{M-S}{S(M-1)}\zeta_*^2$$

$$+2\frac{L\eta}{S}(1+2LSK\eta)\sum_{n=1}^{S}\sum_{m=1}^{S}\sum_{k=0}^{K-1}\mathbb{E}\left[\left\|\boldsymbol{w}_{m,k,n}-\boldsymbol{w}\right\|^2\right]$$

$$-2SK\eta(1-4LSK\eta)D_{\mathcal{L}}(\boldsymbol{w},\boldsymbol{w}^*)$$

Here we substitute $\eta\leq\frac{1}{6LSK}$ to get Lemma.10, as follows:

$$\mathbb{E}\left[\left\|\boldsymbol{w}+\Delta\boldsymbol{w}-\boldsymbol{w}^*\right\|^2\right]\leq\left(1-\frac{\mu SK\eta}{2}\right)\left\|\boldsymbol{w}-\boldsymbol{w}^*\right\|^2+4K\eta^2\sigma^2+4S^2K^2\eta^2\frac{M-S}{S(M-1)}\zeta_*^2$$

$$-\frac{2}{3}SK\eta D_{\mathcal{L}}(\boldsymbol{w},\boldsymbol{w}^*)+\frac{8}{3}\frac{L\eta}{S}\sum_{n=1}^{S}\sum_{m=1}^{S}\sum_{k=0}^{K-1}\mathbb{E}\left[\left\|\boldsymbol{w}_{m,k,n}-\boldsymbol{w}\right\|^2\right]$$

### G.1.2 Bounding the client drift with Assumption. 5

Similar to the "client drift" in PFL [19] and SFL [31], we define the client drift in SPFL:

$$E_r := \frac{1}{S} \sum_{n=1}^{S} \sum_{m=1}^{S} \sum_{k=0}^{K-1} \mathbb{E}\left[\left\|\boldsymbol{w}_{m,k,n}^{(r)} - \boldsymbol{w}^{(r)}\right\|^2\right] \tag{54}$$

**Lemma 11.** *When Assumptions. 2, 3 and 5 hold, and assuming that all local objective functions are $\mu$-strongly convex, then if the learning rate satisfies $\eta \leq \frac{1}{6LSK}$, the client drift is bounded:*

$$E_r \leq \frac{9}{4}S^2K^2\eta^2\sigma^2 + \frac{9}{4}S^2K^3\eta^2\zeta^2 + 3LS^3K^3\eta^2\mathbb{E}\left[D_{\mathcal{L}}(\boldsymbol{w}^{(r)}, \boldsymbol{w}^*)\right] \tag{55}$$

*Proof.* According to our SPFL pseudo code, we can get the model updates of SPFL from $\boldsymbol{w}^{(r)}$ to $\boldsymbol{w}_{m,k,n}^{(r)}$ is

$$\boldsymbol{w}_{m,k,n}^{(r)} - \boldsymbol{w}^{(r)} = -\eta \sum_{i=1}^{m} \sum_{j=0}^{b_{m,k}(i)} \mathbf{g}_{\pi_i^n,j}^{(r)}$$

with $b_{m,k}(i) := \begin{cases} K-1, & i \leq m-1 \\ k-1, & i = m \end{cases}$. In the following, we focus on a single training round, and hence we drop the superscript $r$ for a while, e.g., writing $\boldsymbol{w}_{m,k,n}$ to replace $\boldsymbol{w}_{m,k,n}^{(r)}$. In particular, we would like to use $\boldsymbol{w}$ to replace $\boldsymbol{w}_{1,0,1}$. Without otherwise stated, the expectation is conditioned on $\boldsymbol{w}^r$. We use Jensen's inequality to bound the term $\mathbb{E}\left[\|\boldsymbol{w}_{m,k,n} - \boldsymbol{w}\|^2\right] = \eta^2\mathbb{E}\left[\left\|\sum_{i=1}^{m}\sum_{j=0}^{b_{m,k}(i)}\mathbf{g}_{\pi_i^n,j}\right\|^2\right]$:

$$\mathbb{E}\left[\|\boldsymbol{w}_{m,k,n} - \boldsymbol{w}\|^2\right]$$

$$\leq 4\eta^2\mathbb{E}\left[\left\|\sum_{i=1}^{m}\sum_{j=0}^{b_{m,k}(i)}\left(\mathbf{g}_{\pi_i^n,j} - \nabla\mathcal{L}_{\pi_i^n}(\boldsymbol{w}_{i,j,n})\right)\right\|^2\right]$$

$$+ 4\eta^2\mathbb{E}\left[\left\|\sum_{i=1}^{m}\sum_{j=0}^{b_{m,k}(i)}\left(\nabla\mathcal{L}_{\pi_i^n}(\boldsymbol{w}_{i,j,n}) - \nabla\mathcal{L}_{\pi_i^n}(\boldsymbol{w})\right)\right\|^2\right]$$

$$+ 4\eta^2\mathbb{E}\left[\left\|\sum_{i=1}^{m}\sum_{j=0}^{b_{m,k}(i)}\left(\nabla\mathcal{L}_{\pi_i^n}(\boldsymbol{w}) - \nabla\mathcal{L}_{\pi_i^n}(\boldsymbol{w}^*)\right)\right\|^2\right] + 4\eta^2\mathbb{E}\left[\left\|\sum_{i=1}^{m}\sum_{j=0}^{b_{m,k}(i)}\nabla\mathcal{L}_{\pi_i^n}(\boldsymbol{w}^*)\right\|^2\right]$$

Applying Lemma.8 to the first term and Jensen's inequality to the second, third terms on the right hand side in the preceding inequality, respectively, we can get

$$\mathbb{E}\left[\|\boldsymbol{w}_{m,k,n} - \boldsymbol{w}\|^2\right]$$

$$\leq 4\eta^2 \sum_{i=1}^{m}\sum_{j=0}^{b_{m,k}(i)} \mathbb{E}\left[\left\|\mathbf{g}_{\pi_i^n,j} - \nabla\mathcal{L}_{\pi_i^n}(\boldsymbol{w}_{i,j,n})\right\|^2\right]$$

$$+ 4\eta^2\mathcal{C}_{m,k} \sum_{i=1}^{m}\sum_{j=0}^{b_{m,k}(i)} \mathbb{E}\left[\left\|\nabla\mathcal{L}_{\pi_i^n}(\boldsymbol{w}_{i,j,n}) - \nabla\mathcal{L}_{\pi_i^n}(\boldsymbol{w})\right\|^2\right]$$

$$+ 4\eta^2\mathcal{C}_{m,k} \sum_{i=1}^{m}\sum_{j=0}^{b_{m,k}(i)} \mathbb{E}\left[\|\nabla\mathcal{L}_{\pi_i^n}(\boldsymbol{w}) - \nabla\mathcal{L}_{\pi_i^n}(\boldsymbol{w}^*)\|^2\right] + 4\eta^2\mathbb{E}\left[\left\|\sum_{i=1}^{m}\sum_{j=0}^{b_{m,k}(i)}\nabla\mathcal{L}_{\pi_i^n}(\boldsymbol{w}^*)\right\|^2\right] \tag{56}$$

where $\mathcal{C}_{m,k} := \sum_{i=1}^{m} \sum_{i=0}^{b_{m,k}(i)} 1 = (m-1)K + k$. The first term on the right-hand side in (56) is bounded by $4\mathcal{C}_{m,k}\eta^2\sigma^2$. For the second term on the right-hand side in (56), we have

$$\mathbb{E}\left[\|\nabla\mathcal{L}_{\pi_i^n}(\boldsymbol{w}_{i,j,n}) - \nabla\mathcal{L}_{\pi_i^n}(\boldsymbol{w})\|^2\right] \overset{\text{Asm.2}}{\leq} L^2\mathbb{E}\left[\|\boldsymbol{w}_{i,j,n} - \boldsymbol{w}\|^2\right]$$

For the third term on the right-hand side in (56), we have

$$\mathbb{E}\left[\|\nabla\mathcal{L}_{\pi_i^n}(\boldsymbol{w}) - \nabla\mathcal{L}_{\pi_i^n}(\boldsymbol{w}^*)\|^2\right] \overset{(23)}{\leq} 2L\mathbb{E}\left[D_{\mathcal{L}_{\pi_i^n}}(\boldsymbol{w}, \boldsymbol{w}^*)\right] = 2LD_{\mathcal{L}}(\boldsymbol{w}, \boldsymbol{w}^*)$$

Since $\boldsymbol{w}^*$ is the optimal solution, its gradient $\nabla\mathcal{L}(\boldsymbol{w}^*) = 0$. As a result, we can get

$$\mathbb{E}\left[\|\boldsymbol{w}_{m,k,n} - \boldsymbol{w}\|^2\right] \leq 4\mathcal{C}_{m,k}\eta^2\sigma^2 + 4L^2\eta^2\mathcal{C}_{m,k}\sum_{i=1}^{m}\sum_{j=0}^{b(i)}\mathbb{E}\left[\|\boldsymbol{w}_{i,j,n} - \boldsymbol{w}\|^2\right] + 8L\eta^2\mathcal{C}_{m,k}^2 D_{\mathcal{L}}(\boldsymbol{w}, \boldsymbol{w}^*)$$

$$+ 4\eta^2\mathbb{E}\left[\left\|\sum_{i=1}^{m}\sum_{j=0}^{b_{m,k}(i)}\nabla\mathcal{L}_{\pi_i^n}(\boldsymbol{w}^*)\right\|^2\right]$$

(57)

Then, returning to $E_r := \frac{1}{S}\sum_{n=1}^{S}\sum_{m=1}^{S}\sum_{k=0}^{K-1}\mathbb{E}\left[\|\boldsymbol{w}_{m,k,n} - \boldsymbol{w}\|^2\right]$, we have

$$E_r \leq 4\eta^2\sigma^2\sum_{m=1}^{S}\sum_{k=0}^{K-1}\mathcal{C}_{m,k} + 4L^2\eta^2\sum_{m=1}^{S}\sum_{k=0}^{K-1}\mathcal{C}_{m,k}\sum_{i=1}^{m}\sum_{j=0}^{b_{m,k}(i)}\mathcal{E}\left[\|\boldsymbol{w}_{i,j,n} - \boldsymbol{w}\|^2\right]$$

$$+ 8L\eta^2\sum_{m=1}^{S}\sum_{k=0}^{K-1}\mathcal{C}_{m,k}^2 D_{\mathcal{L}}(\boldsymbol{w}, \boldsymbol{w}^*) + 4\eta^2\sum_{m=1}^{S}\sum_{k=0}^{K-1}\mathbb{E}\left[\left\|\sum_{i=1}^{m}\sum_{j=0}^{b_{m,k}(i)}\nabla\mathcal{L}_{\pi_i^n}(\boldsymbol{w}^*)\right\|^2\right]$$

Applying Lemma.7 with $\boldsymbol{w}_{\pi_i^n} = \nabla\mathcal{L}_{\pi_i^n}(\boldsymbol{w}^*)$ and $\overline{w} = \nabla\mathcal{L}(\boldsymbol{w}^*) = 0$ and Lemma.1 that

$$\frac{1}{S}\sum_{n=1}^{S}\sum_{m=1}^{S}\sum_{k=0}^{K-1}\mathcal{C}_{m,k} = \frac{1}{2}SK(SK-1) \leq \frac{1}{2}S^2K^2,$$

$$\frac{1}{S}\sum_{n=1}^{S}\sum_{m=1}^{S}\sum_{k=0}^{K-1}\mathcal{C}_{m,k}^2 = \frac{1}{3}(SK-1)SK(SK-\frac{1}{2}) \leq \frac{1}{3}S^3K^3$$

we can simplify the preceding inequality:

$$E_r \leq 2S^2K^2\eta^2\sigma^2 + 2L^2S^2K^2\eta^2 E_r + \frac{8}{3}LS^3K^3\eta^2 D_{\mathcal{L}}(\boldsymbol{w}, \boldsymbol{w}^*) + 2S^2K^3\eta^2\zeta_*^2$$

After rearranging the preceding inequality, we get

$$(1 - 2L^2S^2K^2\eta^2)E_r \leq 2S^2K^2\eta^2\sigma^2 + 2S^2K^3\eta^2\zeta_*^2 + \frac{8}{3}LS^3K^3\eta^2 D_{\mathcal{L}}(\boldsymbol{w}, \boldsymbol{w}^*)$$

Finally, using the condition that $\eta \leq \frac{1}{6LSK}$, which implies $1 - 2L^2S^2K^2\eta^2 \geq \frac{8}{9}$, we have

$$E_r \leq \frac{9}{4}S^2K^2\eta^2\sigma^2 + \frac{9}{4}S^2K^3\eta^2\zeta_*^2 + 3LS^3K^3\eta^2 D_{\mathcal{L}}(\boldsymbol{w}, \boldsymbol{w}^*)$$

The claim follows after recovering the superscripts and taking unconditional expectations.

### G.1.3 Proof of strongly convex case of Theorem 1

Proof of the strongly convex case of Theorem 1. Substituting Lemma .11 into Lemma .10 and using $\eta \leq \frac{1}{6LSK}$, we can simplify the recursion as follows:

$$
\begin{aligned}
\mathbb{E}\left[\left\|\boldsymbol{w}^{(r+1)} - \boldsymbol{w}^*\right\|^2\right] &\leq \left(1 - \frac{\mu SK\eta}{2}\right)\|\boldsymbol{w} - \boldsymbol{w}^*\|^2 + 4K\eta^2\sigma^2 + 4S^2K^2\eta^2\frac{M-S}{S(M-1)}\zeta_*^2 \\
&\quad - \frac{2}{3}SK\eta D_{\mathcal{L}}(\boldsymbol{w}, \boldsymbol{w}^*) + \frac{8}{3}L\eta E_r \\
&\leq \left(1 - \frac{\mu SK\eta}{2}\right)\mathbb{E}\left[\left\|\boldsymbol{w}^{(r)} - \boldsymbol{w}^*\right\|^2\right] - \frac{1}{3}SK\eta\mathbb{E}\left[D_{\mathcal{L}}\left(\boldsymbol{w}^{(r)}, \boldsymbol{w}^*\right)\right] \\
&\quad + 4K\eta^2\sigma^2 + 4S^2K^2\eta^2\frac{M-S}{S(M-1)}\zeta_*^2 + 6LS^2K^2\eta^3\sigma^2 + 6LS^2K^3\eta^3\zeta_*^2
\end{aligned}
$$

Let $\tilde{\eta} = MK\eta$, we have

$$
\begin{aligned}
\mathbb{E}\left[\left\|\boldsymbol{w}^{(r+1)} - \boldsymbol{w}^*\right\|^2\right] &\leq \left(1 - \frac{\mu\tilde{\eta}}{2}\right)\mathbb{E}\left[\left\|\boldsymbol{w}^{(r)} - \boldsymbol{w}^*\right\|^2\right] - \frac{\tilde{\eta}}{3}\mathbb{E}\left[D_{\mathcal{L}}(\boldsymbol{w}^{(r)}, \boldsymbol{w}^*)\right] \\
&\quad + \frac{4\tilde{\eta}^2\sigma^2}{S^2K} + \frac{4\tilde{\eta}^2(M-S)\zeta_*^2}{S(M-1)} + \frac{6L\tilde{\eta}^3\sigma^2}{SK} + \frac{6L\tilde{\eta}^3\zeta_*^2}{S}
\end{aligned} \tag{58}
$$

Applying Lemma 3 with $t = r(T = R), \gamma = \tilde{\eta}, r_t = \mathbb{E}\left[\left\|\boldsymbol{w}^{(r)} - \boldsymbol{w}^*\right\|^2\right], a = \frac{\mu}{2}, b = \frac{1}{3}, s_t = \mathbb{E}\left[D_{\mathcal{L}}(\boldsymbol{w}^{(r)}, \boldsymbol{w}^*)\right], \theta_t = (1 - \frac{\mu\tilde{\eta}}{2})^{-(r+1)}, c_1 = \frac{4\sigma^2}{S^2K} + \frac{4(M-S)\zeta_*^2}{S(M-1)}, c_2 = \frac{6L\sigma^2}{SK} + \frac{6L\zeta_*^2}{SK}$ and $\frac{1}{d} = \frac{1}{6L}(\tilde{\eta} = MK\eta \leq \frac{1}{6L})$, it follows that

$$
\begin{aligned}
\mathbb{E}\left[\mathcal{L}(\bar{\boldsymbol{w}}^{(R)}) - \mathcal{L}(\boldsymbol{w}^*)\right] &\leq \frac{1}{W_R}\sum_{r=0}^{R}\theta_r\mathbb{E}\left[\mathcal{L}(\boldsymbol{w}^{(r)}) - \mathcal{L}(\boldsymbol{w}^*)\right] \\
&\leq \frac{9}{2}\mu\left\|\boldsymbol{w}^{(0)} - \boldsymbol{w}^*\right\|^2\exp\left(-\frac{1}{2}\mu\tilde{\eta}R\right) + \frac{12\tilde{\eta}\sigma^2}{S^2K} + \frac{12\tilde{\eta}(M-S)\zeta_*^2}{S(M-1)} + \frac{18L\tilde{\eta}^2\sigma^2}{SK} + \frac{18L\tilde{\eta}^2\zeta_*^2}{SK}
\end{aligned} \tag{59}
$$

where $\bar{\boldsymbol{w}}^{(R)} = \frac{1}{W_R}\sum_{r=0}^{R}\theta_r\boldsymbol{w}^{(r)}$ and we use Jensen's inequality ($\mathcal{L}$ is convex) in the first inequality. Applying Lemma .3 to Eq.59 and using a suitable dynamic learning rate yields:

$$
\begin{aligned}
&\mathbb{E}\left[\mathcal{L}(\bar{\boldsymbol{w}}^{(R)}) - \mathcal{L}(\boldsymbol{w}^*)\right] \\
&= \tilde{\mathcal{O}}\left(\mu\mathcal{A}^2\exp\left(-\frac{\mu R}{12L}\right) + \frac{\sigma^2}{\mu S^2KR} + \frac{(M-S)\zeta_*^2}{\mu SR(M-1)} + \frac{L\sigma^2}{\mu^2SKR^2} + \frac{L\zeta_*^2}{\mu^2SR^2}\right)
\end{aligned} \tag{60}
$$

where $\mathcal{A} := \left\|\boldsymbol{w}^{(0)} - \boldsymbol{w}^*\right\|$. Eq. (59) and Eq . 60 are the upper bounds with partial client participation. In particular, when S = M, we can get the claim of the strongly convex case of Theorem 1.

## G.2 General Convex Case

### G.2.1 Proof of general convex case of Theorem 1

Proof of the general convex case of Theorem 1. Letting $\mu = 0$ in Eq. (58), we get the recursion of the general convex case,

$$
\begin{aligned}
\mathbb{E}\left[\left\|\boldsymbol{w}^{(r+1)} - \boldsymbol{w}^*\right\|^2\right] &\leq \mathbb{E}\left[\left\|\boldsymbol{w}^{(r)} - \boldsymbol{w}^*\right\|^2\right] - \frac{\tilde{\eta}}{3}\mathbb{E}\left[D_{\mathcal{L}}(\boldsymbol{w}^{(r)}, \boldsymbol{w}^*)\right] \\
&\quad + \frac{4\tilde{\eta}^2\sigma^2}{S^2K} + \frac{4\tilde{\eta}^2(M-S)\zeta_*^2}{S(M-1)} + \frac{6L\tilde{\eta}^3\sigma^2}{SK} + \frac{6L\tilde{\eta}^3\zeta_*^2}{S}
\end{aligned}
$$

Applying Lemma.5 with $t = r(T = R), \gamma = \tilde{\eta}, r_t = \mathbb{E}\left[\left\|\boldsymbol{w}^{(r)} - \boldsymbol{w}^*\right\|^2\right], a = 0, b = \frac{1}{3}, s_t = \mathbb{E}\left[D_{\mathcal{L}}(\boldsymbol{w}^{(r)}, \boldsymbol{w}^*)\right], \theta_t = (1 - \frac{\mu\tilde{\eta}}{2})^{-(r+1)}, c_1 = \frac{4\sigma^2}{S^2K} + \frac{4(M-S)\zeta_*^2}{S(M-1)}, c_2 = \frac{6L\sigma^2}{SK} + \frac{6L\zeta_*^2}{SK}$ and $\frac{1}{d} =$

$\frac{1}{6L}(\tilde{\eta} = MK\eta \le \frac{1}{6L})$ ,it follows that

$$\mathbb{E}\left[\mathcal{L}(\bar{\boldsymbol{w}}^{(R)}) - \mathcal{L}(\boldsymbol{w}^*)\right] \le \frac{1}{W_R} \sum_{r=0}^{R} \theta_r \left(\mathcal{L}(\boldsymbol{w}^{(r)}) - \mathcal{L}(\boldsymbol{w}^*)\right)$$

$$\le \frac{3\left\|\boldsymbol{w}^{(0)} - \boldsymbol{w}^*\right\|^2}{\tilde{\eta}R} + \frac{12\tilde{\eta}\sigma^2}{S^2 K} + \frac{12\tilde{\eta}(M-S)\zeta_*^2}{S(M-1)} + \frac{18L\tilde{\eta}^2\sigma^2}{SK} + \frac{18L\tilde{\eta}^2\zeta_*^2}{SK}$$

(61)

where $\bar{\boldsymbol{w}}^{(R)} = \frac{1}{W_R}\sum_{r=0}^{R}\theta_r \boldsymbol{w}^{(r)}$ and we use Jensen's inequality ($\mathcal{L}$ is convex) in the first inequality. By using a suitable dynamic learning rate, we get

$$\mathcal{L}(\bar{\boldsymbol{w}}^{(R)}) - \mathcal{L}(\boldsymbol{w}^*) \le 2\left(\frac{4\sigma^2}{S^2 K} + \frac{4(M-S)\zeta_*^2}{S(M-1)}\right)^{\frac{1}{2}}\left(\frac{\mathbb{E}\left[\left\|\boldsymbol{w}^{(0)} - \boldsymbol{w}^*\right\|^2\right]}{R+1}\right)^{\frac{1}{2}}$$

$$+ 2\left(\frac{6L\sigma^2}{SK} + \frac{6L\zeta_*^2}{SK}\right)^{\frac{1}{3}}\left(\frac{\mathbb{E}\left[\left\|\boldsymbol{w}^{(0)} - \boldsymbol{w}^*\right\|^2\right]}{R+1}\right)^{\frac{2}{3}} + \frac{6L\mathbb{E}\left[\left\|\boldsymbol{w}^{(0)} - \boldsymbol{w}^*\right\|^2\right]}{R+1}$$

Due to the concave nature of the power function(Lemma.2), we can get

$$\mathcal{L}(\bar{\boldsymbol{w}}^{(R)}) - \mathcal{L}(\boldsymbol{w}^*) \le 2\left[\left(\frac{4\sigma^2}{S^2 K}\right) + \left(\frac{4(M-S)\zeta_*^2}{S(M-1)}\right)\right]^{\frac{1}{2}}\left(\frac{\mathbb{E}\left[\left\|\boldsymbol{w}^{(0)} - \boldsymbol{w}^*\right\|^2\right]}{R+1}\right)^{\frac{1}{2}}$$

$$+ 2\left[\left(\frac{6L\sigma^2}{SK}\right) + \left(\frac{6L\zeta_*^2}{SK}\right)\right]^{\frac{1}{3}}\left(\frac{\mathbb{E}\left[\left\|\boldsymbol{w}^{(0)} - \boldsymbol{w}^*\right\|^2\right]}{R+1}\right)^{\frac{2}{3}} + \frac{6L\mathbb{E}\left[\left\|\boldsymbol{w}^{(0)} - \boldsymbol{w}^*\right\|^2\right]}{R+1}$$

After finishing, we can get:

$$\mathcal{L}(\bar{\boldsymbol{w}}^{(R)}) - \mathcal{L}(\boldsymbol{w}^*) = \mathcal{O}\left(\frac{\sigma\mathcal{A}}{\sqrt{S^2 KR}} + \sqrt{1 - \frac{S}{M}}\cdot\frac{\zeta_*\mathcal{A}}{\sqrt{SR}} + \frac{(L\sigma^2\mathcal{A}^4)^{1/3}}{(SK)^{1/3}R^{2/3}} + \frac{(L\zeta_*^2\mathcal{A}^4)^{1/3}}{S^{1/3}R^{2/3}} + \frac{L\mathcal{A}^2}{R}\right)$$

(62)

where $\mathcal{A} := \left\|\boldsymbol{w}^{(0)} - \boldsymbol{w}^*\right\|$. Eq. (61) and Eq. (62) are the upper bounds with partial client participation. In particular, when $S = M$, we can claim the strongly convex case of Theorem 1 and Corollary 1.

### G.3 Nonconvex Case

**Lemma 12.** *Let Assumptions 2,3 and 4 hold. If the learning rate satisfies $\eta = \frac{1}{6LSK}$, then it holds that*

$$\mathbb{E}\left[\mathcal{L}(\boldsymbol{w}^{(r+1)}) - \mathcal{L}(\boldsymbol{w}^{(r)})\right] \le -\frac{SK\eta}{2}\mathbb{E}\left[\left\|\nabla\mathcal{L}(\boldsymbol{w}^{(r)})\right\|^2\right] + LSK\eta^2\sigma^2$$

$$+ \frac{L^2\eta}{2}\sum_{m=1}^{S}\sum_{k=0}^{K-1}\mathbb{E}\left[\left\|\boldsymbol{w}_{m,k}^{(r)} - \boldsymbol{w}^{(r)}\right\|^2\right]$$

(63)

*Proof.* According to the Pseudocode of SPFL, the overall model updates of SPFL after one complete training round (with S clients selected for training) is

$$\Delta\boldsymbol{w} = \boldsymbol{w}^{(r+1)} - \boldsymbol{w}^{(r)} = -\eta\frac{1}{S}\sum_{n=1}^{S}\sum_{m=1}^{S}\sum_{k=0}^{K-1}\mathbf{g}_{\pi_m^n, k}^{(r)}$$

where $\mathbf{g}_{\pi_m^n, k}^{(r)} = \nabla f_{\pi_m^n}(\boldsymbol{w}_{m,k,n}^{(r)}; \xi)$ is the stochastic gradient of $\mathcal{L}_{\pi_m^n}$ regarding the vector $\boldsymbol{w}_{m,k,n}^{(r)}$. Thus,

$$\mathbb{E}\left[\Delta\boldsymbol{w}\right] = -\eta\frac{1}{S}\sum_{n=1}^{S}\sum_{m=1}^{S}\sum_{k=0}^{K-1}\mathbb{E}\left[\nabla\mathcal{L}_{\pi_m^n}(\boldsymbol{w}_{m,k,n})\right]$$

In the following, we focus on the recurrence of adjacent training rounds, so we omit the superscript $r$ for a while, e.g., writing $\boldsymbol{w}_{m,k,n}^r$ as $\boldsymbol{w}_{m,k,n}$. In particular, we would like to use $\boldsymbol{w}$ to replace $\boldsymbol{w}_{1,0,1}$. Without otherwise stated, the expectation is conditioned on $\boldsymbol{w}^r$.

Starting from the smoothness of $F$ (applying Eq. (53)), $D_{\mathcal{L}}(\boldsymbol{x}, \boldsymbol{y}) \leq \frac{L}{2} \|\boldsymbol{x} - \boldsymbol{y}\|^2$ with $x = \boldsymbol{w} + \Delta \boldsymbol{w}$, $y = \boldsymbol{w}$, and substituting the overall updates, we have

$$
\begin{aligned}
& \mathbb{E}\left[\mathcal{L}(\boldsymbol{w} + \Delta \boldsymbol{w}) - \mathcal{L}(\boldsymbol{w})\right] \\
& \leq \mathbb{E}\left[\langle \nabla \mathcal{L}(\boldsymbol{w}), \Delta \boldsymbol{w} \rangle\right] + \frac{L}{2} \mathbb{E}\left[\|\Delta \boldsymbol{w}\|^2\right] \\
& \leq -\eta \frac{1}{S} \sum_{n=1}^{S} \sum_{m=1}^{S} \sum_{k=0}^{K-1} \mathbb{E}\left[\langle \nabla \mathcal{L}(\boldsymbol{w}), \nabla \mathcal{L}_{\pi_m^n}(\boldsymbol{w}_{m,k,n}) \rangle\right] + \frac{L\eta^2}{2} \mathbb{E}\left[\left\|\frac{1}{S} \sum_{n=1}^{S} \sum_{m=1}^{S} \sum_{k=0}^{K-1} \mathbf{g}_{\pi_m^n, k}\right\|^2\right]
\end{aligned}
$$
(64)

For the first term on the right-hand side in Eq.(64), using the fact that $2\langle a, b \rangle = \|a\|^2 + \|b\|^2 - \|a - b\|^2$ with $a = \nabla \mathcal{L}(\boldsymbol{w})$ and $b = \nabla \mathcal{L}_{\pi_m^n}(\boldsymbol{w}_{m,k,n})$, we have

$$
\begin{aligned}
& -\eta \frac{1}{S} \sum_{n=1}^{S} \sum_{m=1}^{S} \sum_{k=0}^{K-1} \mathbb{E}\left[\langle \nabla \mathcal{L}(\boldsymbol{w}), \nabla \mathcal{L}_{\pi_m^n}(\boldsymbol{w}_{m,k,n}) \rangle\right] \\
& = -\frac{\eta}{2} \frac{1}{S} \sum_{n=1}^{S} \sum_{m=1}^{S} \sum_{k=0}^{K-1} \mathbb{E}\left[\|\nabla \mathcal{L}(\boldsymbol{w})\|^2 + \left\|\nabla \mathcal{L}_{\pi_m^n}(\boldsymbol{w}_{m,k,n})\right\|^2 - \left\|\nabla \mathcal{L}_{\pi_m^n}(\boldsymbol{w}_{m,k,n}) - \nabla \mathcal{L}(\boldsymbol{w})\right\|^2\right] \\
& \overset{\text{Asm.2}}{\leq} -\frac{SK\eta}{2} \|\nabla \mathcal{L}(\boldsymbol{w})\|^2 - \frac{\eta}{2} \frac{1}{S} \sum_{n=1}^{S} \sum_{m=1}^{S} \sum_{k=0}^{K-1} \mathbb{E}\left[\left\|\nabla \mathcal{L}_{\pi_m^n}(\boldsymbol{w}_{m,k,n})\right\|^2\right] \\
& \quad + \frac{L^2\eta}{2} \frac{1}{S} \sum_{n=1}^{S} \sum_{m=1}^{S} \sum_{k=0}^{K-1} \mathbb{E}\left[\|\boldsymbol{w}_{m,k,n} - \boldsymbol{w}\|^2\right]
\end{aligned}
$$
(65)

For the third term on the right hand side in Eq. (64), using Jensen's inequality, we have

$$
\begin{aligned}
& \frac{L\eta^2}{2} \mathbb{E}\left[\left\|\frac{1}{S} \sum_{n=1}^{S} \sum_{m=1}^{S} \sum_{k=0}^{K-1} \mathbf{g}_{\pi_m, k}\right\|^2\right] \\
& \leq L\eta^2 \mathbb{E}\left[\left\|\frac{1}{S} \sum_{n=1}^{S} \sum_{m=1}^{S} \sum_{k=0}^{K-1} \mathbf{g}_{\pi_m^n, k} - \frac{1}{S} \sum_{n=1}^{S} \sum_{m=1}^{S} \sum_{k=0}^{K-1} \nabla \mathcal{L}_{\pi_m^n}(\boldsymbol{w}_{m,k,n})\right\|^2\right] \\
& \quad + L\eta^2 \mathbb{E}\left[\left\|\frac{1}{S} \sum_{n=1}^{S} \sum_{m=1}^{S} \sum_{k=0}^{K-1} \nabla \mathcal{L}_{\pi_m^n}(\boldsymbol{w}_{m,k,n})\right\|^2\right] \\
& \leq LK\eta^2 \sigma^2 + LSK\eta^2 \frac{1}{S} \sum_{n=1}^{S} \sum_{m=1}^{S} \sum_{k=0}^{K-1} \mathbb{E}\left[\|\nabla \mathcal{L}_{\pi_m}(\boldsymbol{w}_{m,k})\|^2\right],
\end{aligned}
$$
(66)

where we apply Lemma .6 by seeing the data sample $\xi_{m,k,n}$, the stochastic gradient $\mathbf{g}_{\pi_m^n, k}$, the gradient $\nabla \mathcal{L}_{\pi_m^n}(\xi_{m,k,n})$ as $\xi_i, \boldsymbol{w}_i, \mathbf{e}_i$ respectively, in Lemma 6 for the first term and Jensen's inequality for the second term in the preceding inequality. Substituting Eq. (65) and Eq. (66) into Eq. (64), we have

$$
\begin{aligned}
\mathbb{E}\left[\mathcal{L}(\boldsymbol{w} + \Delta \boldsymbol{w}) - \mathcal{L}(\boldsymbol{w})\right] \leq & -\frac{SK\eta}{2} \|\nabla \mathcal{L}(\boldsymbol{w})\|^2 + LK\eta^2 \sigma^2 + \frac{L^2\eta}{2} \frac{1}{S} \sum_{n=1}^{S} \sum_{m=1}^{S} \sum_{k=0}^{K-1} \mathbb{E}\left[\|\boldsymbol{w}_{m,k,n} - \boldsymbol{w}\|^2\right] \\
& -\frac{\eta}{2}(1 - 2LSK\eta) \frac{1}{S} \sum_{n=1}^{S} \sum_{m=1}^{S} \sum_{k=0}^{K-1} \mathbb{E}\left[\left\|\nabla \mathcal{L}_{\pi_m^n}(\boldsymbol{w}_{m,k,n})\right\|^2\right]
\end{aligned}
$$

Since $\eta \leq \frac{1}{6LSK}$, the last term on the right-hand side in the preceding inequality is negative. Then

$$\mathbb{E}\left[\mathcal{L}(\boldsymbol{w} + \Delta\boldsymbol{w}) - \mathcal{L}(\boldsymbol{w})\right] \leq -\frac{SK\eta}{2}\left\|\nabla\mathcal{L}(\boldsymbol{w})\right\|^2 + LK\eta^2\sigma^2 + \frac{L^2\eta}{2}\frac{1}{S}\sum_{n=1}^{S}\sum_{m=1}^{S}\sum_{k=0}^{K-1}\mathbb{E}\left[\left\|\boldsymbol{w}_{m,k,n} - \boldsymbol{w}\right\|^2\right]$$

This conclusion can be obtained after restoring the superscript and taking the unconditional expectation.

### G.3.1 Bounding the client drift with Assumption 4

Since the proof of Lemma. 11 uses Eq. (28), which is only applicable to convex functions; we cannot use the result of Lemma. 11. Next, we use Assumption 4 to bound the client drift (defined in Eq. (54)).

**Lemma 13.** *Assumptions 2, 3, and 4 hold. If the learning rate satisfies $\eta \leq \frac{1}{6LSK}$, the client drift is bounded*

$$E_r \leq \frac{9}{4}S^2K^2\eta^2\sigma^2 + \frac{9}{4}S^2K^3\eta^2\zeta^2 + \left(\frac{9}{4}\beta^2 S^2 K^3 \eta^2 + \frac{3}{2}S^3 K^3 \eta^2\right)\mathbb{E}\left[\left\|\nabla\mathcal{L}(\boldsymbol{w}^{(r)})\right\|^2\right] \quad (67)$$

*Proof.* Similar to the "client drift" in PFL [19] and SFL [31], we define the client drift in SPFL:

$$E_r := \frac{1}{S}\sum_{n=1}^{S}\sum_{m=1}^{S}\sum_{k=0}^{K-1}\mathbb{E}\left[\left\|\boldsymbol{w}_{m,k,n}^{(r)} - \boldsymbol{w}^{(r)}\right\|^2\right] \quad (68)$$

with $b_{m,k}(i) := \begin{cases} K-1, & i \leq m-1 \\ k-1, & i = m \end{cases}$ . In the following, we focus on a single training round, and hence we drop the superscript $r$ for a while, e.g., writing $\boldsymbol{w}_{m,k,n}$ to replace $\boldsymbol{w}_{m,k,n}^{(r)}$. In particular, we would like to use $\boldsymbol{w}$ to replace $\boldsymbol{w}_{1,0,1}$. Without otherwise stated, the expectation is conditioned on $\boldsymbol{w}^r$. We use Jensen's inequality to bound the term $\mathbb{E}\left[\left\|\boldsymbol{w}_{m,k,n} - \boldsymbol{w}\right\|^2\right] = \eta^2\mathbb{E}\left[\left\|\sum_{i=1}^{m}\sum_{j=0}^{b_{m,k}(i)} \mathbf{g}_{\pi_i^n,j}\right\|^2\right]$:

$$\mathbb{E}\left[\left\|\boldsymbol{w}_{m,k,n} - \boldsymbol{w}\right\|^2\right]$$

$$\leq 4\eta^2\mathbb{E}\left[\left\|\sum_{i=1}^{m}\sum_{j=0}^{b_{m,k}(i)}\left(\mathbf{g}_{\pi_i^n,j} - \nabla\mathcal{L}_{\pi_i^n}(\boldsymbol{w}_{i,j,n})\right)\right\|^2\right]$$

$$+ 4\eta^2\mathbb{E}\left[\left\|\sum_{i=1}^{m}\sum_{j=0}^{b_{m,k}(i)}\left(\nabla\mathcal{L}_{\pi_i^n}(\boldsymbol{w}_{i,j,n}) - \nabla\mathcal{L}_{\pi_i^n}(\boldsymbol{w})\right)\right\|^2\right]$$

$$+ \underbrace{4\eta^2\mathbb{E}\left[\left\|\sum_{i=1}^{m}\sum_{j=0}^{b_{m,k}(i)}\left(\nabla\mathcal{L}_{\pi_i^n}(\boldsymbol{w}) - \nabla\mathcal{L}(\boldsymbol{w})\right)\right\|^2\right]}_{T_1} + 4\eta^2\mathbb{E}\left[\left\|\sum_{i=1}^{m}\sum_{j=0}^{b_{m,k}(i)}\nabla\mathcal{L}(\boldsymbol{w})\right\|^2\right]$$

Applying Lemma. 8, Jensen's inequality and Jensen's inequality to the first, third, and fourth terms on the right side of the previous inequality, respectively, we can get

$$\mathbb{E}\left[\left\|\boldsymbol{w}_{m,k,n} - \boldsymbol{w}\right\|^2\right]$$

$$\leq 4\eta^2\sum_{i=1}^{m}\sum_{j=0}^{b_{m,k}(i)}\mathbb{E}\left[\left\|\mathbf{g}_{\pi_i^n,j} - \nabla\mathcal{L}_{\pi_i^n}(\boldsymbol{w}_{i,j,n})\right\|^2\right] \quad (69)$$

$$+ 4\eta^2\mathcal{C}_{m,k}\sum_{i=1}^{m}\sum_{j=0}^{b_{m,k}(i)}\mathbb{E}\left[\left\|\nabla\mathcal{L}_{\pi_i^n}(\boldsymbol{w}_{i,j,n}) - \nabla\mathcal{L}_{\pi_i^n}(\boldsymbol{w})\right\|^2\right] + 4\eta^2 T_1 + 4\mathcal{C}_{m,k}^2\eta^2\left\|\nabla\mathcal{L}(\boldsymbol{w})\right\|^2$$

where $\mathcal{C}_{m,k} := \sum_{i=1}^{m} \sum_{i=0}^{b_{m,k}(i)} 1 = (m-1)K + k$. The first term on the right-hand side in (69) is bounded by $4\mathcal{C}_{m,k}\eta^2\sigma^2$ with Assumption 3. With Assumption 2, the second term on the right-hand side in (56) can be defined as

$$4\eta^2 \mathcal{C}_{m,k} \sum_{i=1}^{m} \sum_{j=0}^{b_{m,k}(i)} \mathbb{E}\left[\left\|\nabla\mathcal{L}_{\pi_i^n}(\boldsymbol{w}_{i,j,n}) - \nabla\mathcal{L}_{\pi_i^n}(\boldsymbol{w})\right\|^2\right] \leq 4L^2\eta^2 \mathcal{C}_{m,k} \sum_{i=1}^{m} \sum_{j=0}^{b_{m,k}(i)} \mathbb{E}\left[\|\boldsymbol{w}_{i,j,n} - \boldsymbol{w}\|^2\right]$$

Then, returning to $E_r := \frac{1}{S}\sum_{n=1}^{S}\sum_{m=1}^{S}\sum_{k=0}^{K-1}\mathbb{E}\left[\|\boldsymbol{w}_{m,k,n} - \boldsymbol{w}\|^2\right]$, we have

$$E_r \leq 4\eta^2\sigma^2 \frac{1}{S}\sum_{n=1}^{S}\sum_{m=1}^{S}\sum_{k=0}^{K-1}\mathcal{C}_{m,k} + 4L^2\eta^2\frac{1}{S}\sum_{n=1}^{S}\sum_{m=1}^{S}\sum_{k=0}^{K-1}\mathcal{C}_{m,k}\sum_{i=1}^{m}\sum_{j=0}^{b_{m,k}(i)}\mathbb{E}\left[\|\boldsymbol{w}_{i,j,n} - \boldsymbol{w}\|^2\right]$$

$$+ 4\eta^2\frac{1}{S}\sum_{n=1}^{S}\sum_{m=1}^{S}\sum_{k=0}^{K-1}\mathbb{E}\left[\left\|\sum_{i=1}^{m}\sum_{j=0}^{b_{m,k}(i)}\left(\nabla\mathcal{L}_{\pi_i^n}(\boldsymbol{w}) - \nabla\mathcal{L}(\boldsymbol{w})\right)\right\|^2\right]$$

$$+ 4\eta^2\frac{1}{S}\sum_{n=1}^{S}\sum_{m=1}^{S}\sum_{k=0}^{K-1}\mathcal{C}_{m,k}^2\left\|\nabla\mathcal{L}(\boldsymbol{w})\right\|^2$$

$$\tag{70}$$

Applying Lemma .7 with $\boldsymbol{w}_{\pi_i^n} = \nabla\mathcal{L}_{\pi_i^n}(\boldsymbol{w})$ and $\bar{x} = \nabla\mathcal{L}(\boldsymbol{w})$ to the third term and $\frac{1}{S}\sum_{n=1}^{M}\sum_{m=1}^{M}\sum_{k=0}^{K-1}\mathcal{C}_{m,k} \leq \frac{1}{2}S^2K^2$ and $\frac{1}{S}\sum_{n=1}^{S}\sum_{m=1}^{S}\sum_{k=0}^{K-1}\mathcal{C}_{m,k}^2 \leq \frac{1}{3}S^3K^3$ to the other terms on the right Hand side of the preceding inequality, we can simplify it:

$$E_r \leq 2S^2K^2\eta^2\sigma^2 + 2L^2S^2K^2\eta^2 E_r + 2S^2K^3\eta^2\left(\frac{1}{M}\sum_{i=1}^{M}\|\nabla\mathcal{L}_i(\boldsymbol{w}) - \nabla\mathcal{L}(\boldsymbol{w})\|^2\right)$$

$$+ \frac{4}{3}S^3K^3\eta^2\left\|\nabla\mathcal{L}(\boldsymbol{w})\right\|^2$$

$$\overset{Asm.2}{\leq} 2S^2K^2\eta^2\sigma^2 + 2L^2S^2K^2\eta^2 E_r + 2S^2K^3\eta^2\zeta^2 + 2\beta^2 S^2K^3\eta^2\left\|\nabla\mathcal{L}(\boldsymbol{w})\right\|^2$$

$$+ \frac{4}{3}S^3K^3\eta^2\left\|\nabla\mathcal{L}(\boldsymbol{w})\right\|^2$$

After rearranging the preceding inequality, we get

$$(1 - 2L^2S^2K^2\eta^2)E_r \leq 2S^2K^2\eta^2\sigma^2 + 2S^2K^3\eta^2\zeta^2 + 2\beta^2 S^2K^3\eta^2\left\|\nabla\mathcal{L}(\boldsymbol{w})\right\|^2 + \frac{4}{3}S^3K^3\eta^2\left\|\nabla\mathcal{L}(\boldsymbol{w})\right\|^2$$

Finally, using the condition that $\eta \leq \frac{1}{6LSK}$, which implies $1 - 2L^2S^2K^2\eta^2 \geq \frac{8}{9}$, we have

$$E_r \leq \frac{9}{4}S^2K^2\eta^2\sigma^2 + \frac{9}{4}S^2K^3\eta^2\zeta^2 + \frac{9}{4}\beta^2 S^2K^3\eta^2\left\|\nabla\mathcal{L}(\boldsymbol{w})\right\|^2 + \frac{3}{2}S^3K^3\eta^2\left\|\nabla\mathcal{L}(\boldsymbol{w})\right\|^2$$

The claim follows after recovering the superscripts and taking unconditional expectations.

### G.3.2  Proof of nonconvex case of Theorem 1

*Proof.*  Substituting Lemma.12 into Lemma.13 and using $\eta \leq \frac{1}{6LSK}min\{1, \frac{\sqrt{S}}{\beta}\}$ we can simplify the recursion as follows:

$$\mathbb{E}\left[\mathcal{L}(\boldsymbol{w}^{(r+1)}) - \mathcal{L}(\boldsymbol{w}^{(r)})\right] \leq -\frac{1}{3}SK\eta\mathbb{E}\left[\left\|\nabla\mathcal{L}(\boldsymbol{w}^{(r)})\right\|^2\right] + LK\eta^2\sigma^2 + \frac{9}{8}L^2S^2K^2\eta^3\sigma^2 + \frac{9}{8}L^2S^2K^3\eta^3\zeta^2$$

Letting $\tilde{\eta} := SK\eta$ Subtracting $\mathcal{L}^*$ from both sides and rearranging the terms, we have

$$\mathbb{E}\left[\mathcal{L}(\boldsymbol{w}^{(r+1)}) - \mathcal{L}^*\right] \leq \mathbb{E}\left[\mathcal{L}(\boldsymbol{w}^{(r)}) - \mathcal{L}^*\right] - \frac{\tilde{\eta}}{3}\mathbb{E}\left[\left\|\nabla\mathcal{L}(\boldsymbol{w}^{(r)})\right\|^2\right] + \frac{L\tilde{\eta}^2\sigma^2}{S^2K} + \frac{9L^2\tilde{\eta}^3\sigma^2}{8SK} + \frac{9L^2\tilde{\eta}^3\zeta^2}{8S}$$

Then applying Lemma .5 with $t = r(T = R), \gamma = \tilde{\eta}, r_t = \mathbb{E}\left[\mathcal{L}(\boldsymbol{w}^{(r)}) - \mathcal{L}^*\right], b = \frac{1}{3}$, $s_t = \mathbb{E}\left[\left\|\nabla\mathcal{L}(\boldsymbol{w}^{(r)})\right\|^2\right], \theta_t = 1, c_1 = \frac{L\sigma^2}{S^2 K}, c_2 = \frac{9L^2\sigma^2}{8SK} + \frac{9L^2\zeta^2}{8S}$ and $\frac{1}{d} = \frac{1}{6L}\min\left\{1, \frac{\sqrt{S}}{\beta}\right\}$ ($\tilde{\eta} = \frac{1}{6L(\beta+1)} \leq \min\left\{1, \frac{\sqrt{S}}{\beta}\right\}$) we have

$$
\begin{aligned}
\mathcal{L}(\bar{\boldsymbol{w}}^{(R)}) - \mathcal{L}(\boldsymbol{w}^*) \leq & 2\left(\frac{L\sigma^2}{S^2 K}\right)^{\frac{1}{2}}\left(\frac{\mathbb{E}\left[\mathcal{L}(\boldsymbol{w}^{(0)}) - \mathcal{L}^*\right]}{R+1}\right)^{\frac{1}{2}} \\
& + 2\left(\frac{9L^2\sigma^2}{8SK} + \frac{9L^2\zeta^2}{8S}\right)^{\frac{1}{3}}\left(\frac{\mathbb{E}\left[\mathcal{L}(\boldsymbol{w}^{(0)}) - \mathcal{L}^*\right]}{R+1}\right)^{\frac{2}{3}} + \frac{6L\mathbb{E}\left[\mathcal{L}(\boldsymbol{w}^{(r)}) - \mathcal{L}^*\right]}{R+1}
\end{aligned}
$$

Due to the concave nature of the power function(Lemma.2), we can get

$$
\begin{aligned}
\mathcal{L}(\bar{\boldsymbol{w}}^{(R)}) - \mathcal{L}(\boldsymbol{w}^*) \leq & 2\left(\frac{L\sigma^2}{S^2 K}\right)^{\frac{1}{2}}\left(\frac{\mathbb{E}\left[\mathcal{L}(\boldsymbol{w}^{(0)}) - \mathcal{L}^*\right]}{R+1}\right)^{\frac{1}{2}} \\
& + 2\left[\left(\frac{9L^2\sigma^2}{8SK}\right) + \left(\frac{9L^2\zeta^2}{8S}\right)\right]^{\frac{1}{3}}\left(\frac{\mathbb{E}\left[\mathcal{L}(\boldsymbol{w}^{(0)}) - \mathcal{L}^*\right]}{R+1}\right)^{\frac{2}{3}} + \frac{6L\mathbb{E}\left[\mathcal{L}(\boldsymbol{w}^{(r)}) - \mathcal{L}^*\right]}{R+1}
\end{aligned}
$$

After finishing, we can get:

$$
\min_{0 \leq r \leq R} \mathbb{E}\left[\left\|\nabla\mathcal{L}(\boldsymbol{w}^{(r)})\right\|^2\right] \leq \frac{3\left(\mathcal{L}(\boldsymbol{w}^0) - \mathcal{L}^*\right)}{\tilde{\eta}R} + \frac{3L\tilde{\eta}\sigma^2}{S^2 K} + \frac{27L^2\tilde{\eta}^2\sigma^2}{8SK} + \frac{27L^2\tilde{\eta}^2\zeta^2}{8S} \quad (71)
$$

where we use $\min_{0 \leq r \leq R} \mathbb{E}\left[\left\|\nabla\mathcal{L}(\boldsymbol{w}^{(r)})\right\|^2\right] \leq \frac{1}{R+1}\sum_{r=0}^{R}\mathbb{E}\left[\left\|\nabla\mathcal{L}(\boldsymbol{w}^{(r)})\right\|^2\right]$ Then, using $\tilde{\eta} = \frac{1}{6L(\beta+1)} \leq \min\left\{1, \frac{\sqrt{S}}{\beta}\right\}$ and by dynamically adjusting the learning rate, we have

$$
\min_{0 \leq r \leq R} \mathbb{E}\left[\left\|\nabla\mathcal{L}(\boldsymbol{w}^{(r)})\right\|^2\right] = \mathcal{O}\left(\frac{(L\sigma^2\mathcal{B})^{1/2}}{\sqrt{SKR}} + \frac{(L^2\sigma^2\mathcal{B}^2)^{1/3}}{(S^2 K)^{1/3}R^{2/3}} + \frac{(L^2\zeta^2\mathcal{B}^2)^{1/3}}{S^{1/3}R^{2/3}} + \frac{L\beta\mathcal{B}}{R}\right) \quad (72)
$$

where $\mathcal{B} := \mathcal{L}(\boldsymbol{w}^0) - \mathcal{L}^*$. Eq.71 and Eq. 72 are upper bounds for the case of partial client participation. In particular, when $S = M$, we obtain the conclusions of Theorem 1 in the non-convex case.

# H  Theoretical Analysis of Update Order Sensitivity

What differentiates PFL and SFL is that the current updated gradient is dependent on the gradient from the previous round; thus, we represent their dependency using conditional probability, as follows Eq. 73. This is why order sensitivity exists.

$$
\mathcal{L}_{\pi_m^n, k}\left(f(x_i^{\pi_m^n, k}; \boldsymbol{w}_{\pi_m^n, k}), y_i^{\pi_m^n, k} | \sum_{j=1}^{m-1} \nabla\mathcal{L}_{\pi_j^n, k}\right) \neq \mathcal{L}_{\pi_m^n, k}\left(f(x_i^{\pi_m^n, k}; \boldsymbol{w}_{\pi_m^n, k}), y_i^{\pi_m^n, k}\right) \quad (73)
$$

We provide two examples to illustrate that the update order in SFL introduces variance: one from the perspective of loss function differences, and the other from the data distribution perspective.

## H.1  Loss Function Differences

We consider two clients $A$ and $B$, each taking $K = 1$ steps of SGD in each training round, with a learning rate of $\eta$.

**Definition:**

- $x^{(r)}$: the global model at the beginning of round $r$;

- The local gradient of client A is $g_A(x) = \nabla F_A(x)$, and that of client B is $g_B(x) = \nabla F_B(x)$.

Order 1 (A $\rightarrow$ B):

$$x^{(r)} \xrightarrow{\text{A}} x_1 = x^{(r)} - \eta g_A(x^{(r)}) \xrightarrow{\text{B}} x^{(r+1)} = x_1 - \eta g_B(x_1)$$

Thus:

$$x^{(r+1)} = x^{(r)} - \eta g_A(x^{(r)}) - \eta g_B\left(x^{(r)} - \eta g_A(x^{(r)})\right)$$

Order 2 (B $\rightarrow$ A):

$$x^{(r)} \xrightarrow{\text{B}} x_1' = x^{(r)} - \eta g_B(x^{(r)}) \xrightarrow{\text{A}} x_{\text{alt}}^{(r+1)} = x_1' - \eta g_A(x_1')$$

Thus:

$$x_{\text{alt}}^{(r+1)} = x^{(r)} - \eta g_B(x^{(r)}) - \eta g_A(x^{(r)} - \eta g_B(x^{(r)}))$$

Here is the translated and refined version in academic style:

As an example, let $F_A(x) = \frac{1}{2}(x-1)^2$, $F_B(x) = \frac{1}{2}(x+1)^2$, representing left- and right-biased objectives, respectively:

That is, $\nabla F_A(x) = x - 1$, $\nabla F_B(x) = x + 1$. We get:

$$\begin{aligned}
x^{(r+1)} &= x^{(r)} - \eta g_A(x^{(r)}) - \eta g_B\left(x^{(r)} - \eta g_A(x^{(r)})\right) \\
&= x^{(r)} - \eta(x^{(r)} - 1) - \eta(x^{(r)} - \eta(x^{(r)} - 1) + 1) \\
&= (1 - 2\eta + \eta^2)x^{(r)} - \eta \\
x_{\text{alt}}^{(r+1)} &= x^{(r)} - \eta g_B(x^{(r)}) - \eta g_A(x^{(r)} - \eta g_B(x^{(r)})) \\
&= x^{(r)} - \eta(x^{(r)} + 1) - \eta(x^{(r)} - \eta(x^{(r)} + 1) - 1) \\
&= (1 - 2\eta + \eta^2)x^{(r)} + \eta
\end{aligned}$$

By substituting specific values (e.g., $x^{(r)} = 0$, $\eta = 0.1$), we can verify that the results differ under different update orders. Clearly, $x_{\text{alt}}^{(r+1)} \neq x^{(r+1)}$. When the loss functions are inconsistent, the update order in SFL influences the gradient used for model updates.

## H.2 The Data Distribution Differences

Suppose two clients, A and B, have different local data distributions $\mathcal{D}_A \neq \mathcal{D}_B$. Then their corresponding local objective functions differ:

- $F_A(x) = \mathbb{E}_{\xi \sim \mathcal{D}_A}[f(x; \xi)]$
- $F_B(x) = \mathbb{E}_{\xi \sim \mathcal{D}_B}[f(x; \xi)]$

Therefore, for the same $x$, their gradients are also different:

$$\nabla F_A(x) \neq \nabla F_B(x) \tag{74}$$

This defines the mathematical nature of data heterogeneity: gradient diversity, i.e., inconsistency in gradient distributions.

**Example:** Suppose we use MSE loss (Mean Squared Error)

For a linear regression task:

- Client A's data: target value $y = ax$
- Client B's data: target value $y = bx$

Let the model be $\hat{y} = \theta x$. The loss function is:

$$f(x; \xi) = \frac{1}{2}(\hat{y} - y)^2 = \frac{1}{2}(wx - y)^2 \tag{75}$$

Then:

- The expected loss gradient for client A is:

$$\nabla F_A = \mathbb{E}_x \left[ \frac{\partial}{\partial w} \left( \frac{1}{2}(wx - ax)^2 \right) \right] = \mathbb{E}_x \left[ (wx - ax)x \right]$$

- The expected loss gradient for client B is:

$$\nabla F_B = \mathbb{E}_x \left[ \frac{\partial}{\partial w} \left( \frac{1}{2}(wx - bx)^2 \right) \right] = \mathbb{E}_x \left[ (wx - bx)x \right]$$

If the distribution of $x$ is the same (e.g., $x$ follows a standard normal distribution), then:

$$\nabla F_A(w) = (w - a)\mathbb{E}[x^2], \quad \nabla F_B(w) = (w - b)\mathbb{E}[x^2] \tag{76}$$

Order 1 (A → B):

$$
\begin{aligned}
w^{(r+1)} &= w^{(r)} - \eta \nabla F_A(w^{(r)}) - \eta \nabla F_B \left( w^{(r)} - \eta \nabla F_A(w^{(r)}) \right) \\
&= w^{(r)} - \eta(w^{(r)} - a)\mathbb{E}[x^2] - \eta(w^{(r)} - \eta(w^{(r)} - a)\mathbb{E}[x^2] - b)\mathbb{E}[x^2] \\
&= w - 2\eta w \mathbb{E}[x^2] + \eta^2 w \mathbb{E}^2[x^2] - (a + b)\eta \mathbb{E}[x^2] - a\eta^2 \mathbb{E}^2[x^2]
\end{aligned}
$$

Order 2 (B → A):

$$
\begin{aligned}
w_{alt}^{(r+1)} &= w^{(r)} - \eta \nabla F_B(w^{(r)}) - \eta \nabla F_A \left( w^{(r)} - \eta \nabla F_B(w^{(r)}) \right) \\
&= w^{(r)} - \eta(w^{(r)} - b)\mathbb{E}[x^2] - \eta(w^{(r)} - \eta(w^{(r)} - b)\mathbb{E}[x^2] - a)\mathbb{E}[x^2] \\
&= w - 2\eta w \mathbb{E}[x^2] + \eta^2 w \mathbb{E}^2[x^2] - (a + b)\eta \mathbb{E}[x^2] - b\eta^2 \mathbb{E}^2[x^2]
\end{aligned}
$$

By substituting specific values (e.g., $a = 1$, $b = 2$), we can verify that the results differ under different update orders. Clearly, $w_{alt}^{(r+1)} \neq w^{(r+1)}$. When the Data Distributions are inconsistent, the update order in SFL influences the gradient used for model updates.

# I   More Experimental Details

## I.1   Dataset.

We researched our proposed method using two types of datasets. One category consists of datasets with a category shift, which necessitates our partitioning. Our partitioning is based on the Dirichlet distribution[35] Dir $\beta$. Where $\beta$ (default 0.5) is an argument correlated with the heterogeneity level. In this problem, we set the default number of clients to 10. There are three data sets we need to divide here, CIFAR-10[42], CIFAR-100[42], and CINIC-10[43]. CIFAR-10 is a widely used dataset consisting of 60,000 32x32 color images in 10 classes, with 6,000 images per class, primarily used for image classification tasks. CIFAR-100 is similar to CIFAR-10, but with 100 classes containing 600 images each, providing a more granular challenge for image classification tasks. CINIC-10 is an extended version of CIFAR-10, containing 270,000 images split into 10 classes, designed to bridge the gap between CIFAR-10 and ImageNet[60] for improved training scalability.

For domain shift, we conducted experiments on two benchmark datasets: PACS [61] and Office-Home[62]. (*i*) The **PACS** [61] dataset contains four distinct domains (*Photos, Art Paintings, Cartoons, and Sketches*), with a total of 9,991 images. Each domain shares the same 7-class label space, despite the variations in image styles.

(*ii*) The **Office-Home** [62] dataset contains approximately 15,500 images across 65 categories from four domains (*Art, Clipart, Product, and Real-World*), offering a diverse range of categories that better test the robustness of our method. We have included the Office-Home experiment in the supplementary materials.

Table 9: Comparison of SPFL and PFL on Tiny-ImageNet and CIFAR-100 across various FL methods (Dirichlet $\alpha = 0.1$).

| Method | Tiny-ImageNet | Tiny-ImageNet (SPFL) | CIFAR-100 | CIFAR-100 (SPFL) |
|---|---|---|---|---|
| FedAvg | 15.69 | 33.82 (+18.13) | 57.55 | **64.33 (+6.78)** |
| FedProx | 15.47 | 16.65 (+1.18) | 57.56 | 60.46 (+2.90) |
| MOON | 15.42 | 16.47 (+1.05) | 58.44 | 63.54 (+5.20) |
| FedDyn | 15.22 | 17.66 (+2.44) | 64.11 | 64.24 (+0.13) |
| Scaffold | 14.58 | 18.16 (+3.58) | 56.15 | 65.05 (+8.90) |
| FedDC | 15.37 | 16.45 (+1.08) | **64.44** | **69.88 (+5.44)** |
| FedNova | 15.53 | 16.45 (+0.98) | 57.64 | 64.47 (+6.83) |
| FedDisco | **33.55** | **34.50 (+0.95)** | 57.50 | 64.44 (+6.94) |

Table 10: Accuracy (%) on PACS under Domain Shift Setting.

| Method | P | A | C | S | Avg |
|---|---|---|---|---|---|
| PFL (120R) | 91.02 | 68.61 | **70.85** | 66.84 | 74.33 |
| PFL (100R)+SPFL (20R) | 93.11 | 73.24 | 69.36 | 64.92 | 75.16 |
| PFL (100R)+SPFL *with* GLAM (20R) | **93.71** | **74.45** | 70.21 | 64.45 | **75.71** |

Table 11: Performance comparison of GLAM+SPFL with recent FL methods under domain shift.

| Method | P | A | C | S | Avg |
|---|---|---|---|---|---|
| CAN [18]+SPFL | 90.06 | 70.32 | **77.63** | 66.47 | 76.12 |
| FedDA [51]+SPFL | 90.41 | 73.83 | 69.01 | **74.01** | 76.82 |
| FedGALA [64]+SPFL | 87.58 | 68.58 | 68.17 | 66.56 | 72.72 |
| GLAM+SPFL (Ours) | **92.28** | **77.44** | 74.40 | 67.32 | **77.86** |

## I.2 Implementation Details.

When we are in the domain shift setting, for local model training across the PACS and OfficeHome datasets, we utilize architectures ResNet18 and ResNet50, as detailed by [3], which are pre-trained on the ImageNet [60]. We adopt a leave-one-domain-out evaluation method for all benchmarks, where one domain is reserved for testing and the remaining domains are used for training and validation. To ensure consistency and fairness in experiments, we standardize the batch size and learning rate at 128 and 0.2 in local training. Furthermore, to guarantee that the local models reach convergence within each training phase, we set the number of local epochs $E$ to 1 and define the communication rounds $R$ as 100.

In the category shift setting, we utilized a pre-trained model from PFL (FedAvg) that was trained for 100 rounds. Subsequently, we ran our new algorithm for an additional 50 rounds. For datasets Cifar10 and Cifar100, we employed the ResNet18 model framework, while for dataset CINIC-10, we utilized a Simple-CNN [63] model framework. We consistently employed Accuracy (acc) as the performance metric for our evaluations.

In the training phase, we conducted our experiments on a single NVIDIA Tesla A800 GPU. We used four NVIDIA Tesla A800 GPUs during testing to obtain results more quickly. However, the experiments only required 5-10GB of GPU memory due to a batch size of 128.

## I.3 More comparative experiments

**Performance comparison on complex datasets.** Tab. 9 compares SPFL and PFL on Tiny-ImageNet and CIFAR-100 under different FL methods. SPFL consistently outperforms PFL across all settings, showing significant gains in both accuracy and convergence speed. These results demonstrate that SPFL achieves better generalization with most FL methods.

**Performance of SPFL and GLAM under hybrid strategies.** As shown in Tab. 10, when adopting a hybrid strategy, SPFL maintains a lower convergence margin on the main shift but converges more

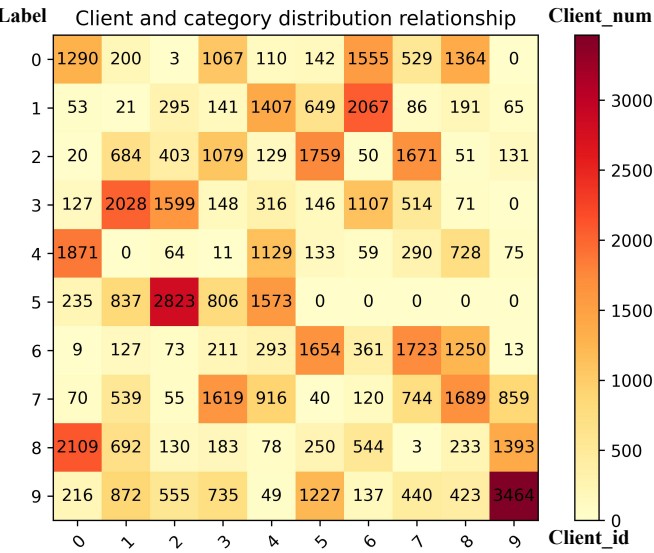

Figure 6: Distribution relationship between client and category

Table 12: The combination experiments of the two different loss functions were conducted on the Office-Home dataset using a pre-trained ResNet-18 (Best in **bold**)

| Method | $\mathcal{L}_{ag}$ | $\mathcal{L}_{ap}$ | *Product* | *Art* | *Clipart* | *Real-World* | Avg |
|--------|------|------|---------|-------|---------|-------------|-------|
| SPFL | × | × | 57.22 | 43.10 | 46.07 | 59.22 | 51.40 |
| SPFL | ✓ | × | **58.08** | 43.88 | 47.10 | 59.10 | 52.04 |
| SPFL | × | ✓ | 56.75 | 44.27 | 47.62 | **59.49** | 52.03 |
| SPFL | ✓ | ✓ | 57.40 | **44.47** | **47.40** | 59.20 | **52.14** |

slowly. Therefore, first applying PFL to reach the boundary, followed by a low number of SPFL rounds, can achieve a lower convergence margin. Meanwhile, GLAM remains effective under this hybrid strategy.

**Comparison with state-of-the-art methods.** CAN [18] represents the latest federated continual learning approach [65, 26], while FedDA [51] and FedGala [64] are recent advances in federated domain generalization. However, unlike GLAM, these methods do not align the model transmitted from the previous client, leading to limited performance gains when adapting to new domains. In contrast, GLAM provides a more comprehensive alignment mechanism, allowing flexible adjustment of the comparison strength through parameters $\tau$ and $\rho$, as shown in Tab. 11.

### I.4  More Ablation Study

**Performance in Office-Home.** Through ablation studies on PACS and Office-Home, the effectiveness of the GLAM module is thoroughly demonstrated, with each component contributing to the overall model performance, as shown in Fig. 12.

**Comparison of different update methods.** We compare the two existing update schemes, PFL and SFL, with our proposed SPFL. As shown in Tab. 13, SPFL consistently outperforms both methods across various FL frameworks, demonstrating its superior effectiveness and adaptability. To quickly converge, the initial model was initialized using a FedAvg pre-trained for 200 rounds on CIFAR-10, with 10 clients.

**Comparison of different Dirichlet $\alpha$.** We conduct an ablation study on parameter $\alpha$. As shown in Tab. 14, the performance improvement of SPFL becomes more pronounced as the degree of non-IID increases.To quickly converge, the initial model was initialized using a FedAvg pre-trained for 200 rounds on CIFAR-10, with 10 clients.

Table 13: Comparison of different update strategies (PFL, SFL, and our SPFL) under various FL methods on CIFAR-10 (Dirichlet $\alpha = 0.1$).

| Update\Method | FedAvg | FedProx | FedDC | Moon |
|---|---|---|---|---|
| PFL | 55.60 | 56.65 | 55.16 | 56.33 |
| SFL | 68.59 | 53.75 | 54.41 | 55.24 |
| SPFL (ours) | **71.39** | **57.66** | **55.26** | **58.44** |

Table 14: Accuracy (%) of SPFL and PFL (FedAvg) under different Dirichlet $\alpha$ settings.

| Update\Dir | 0.1 | 0.2 | 0.5 | 1 | 100 |
|---|---|---|---|---|---|
| PFL (FedAvg) | 39.25 | 43.40 | 42.85 | 46.03 | 53.37 |
| SPFL (Ours) | **47.55** | **50.96** | **49.38** | **49.47** | **55.27** |

Table 15: Accuracy (%) of PFL, multiple SFL variants, and SPFL under different Dirichlet $\alpha$ settings.

| Method | PFL | SFL(1) | SFL(2) | SFL(3) | SFL(4) | SFL(5) | SPFL (ours) |
|---|---|---|---|---|---|---|---|
| Dir(0.1) | 41.85 | 49.56 | 54.45 | 49.51 | 47.98 | **55.56** | **55.33** |
| Dir(0.5) | 43.60 | 50.22 | 47.95 | **55.18** | 54.20 | 49.52 | **55.16** |

**Comparison of different update orders.**   We conduct experiments with five clients for training, where SFL($i$) denotes using client $i$ as the starting point. As shown in Tab. 15, different starting points result in significant performance variations, highlighting the update sensitivity problem in SFL. In contrast, SPFL effectively mitigates this issue, achieving more stable and consistent performance across clients.

**Effect of client participation rate.**   Tab. 16 compares PFL and SPFL under different client participation rates ($E$) with 100 clients. As $E$ increases, SPFL consistently outperforms PFL, showing greater stability and accuracy, which demonstrates its stronger adaptability to varying participation levels.

**Effect under Category and Domain Shifts.**   As shown in Tab. 17, SPFL significantly improves performance under both category and domain shifts compared to PFL and SFL, demonstrating its stronger generalization capability. Furthermore, integrating the GLAM module further enhances performance, especially under domain shift, by effectively aligning inter-domain representations. This confirms that SPFL addresses update sensitivity, while GLAM strengthens cross-domain consistency and adaptability.

# J   Limitation

Although SPFL demonstrates strong performance in addressing category shift and achieves comparable results to PFL under domain shift, this improvement imposes stricter requirements on the deployment environment, specifically, it assumes that clients remain continuously online. Additionally, while increasing the number of clients can accelerate convergence and lower the theoretical convergence upper bound, it also introduces substantial communication overhead. As a result, SPFL is better suited for cross-silo scenarios rather than cross-device settings.

Moreover, cross-silo environments typically entail higher computational costs, whereas SPFL is able to match the performance of multi-epoch PFL using significantly fewer epochs. Therefore, although SPFL is theoretically applicable to cross-device scenarios, it is more practically aligned with cross-silo applications. A detailed comparison of computational and communication costs is presented in Tab. 18.

Where $\mathcal{M}$ is the model size, $M$ is the number of clients, $E$ is the epoch, and $T$ is the number of communication rounds.

Table 16: Accuracy comparison between PFL and SPFL under different client participation rates ($E$) with 100 total clients.

| Update\E | 0.1 | 0.2 | 0.3 | 0.4 | 0.5 | 0.6 | 0.7 | 0.8 | 0.9 | 1.0 |
|---|---|---|---|---|---|---|---|---|---|---|
| PFL | 41.78 | 46.99 | 52.43 | 53.31 | 54.04 | 54.52 | 54.47 | 54.45 | 54.35 | 54.59 |
| SPFL | 51.72 | 63.57 | 72.33 | 76.19 | 80.84 | 82.68 | 84.43 | 85.41 | 85.98 | **89.95** |

Table 17: Performance under Category and Domain Shift with PFL, SFL, SPFL, and SPFL+GLAM.

| Shift Type | PFL | SFL | SPFL | SPFL+GLAM |
|---|---|---|---|---|
| Category Shift | 55.60 | 68.59 | 71.39 | **71.71** |
| Domain Shift (Avg) | **78.37** | 60.04 | 76.32 | 77.87 |

Table 18: Cost calculation analysis

| Method | Calculate costs | Communication costs |
|---|---|---|
| PFL | $T \times M \times E \times 3 \times \mathcal{M} \times Batch\_Size$ | $T \times K \times 2 \times \mathcal{M} \times 4$ |
| SFL | $T \times M \times E \times 3 \times \mathcal{M} \times Batch\_Size$ | $T \times M \times \mathcal{M} \times 4$ |
| SPFL | $T \times M \times E \times 3 \times \mathcal{M} \times Batch\_Size \times K$ | $T \times M \times 2 \times \mathcal{M} \times 4 \times (K + 1/2)$ |

# K   Broader impacts

The proposed SPFL framework enhances the robustness and generalization of federated learning systems under category and domain shifts, potentially benefiting applications involving privacy-sensitive and heterogeneous data, such as healthcare, finance, and education. By reducing the dependency on data centralization, it supports data sovereignty and regulatory compliance.

However, stronger model generalization across clients may inadvertently increase the risk of model inversion or membership inference attacks. Additionally, more complex update schemes could amplify computational inequality between clients with varying resources. Future work should consider fairness and security safeguards to mitigate these risks.

