# OpenReview forum: "SPFL: Sequential updates with Parallel aggregation for Enhanced Federated Learning under Category and Domain Shifts"
_NeurIPS.cc/2025/Conference — NeurIPS 2025 poster_

### Official Review · Reviewer_wzqr · 2025-06-19

**Clarity:** 1
**Significance:** 2
**Originality:** 2
**Rating:** 3
**Confidence:** 3

**Summary:**

This paper proposes SPFL, a novel federated learning framework that combines sequential updates with parallel aggregation to enhance data utilization and mitigate update order sensitivity in non-identically distributed (non-IID) settings. By introducing a Global-Local Alignment Module (GLAM) to mitigate catastrophic forgetting and providing convergence guarantees under various conditions, SPFL achieves better or comparable performance compared to traditional PFL and SFL methods in scenarios involving category and domain shifts.

**Questions:**

- Are there better ways to combine sequential and parallel FL to better balance computational cost and performance?
- See weaknesses above.

**Ethical Concerns:**

["NO or VERY MINOR ethics concerns only"]

**Final Justification:**

Thank you to the authors for their detailed response. After reviewing all the reviews and responses, I observe that the authors adopt a new two-stage training method, first using standard PFL, followed by a small number of SPFL rounds, to mitigate performance issues. This hybrid training scheme requires modifications to the original algorithm and presentation. Therefore, I have decided to maintain my original rating.

**Limitations:**

Yes.

**Quality:**

2

**Strengths And Weaknesses:**

Strengths:
- The setting is interesting, and the problem is important to investigate for domain and category shifts in FL. Also, it is intuitively useful and helpful to investigate combining sequential and parallel FL for better performance under distributional shifts.
- The authors provide theoretical guarantees (compared with other baselines) for their framework.

Weaknesses:
- In sections 3.1 and 3.2, it is unclear to me what the differences are among k, m, and n for the loss equations and their lack of explanations for that (which is confusing).
- The performance of this paper's framework is kind of underwhelming for me - on PACS, it is worse than FedAvg on average, yet the theoretical analysis seems to show it should be better than FedAvg.
- I felt like the trade-off between efficiency and performance is not addressed very well in this paper: SPFL will increase the computational cost significantly, yet the performance improvements are limited.
- The related work is not sufficiently discussed, especially the works on federated domain adaptation [1] /generalization [2], e.g., they used gradient matching methods to find the optimal solution for the new data distribution - It would be better to write a paragraph on this topic in the related work for distribution shifts in federated learning.

[1] Principled Federated Domain Adaptation: Gradient Projection and Auto-Weighting.

[2] Federated Domain Generalization with Data-free On-server Matching Gradient.

---

> ### Author Rebuttal · Authors · 2025-07-30
>
> Dear Reviewer wzqr：
>
> We sincerely thank you for the constructive and insightful feedback. We greatly appreciate your recognition of the significance of addressing domain and category shifts in federated learning, as well as the intuitive value of integrating sequential and parallel update strategies. Your comments regarding theoretical clarity, the trade-off between performance and efficiency, and the discussion of related work are highly valuable and will substantially enhance the quality of the final version. Below, we provide detailed responses to each of your concerns.
>
> **Note: To quickly converge, the most model below was initialized using a FedAvg pre-trained checkpoint trained for 200 rounds on CIFAR-10; the number of clients is 10.**
> ### Response to Weakness
> >**W1** :In sections 3.1 and 3.2, it is unclear to me what the differences are among k, m, and n for the loss equations and their lack of explanations for that (which is confusing).
>
> We thank you for raising this question. As also noted by other reviewers, the current placement of key notations may hinder readability. Although the full definitions of $k$, $m$, and $n$ are provided in the appendix, we agree that introducing these symbols earlier in the main text would significantly improve clarity. For completeness, we briefly clarify them here:
>
> * $K , k$  denotes the **index of local update steps** ;
> * $M , m$ denotes the number of participating clients and the **Client index** in the current sequential update order;
> * $ n$ represents   the starting **Client index**.
>
> In the final version, we will revise Sections 3.1 and 3.2 to move the definitions of key notations closer to where they are first used, and we have a summary table of notations for quick reference in the appendix. We appreciate your attention to this detail, as it helps improve the overall accessibility and precision of the theoretical exposition.
>
> >**W2:** The performance of this paper's framework is kind of underwhelming for me - on PACS, it is worse than FedAvg on average, yet the theoretical analysis seems to show it should be better than FedAvg.
>
> Thank you for your insightful observation. We acknowledge that the performance of SPFL on PACS appears less impressive, particularly in the presence of a significant domain shift. Theoretically, most existing convergence analyses—including ours—assume **bounded model variance $\sigma$**. However, in cross-domain scenarios like PACS, this assumption no longer holds, as domain-induced distribution gaps can cause the variance  $\sigma$ to grow unbounded, leading to loose or invalid convergence guarantees. We acknowledge this limitation and will consider extending our analysis to the unbounded variance regime in future work.
>
> Moreover, SPFL was originally designed to tackle **category shift**, where it demonstrates consistent advantages over PFL. Nevertheless, to adapt SPFL to more challenging **domain shift** scenarios and in response to reviewer suggestions, we developed a **two-stage training strategy**: we first perform standard PFL to enable the model to acquire basic recognition ability, and then apply a few SPFL rounds to enhance generalization by exposing the model to more diverse domain knowledge through client rotation. And our GLAM module mitigates domain-induced forgetting by aligning client updates and improving consistency.
>
>  As shown in Tab 1, this hybrid strategy—first training with PFL followed by a few rounds of SPFL with GLAM—demonstrates that **SPFL can be effectively leveraged to improve performance on PACS**, despite its limited effectiveness when used alone. R represents the number of training rounds.
>
> **Tab 1: Accuracy (%) on PACS under Domain Shift Setting**
>
> |Method|P|A|C|S|Avg|
> |-|-|-|-|-|-|
> |PFL(120R)|91.02|68.61|**70.85**|**66.84**|74.33|
> |PFL(100R)+SPFL(20R)|93.11|73.24|69.36|64.92|75.16|
> |PFL(100R)+SPFL *with* GLAM (20R)|**93.71**|**74.45**|70.21|64.45| **75.71** |
>
>
> >**W3:** I felt like the trade-off between efficiency and performance is not addressed very well in this paper: SPFL will increase the computational cost significantly, yet the performance improvements are limited.
>
> We thank you for raising this important point regarding the trade-off between computational efficiency and performance. While SPFL introduces additional per-round overhead due to its sequential update mechanism, it converges substantially faster than traditional PFL methods, thereby reducing the total communication and computation burden across training. We fully agree that if SPFL were applied uniformly throughout all training rounds, the per-round cost might outweigh its benefits.
>
> **Motivated by reviewer suggestions, we adopt a two-stage training strategy to mitigate this issue: the model is first trained using FedAvg to reach a reasonable initialization, followed by a few rounds of SPFL to rapidly refine and boost generalization. This design retains the efficiency of PFL while leveraging SPFL’s robustness and effectiveness in handling category shifts.**
> As shown in Tab 2 and 3, SPFL achieves superior performance with dramatically fewer communication rounds and lower computational cost:
>
> **Tab 2: Efficiency Comparison on CIFAR-10 (Dirichlet α=0.1,Number of clients=10)**
>
> |Method|Accuracy(%)|Rounds|Comm. Cost (GB)|Total Time(seconds)|Compute Cost(GB)|
> |-|-|-|-|-|-|
> |PFL|79.15|500|116.90|3419.91|175.35|
> |SPFL(Ours)|84.17|15|21.042|678.6|87.675|
>
> **Tab 3: Efficiency Comparison on CIFAR-10 (Dirichlet α=0.1,Number of clients=100)**
>
> |Method|Accuracy(%)|Rounds|Comm. Cost(TB)|Total Time(seconds)|Compute Cost(TB)|
> |-|-|-|-|-|-|
> |PFL|81.38|2000|18.30|30259.82|27.46|
> |SPFL(Ours)|86.68|10|4.68|14713.13|22.85|
>
> These results demonstrate that SPFL’s higher per-round cost is more than offset by its faster convergence, leading to better final performance and greater efficiency. We will highlight this trade-off more clearly in the revised manuscript and foreground the supporting empirical evidence accordingly.
>
> >**W4:** The related work is not sufficiently discussed, especially the works on federated domain adaptation [1] /generalization [2], e.g., they used gradient matching methods to find the optimal solution for the new data distribution - It would be better to write a paragraph on this topic in the related work for distribution shifts in federated learning.
>
> We thank you for pointing out the missing discussion on recent gradient matching methods for federated domain adaptation and generalization. As noted, we have included a section in the appendix discussing more domain generalization (DG) approaches. However, the recent works leveraging gradient matching, such as [1] and [2], represent an important and emerging direction that was not sufficiently addressed in the current draft.
> In the final version, we will revise the Related Work section to explicitly highlight these approaches and move key discussions from the appendix to the main for better visibility. Below is the paragraph we plan to add:
>
> >While most federated learning (FL) methods have made progress in addressing category shift, domain shift remains a more persistent challenge. To this end, AutoFedGP [1] tackles the domain adaptation (DA) problem by projecting local gradients and assigning adaptive weights to align source and target domains. For domain generalization (DG), FedOMG [2] enhances robustness to unseen domains by aligning gradient directions across clients without access to target data. However, both approaches focus on model-level alignment, which may be insufficient when domain shifts induce severe forgetting during sequential client updates. In contrast, our proposed SPFL explicitly targets this issue by introducing GLAM. This global-local alignment mechanism concurrently aligns each client with both its predecessor and the global model, effectively mitigating the forgetting problem caused by domain shift.
>
> [1] Principled Federated Domain Adaptation: Gradient Projection and Auto-Weighting.
>
> [2] Federated Domain Generalization with Data-free On-server Matching Gradient.
>
> ### Response to Question
> >**Q1**:Are there better ways to combine sequential and parallel FL to better balance computational cost and performance?
> We appreciate your interest and recognition of our work. Like most reviewers, we hope that SPFL can serve as a practical and effective alternative to PFL in cross-silo federated learning.
>
> To address the concerns regarding efficiency and optimization, we propose a two-stage improvement strategy informed by reviewers' feedback. First, **we adopt a hybrid training scheme where the model is initially trained using standard PFL, followed by a small number of SPFL rounds to refine the model and enhance generalization under category shifts.** Second, we explore the impact of the client participation ratio during training. Specifically, we conduct experiments with 100 clients and vary the participation ratio $E \in [0.1, 1.0]$. As shown in the tab 4, we find that a moderate participation ratio (e.g., 50%) offers the best trade-off between communication/computation cost and performance, highlighting the robustness of SPFL under partial participation.
>
> **Tab 4: Accuracy comparison between PFL and SPFL under different client participation rates (\$E\$) with 100 total clients.**
>
> |Update\E|0.1|0.2|0.3|0.4|0.5|0.6|0.7|0.8|0.9|1|
> |-|-|-|-|-|-|-|-|-|-|-|
> |PFL|41.78|46.99|52.43|53.31|54.04|54.52|54.47|54.45|54.35|54.59|
> |SPFL|51.72|63.57|72.33|76.19|80.84|82.68|84.43|85.41|85.98|89.95|
>
> **We sincerely appreciate your insightful feedback and have addressed your concerns to the best of our ability. Should you have any further questions or recommendations, we would be glad to engage in further discussion.**

---

> ### Comment · Reviewer_wzqr · 2025-08-04
>
> Thank you to the authors for their detailed response. After reviewing all the reviews and responses, I observe that the authors adopt a new two-stage training method, first using standard PFL (which appears to require a large number of them), followed by a small number of SPFL rounds, to mitigate performance issues. This hybrid training scheme requires modifications to the original algorithm and presentation. Therefore, I have decided to maintain my original rating.

---

> > ### Author Response · Authors · 2025-08-05
> >
> > Dear  Reviewer wzqr,
> >
> > Thank you for your thoughtful follow-up and for carefully considering our response. We understand your concern that the hybrid strategy may appear to deviate from the original SPFL framework. However, we would like to emphasize that this strategy is not a core change to the SPFL algorithm itself, but rather a practical training schedule designed to enhance communication efficiency, particularly in environments with high client heterogeneity. **At the same time, this hybrid strategy is plug-and-play and introduces no additional algorithmic complexity.**
> >
> > To fully address your concern, we will separate the core SPFL framework and the optional hybrid extension in the final version. Specifically, we will:
> >
> > - Focus the main presentation on the original SPFL algorithm and its theoretical contributions;
> >
> > - **Move the hybrid training experiments and discussions to the supplementary material, with a clear note that the hybrid strategy is not an innovation of SPFL but an efficient solution for specific practical applications.**
> >
> > We sincerely appreciate your critical feedback, which will help us improve the clarity and rigor of the final manuscript.

---

### Official Review · Reviewer_cWLJ · 2025-06-23

**Clarity:** 4
**Significance:** 4
**Originality:** 4
**Rating:** 5
**Confidence:** 4

**Summary:**

This paper proposes a novel update framework, SPFL, which effectively addresses category and domain shift in federated learning. By combining the data efficiency of sequential updates with the stability of parallel aggregation, SPFL significantly reduces update order sensitivity and improves model generalization in non-IID settings. The introduction of the GLAM module further alleviates catastrophic forgetting by aligning local and global predictions. Theoretical convergence analysis under strong, general, and non-convex cases, along with extensive experiments on benchmarks like CIFAR-10 and PACS, demonstrates that SPFL consistently outperforms existing methods and offers a powerful new direction for federated optimization. In addition, by solving category shift, SPFL points the way for cross-soli FL, showing strong innovation and practical significance. The authors further demonstrate through experiments that even when PFL is allocated twice the computational budget of SPFL, it still struggles to outperform SPFL. This result presents a remarkable departure from conventional Cognition in the FL, which may lead to the SPFL replacing the PFL in cross-soli scenarios in the future.

**Questions:**

1. The author shows that SPFL is compatible with multiple PFL algorithms, such as FedAvg, FedDyn, etc. Is it necessary to make special adjustments to each method when adapting? Are there any optimization methods that are not compatible?
2. The number of clients set by the authors in the experiment seems to be fixed, and the impact of the change in the number of clients on the performance of the method has not been explored. However, in the convergence analysis, the number of clients has an important impact on the convergence speed. Therefore, it is recommended that the authors further evaluate the actual impact of the number of clients on the effect and convergence performance of the SPFL method.
3. I encourage the authors to move the key experimental settings from the appendix to the main text in the final version. Presenting these details upfront would improve the clarity and accessibility of the experimental section, making it easier for readers to follow and reproduce the results.

**Ethical Concerns:**

["NO or VERY MINOR ethics concerns only"]

**Final Justification:**

I thank the authors for their response, which has addressed most of my concerns. While I still have some remaining questions and concerns, I’ve seen relevant experiments and analyses in the responses to other reviewers. The transferability of unlearnable examples in FL is indeed a novel and promising direction that deserves more attention from the community. Overall, I think the current theoretical analysis and experimental results are sufficient to support the SPFL work. I will maintain my score and recommend acceptance.

**Limitations:**

Yes.

**Paper Formatting Concerns:**

NA.

**Quality:**

4

**Strengths And Weaknesses:**

**Strengths:**

1. This paper proposes a novel "multi-start sequential update and parallel aggregation" strategy, which significantly improves data utilization across clients. This hybrid approach bridges the gap between federated learning and centralized learning, yielding remarkable empirical results. SPFL provides a novel solution idea for category shift in PFL, enabling global models to achieve improved generalization in heterogeneous data. Meanwhile, it establishes a new paradigm for SFL by alleviating the challenge of update order sensitivity in SFL.
2. The authors provide a thorough convergence analysis under strong convexity, general convexity, and non-convex conditions. This theoretical rigor not only validates the effectiveness of SPFL but also contributes valuable insights to the broader understanding of optimization dynamics in SPFL.
3. The GLAM leverages KL divergence and mutual cross-entropy loss to align the current model with both the previous local model and the global model. This design effectively mitigates catastrophic forgetting during SFL, providing a practical solution to one of the core challenges in SPFL. This work represents the first attempt to introduce alignment mechanisms within the SPFL framework, offering valuable insights and a strong reference for future research in SFL.
4. The experiment covers category shift and domain shift, multiple baseline methods (FedAvg, FedProx, SCAFFOLD, etc.), multiple data sets (CIFAR-10, PACS, etc.), and provides ablation experiments and hyperparameter sensitivity analysis
5. SPFL does not rely on weighted aggregation of client data volume, improves privacy protection capabilities, and has good adaptability in privacy-sensitive application scenarios.

**Weaknesses:**

1. Although the authors point out that the problem of high computational cost per round tends to converge after multiple rounds of training, SPFL performs multiple sequential trainings in each round of updates, which may not be friendly to resource-constrained devices.
2. Category shift has always been a common problem in federated learning, but the author's experiment involved Dirichlet distribution and did not seem to consider the impact of different $\beta$ on the experiment.

---

> ### Author Rebuttal · Authors · 2025-07-30
>
> Dear Reviewer cWLJ：
>
> We sincerely thank you for the encouraging and thoughtful feedback. We are glad that you recognize the novelty of our SPFL framework and the module of GLAM, the theoretical and empirical contributions, and the practical significance of addressing category and domain shift in federated learning. We also appreciate the constructive suggestions regarding client scalability analysis, compatibility evaluation, and presentation clarity, which we will add in the final version.
>
> **Note: In order to quickly converge,the most model below was initialized using a FedAvg pre-trained checkpoint trained for 200 rounds on CIFAR-10,the number of clients is 10.**
> ### Response to Weakness
> >**W1**:Although the authors point out that the problem of high computational cost per round tends to converge after multiple rounds of training, SPFL performs multiple sequential trainings in each round of updates, which may not be friendly to resource-constrained devices.
>
> Thank you for pointing this out. We agree that SPFL introduces additional coordination for sequential and parallel updates. We acknowledge that SPFL increases the per-round communication volume by a factor of M/2 and the computational cost by approximately 5M/3 during training.
> **However, since SPFL converges faster, SPFL effectively reduces the number of training rounds, thereby reducing the overall communication and computation costs, as shown in Tab 1 and 2.**
>
>
> **Tab 1: Efficiency Comparison on CIFAR-10 (Dirichlet α=0.1,Number of clients=10)**
>
> |Method|Accuracy(%)|Rounds|Comm. Cost (GB)|Total Time(seconds)|Compute Cost(GB)|
> |-|-|-|-|-|-|
> |PFL(FedAvg)|79.15|500|116.90|3419.91|175.35|
> |SPFL(Ours)|84.17|15|21.042|678.6|87.675|
>
> **Tab 2: Efficiency Comparison on CIFAR-10 (Dirichlet α=0.1,Number of clients=100)**
>
> |Method|Accuracy(%)|Rounds|Comm. Cost(TB)|Total Time(seconds)|Compute Cost(TB)|
> |-|-|-|-|-|-|
> |PFL(FedAvg)|81.38|2000|18.30|30259.82|27.46|
> |SPFL(Ours)|86.68|10|4.68|14713.13|22.85|
>
> >**W2**:Category shift has always been a common problem in federated learning, but the author's experiment involved the Dirichlet distribution and did not seem to consider the impact of different on the experiment.
>
> Thank you for highlighting the importance of evaluating category shift. While our main experiments were conducted under a strong category shift setting ($\alpha = 0.1$), showing performance under varying heterogeneity levels is vital. We have therefore added results across a broader range of $\alpha$ values, including the IID case ($\alpha = 100$). As shown in tab 3 below, SPFL consistently outperforms PFL (FedAvg), with larger gains under stronger shifts, demonstrating its robustness to different degrees of category imbalance.
>
> **Tab 3: Accuracy (%) of SPFL and PFL (FedAvg) under Different Dirichlet $\alpha$ Settings**
>
> |Update\Dir|0.1|0.2|0.5|1|100|
> |-|-|-|-|-|-|
> |PFL(FedAvg)|39.25|43.40|42.85|46.03|53.37|
> |SPFL(Our)|47.55|50.96|49.38|49.47|55.27|
>
> We will include these additional results and corresponding discussion in the updated manuscript.
>
> ### Response to Question
>
> >**Q1:** The author shows that SPFL is compatible with multiple PFL algorithms, such as FedAvg, FedDyn, etc. Is it necessary to make special adjustments to each method when adapting? Are there any optimization methods that are not compatible?
>
> We thank you for the valuable question. Indeed, Sequential Federated Learning (SFL) is inherently decentralized, which makes it difficult to couple with mainstream Parallel Federated Learning (PFL) methods that rely on server-mediated aggregation. This structural incompatibility limits the practical extension of SFL to more advanced optimization strategies.
>
> In contrast, our proposed SPFL framework reintroduces the server as a coordinator, enabling a seamless integration with existing PFL algorithms such as FedAvg, FedDyn, and others. Since SPFL retains the client-level sequential update pathway while supporting centralized aggregation, it naturally bridges the gap between SFL and PFL.
>
> Importantly, SPFL does not require any modifications to the local update rules of existing methods. Instead, it wraps them into a multi-start sequential training process with parallel aggregation, preserving their theoretical and empirical advantages. This design not only ensures broad compatibility but also enhances convergence in challenging non-IID settings.We will further clarify this coupling capability in the revised manuscript.
>
> >**Q2:** The number of clients set by the authors in the experiment seems to be fixed, and the impact of the change in the number of clients on the performance of the method has not been explored. However, in the convergence analysis, the number of clients has an important impact on the convergence speed. Therefore, it is recommended that the authors further evaluate the actual impact of the number of clients on the effect and convergence performance of the SPFL method.
>
> Thank you for your question. We conducted experiments with 100 clients under a partial participation setting, where only a fraction of clients ($E \in [0.1, 1.0]$) were randomly selected to participate in each round. As shown in the table below, SPFL consistently outperforms PFL across all participation rates, demonstrating strong robustness even with limited client availability. These results further validate the effectiveness of SPFL in partial participation scenarios, aligning well with our theoretical insights.
>
> **Tab 4: Accuracy comparison between PFL and SPFL under different client participation rates (\$E\$) with 100 total clients.**
>
> |Update\E|0.1|0.2|0.3|0.4|0.5|0.6|0.7|0.8|0.9|1|
> |-|-|-|-|-|-|-|-|-|-|-|
> |PFL|41.78|46.99|52.43|53.31|54.04|54.52|54.47|54.45|54.35|54.59|
> |SPFL|51.72|63.57|72.33|76.19|80.84|82.68|84.43|85.41|85.98|89.95|
>
> >**Q3:** I encourage the authors to move the key experimental settings from the appendix to the main text in the final version. Presenting these details upfront would improve the clarity and accessibility of the experimental section, making it easier for readers to follow and reproduce the results.
>
> Thank you for the helpful suggestion. Taking into account the opinions of most reviewers, we will move the key experimental settings (e.g., dataset partitions, client numbers, Dirichlet $\alpha$, model architectures, and training schedules) from the appendix to the main text in the final version to improve clarity and reproducibility.
>
> **We sincerely appreciate your insightful feedback and have addressed your concerns to the best of our ability. Should you have any further questions or recommendations, we would be glad to engage in further discussion.**

---

> > ### Comment · Reviewer_cWLJ · 2025-08-03
> >
> > I thank the authors for their response, which has addressed most of my concerns. While I still have some remaining questions and concerns, I’ve seen relevant experiments and analyses in the responses to other reviewers. The transferability of unlearnable examples in FL is indeed a novel and promising direction that deserves more attention from the community. Overall, I think the current theoretical analysis and experimental results are sufficient to support the SPFL work. I will maintain my score and recommend acceptance.

---

> > > ### Author Response · Authors · 2025-08-03
> > >
> > > Thank you for your reply. We sincerely appreciate your thoughtful questions and recognition of our work. If you have any further comments or suggestions, we would be glad to continue the discussion.

---

### Official Review · Reviewer_ARDv · 2025-06-30

**Clarity:** 2
**Significance:** 3
**Originality:** 3
**Rating:** 5
**Confidence:** 3

**Summary:**

This paper presents a new FL framework named SPFL, designed to enhance learning performance in the presence of category shift and domain shift, two prevalent types of data heterogeneity in FL. The proposed SPFL integrates the strengths of both SFL and PFL by training models sequentially across clients while aggregating in parallel. To address catastrophic forgetting—a common challenge in sequential updates—the authors propose GLAM, which aligns local predictions with global and prior local models. The authors also provide convergence analysis under different convexity assumptions and demonstrate improved performance across various datasets.

**Questions:**

1.	Can the authors elaborate on the computational cost and communication latency of SPFL compared to traditional PFL in large-scale real-world settings (e.g., >100 clients)?
2.	How sensitive is SPFL’s performance to the choice of update order, especially under extreme domain shift? Does parallel aggregation fully mitigate the update-order sensitivity or only reduce it?
3.	Does the proposed GLAM introduce privacy risks by requiring clients to align outputs with previous client models? How is privacy preserved if models are indirectly influenced by earlier clients' data distributions?
4.	Can SPFL work in asynchronous or partial participation settings where not all clients are available in each round?

**Ethical Concerns:**

["NO or VERY MINOR ethics concerns only"]

**Final Justification:**

The authors have addressed the main concerns effectively, particularly the issues related to clarity, communication, and computation costs of SPFL. They have also added valuable comparisons with recent works on catastrophic forgetting and domain adaptation, and provided further experimental results and ablation studies to clarify the contributions of SPFL and GLAM. There are no major unresolved issues, and the revisions have significantly improved the clarity and depth of the paper. Based on the thoroughness of the authors’ responses and the quality of the additional experiments, I recommend accepting this paper.

**Limitations:**

yes

**Quality:**

3

**Strengths And Weaknesses:**

Strengths:

1.	The paper identifies limitations of SFL in the presence of non-IID data and proposes a principled and well-motivated solution (SPFL) that combines the advantages of both SFL and PFL paradigms.
2.	The convergence analysis is carefully presented for strongly convex, general convex, and non-convex settings. The results suggest a theoretically grounded improvement over SFL and PFL baselines.
3.	SPFL is designed as a plug-in update strategy, showing compatibility with many popular FL optimizers.

Weaknesses:

1.	The paper is at times difficult to follow due to dense writing and imprecise descriptions. For example, the mathematical expressions could benefit from clearer notation, and the explanations in sections like 3.2 and 3.4 could be streamlined.
2.	While SPFL is shown to improve accuracy, its communication and computation overheads—especially due to sequential updates across clients and the GLAM module—are not fully discussed. Practical feasibility for large-scale FL deployments remains unclear.
3.	While the paper compares with several popular FL methods, it misses comparisons with recent works focused explicitly on catastrophic forgetting or domain adaptation in FL.
4.	The ablation on hyperparameters is appreciated, but more insight into how much each component (sequential update, parallel aggregation, GLAM) contributes independently to performance would be valuable.

---

> ### Author Rebuttal · Authors · 2025-07-30
>
> Dear Reviewer ARDv：
>
> Thank you for your thoughtful and constructive feedback. We appreciate your recognition of our contributions and  we provide detailed responses to the concerns and questions below.
>
> **Note: In order to quickly converge, the most model below was initialized using a FedAvg pre-trained checkpoint trained for 200 rounds on CIFAR-10, the number of clients is 10.**
> ### Response to Weakness
> >**W1**:The paper is at times difficult to follow due to dense writing and imprecise descriptions. For example, the mathematical expressions could benefit from clearer notation, and the explanations in sections like 3.2 and 3.4 could be streamlined.
>
> Thank you for the helpful suggestion. We will move key definitions from the appendix into the main in the final version and simplify the corresponding descriptions for better clarity. Thanks again for your insightful comments.
>
> >**W2&Q1**:While SPFL is shown to improve accuracy, its communication and computation overheads—especially due to sequential updates across clients and the GLAM module—are not fully discussed. Practical feasibility for large-scale FL deployments remains unclear.&Can the authors elaborate on the computational cost and communication latency of SPFL compared to traditional PFL in large-scale real-world settings (e.g., >100 clients)?
>
> Thank you for your suggestions. We acknowledge that SPFL introduces higher communication and computation costs in both large-scale (>100 clients) and small-scale (< 10 clients) settings. Based on your feedback, we observed that SPFL’s early-stage performance gains were not sufficient to offset the increased communication cost. To address this, we adopted a hybrid strategy: pre-training the model with PFL, followed by SPFL fine-tuning. As shown in Tab 1 and 2, this significantly accelerates convergence, achieving strong performance in just a few rounds while reducing total communication and computation overhead.
>
> **Tab 1: Efficiency Comparison on CIFAR-10 (Dirichlet α=0.1,Number of clients=10)**
>
> |Method|Accuracy(%)|Rounds|Comm. Cost (GB)|Total Time(seconds)|Compute Cost(GB)|
> |-|-|-|-|-|-|
> |PFL(FedAvg)|79.15|500|116.90|3419.91|175.35|
> |SPFL(Ours)|84.17|15|21.042|678.6|87.675|
>
> **Tab 2: Efficiency Comparison on CIFAR-10 (Dirichlet α=0.1,Number of clients=100)**
>
> |Method|Accuracy(%)|Rounds|Comm. Cost(TB)|Total Time(seconds)|Compute Cost(TB)|
> |-|-|-|-|-|-|
> |PFL(FedAvg)|81.38|2000|18.30|30259.82|27.46|
> |SPFL(Ours)|86.68|10|4.68|14713.13|22.85|
>
> >**W3**:While the paper compares with several popular FL methods, it misses comparisons with recent works focused explicitly on catastrophic forgetting or domain adaptation in FL.
>
> We thank you for pointing out the lack of comparison with recent methods addressing catastrophic forgetting and domain adaptation in FL. Following your suggestion, we added comparisons with recent methods targeting catastrophic forgetting and domain adaptation in FL. We selected FedDA (ICLR-2024) and FedGALA (AAAI-2025) as representative methods that explicitly address domain adaptation and CAN (ICML-2025) for forgetting. All methods were implemented under the same sequential federated learning setting on CIFAR10 with 10 clients and Dir(α=0.1) distribution to ensure a fair comparison.
> As shown in the Tab 3, most existing methods focus on server-side alignment in the PFL setting, while the primary challenge of SPFL arises from **cross-client**, where catastrophic forgetting occurs across clients. Consequently, these methods are less effective in addressing the degradation of SPFL under domain shift. In contrast, our proposed GLAM module explicitly mitigates cross-client forgetting and demonstrates improved robustness in such scenarios.
>
> **Tab 3: Performance comparison of GLAM+SPFL with recent FL methods under domain shift.**
>
> |Method|P|A|C|S|Avg|
> |-|-|-|-|-|-|
> |CAN(ICML-2025)+SPFL|90.06|70.32|**77.63**|66.47|76.12|
> |FedDA(ICLR-2024)+SPFL|90.41|73.83|69.01|**74.01**|76.82|
> |FedGALA(AAAI-2025)+SPFL|87.58|68.58|68.17|66.56|72.72|
> |GLAM+SPFL(Ours)|**92.28**|**77.44**|74.40|67.32|**77.86**|
>
>
> >**W4**:The ablation on hyperparameters is appreciated, but more insight into how much each component (sequential update, parallel aggregation, GLAM) contributes independently to performance would be valuable
>
> Thank you for your valuable feedback. As shown in Tab 4 and 5 (to be added in the appendix), we further demonstrate the roles of the SPFL and GLAM modules under both category shift and domain shift. The GLAM module does not provide significant gains under category shift; instead, it serves as a compensatory mechanism to mitigate forgetting caused by the performance degradation of SPFL under domain shift. This will enable SPFL to serve as a viable alternative to PFL in cross-island settings (i.e., limited client base).
>
> **Tab 4: Performance under Category and Domain Shift with PFL, SFL, SPFL, and SPFL+GLAM**
>
> |Shift Type|PFL|SFL|SPFL|SPFL+GLAM|
> |-|-|-|-|-|
> |Category Shift|55.60|68.59|71.39|71.71|
> |Domain Shift(Avg)|78.37|60.04|76.32|77.87|
>
>
> ### Response to Question
> >**Q2:** How sensitive is SPFL’s performance to the choice of update order, especially under extreme domain shift? Does parallel aggregation fully mitigate the update-order sensitivity or only reduce it?
>
> Thank you for the insightful question. SPFL reduces update-order sensitivity by **aggregating multiple SFL orders**, avoiding the instability of any single update sequence. As shown in Tab 5, while SFL accuracy varies significantly across orders (up to 7.5% under α = 0.1), SPFL achieves stable and strong performance.
>
> However, under extreme domain shift, the main challenge is **catastrophic forgetting**. To address this, SPFL incorporates **GLAM**, which aligns local and global predictions to maintain consistency across clients. This design further enhances robustness in the face of severe heterogeneity. Here, SFL(i) denotes a sequential update process where client i initiates the update order.
>
> **Tab 5: Accuracy (%) of PFL, multiple SPL Different Order, and SPFL under different Dirichlet \alpha settings**
>
> |Method|PFL|SFL(1)|SFL(2)|SFL(3)|SFL(4)|SFL(5)|SPFL(ours)|
> |-|--|-|-|-|-|-|-|
> |Dir(0.1)|41.85|49.56|54.45|49.51|47.98|55.56|55.33|
> |Dir(0.5)|43.60|50.22|47.95|55.18|54.20|49.52|55.16|
>
> >**Q3:** Does the proposed GLAM introduce privacy risks by requiring clients to align outputs with previous client models? How is privacy preserved if models are indirectly influenced by earlier clients' data distributions?
>
> Thank you for raising this important point. GLAM utilizes the previous client's model solely for local inference, which differs fundamentally from data or gradient sharing and is considered a lower-risk form of model sharing. The process is entirely local—no data or intermediate outputs are transmitted—serving only as a reference for alignment. Similar strategies have been adopted in prior works such as FedDF and FedGen, which have demonstrated low privacy risk. In future work, we plan to further enhance security through techniques such as model encryption or differential privacy.
>
> >**Q4:** Can SPFL work in asynchronous or partial participation settings where not all clients are available in each round?
>
> Thank you for your insight comment. We conducted experiments with 100 clients under a partial participation setting, where only a fraction of clients ($E \in [0.1, 1.0]$) were randomly selected to participate in each round. As shown in the table below, SPFL consistently outperforms PFL across all participation rates, demonstrating strong robustness even with limited client availability. These results further validate the effectiveness of SPFL in partial participation scenarios, aligning well with our theoretical insights.
>
> **Tab 6: Accuracy comparison between PFL and SPFL under different client participation rates (\$E\$) with 100 total clients.**
>
> |Update\E|0.1|0.2|0.3|0.4|0.5|0.6|0.7|0.8|0.9|1|
> |-|-|-|-|-|-|-|-|-|-|-|
> |PFL|41.78|46.99|52.43|53.31|54.04|54.52|54.47|54.45|54.35|54.59|
> |SPFL|51.72|63.57|72.33|76.19|80.84|82.68|84.43|85.41|85.98|89.95|
>
> **We sincerely appreciate your insightful feedback and have addressed your concerns to the best of our ability. Should you have any further questions or recommendations, we would be glad to engage in further discussion.**

---

> > ### Comment · Reviewer_ARDv · 2025-08-04
> >
> > Thank you for your revisions and detailed responses.
> >
> > I appreciate the effort you’ve put into addressing my concerns. The improvements in clarity, particularly in the notation and simplification of the descriptions, are commendable. The explanation of the communication and computation costs of SPFL, along with the hybrid strategy for pre-training, effectively addresses concerns regarding scalability. The additional comparisons with recent methods enhance the paper and clearly highlight the advantages of the proposed approach. The ablation studies further clarify the contributions of SPFL and GLAM.
> >
> > Overall, I find the revisions well-executed and will adjust my recommendation to support acceptance.

---

> > > ### Author Response · Authors · 2025-08-04
> > >
> > > Thank you very much for your thoughtful and encouraging comments. We truly appreciate your recognition of our revisions and the improvements in clarity and scalability. We're glad that the additional experiments and comparisons helped clarify the effectiveness of our proposed SPFL framework and the GLAM module. We are grateful for your support and recommendation again.

---

> > ### Comment · Reviewer_BJo3 · 2025-08-05
> >
> > The rebuttal addresses some of my comments, yet many crucial experiments are missing from the main manuscript and appear only in the rebuttal. Because non-IID data is the core challenge for federated learning under category and domain shifts, omitting these results weakens the contribution. Therefore, i am changing my decision.

---

> > > ### Author Response · Authors · 2025-08-05
> > >
> > > Thank you for your comment and for taking the time to review our work.
> > >
> > > We would like to clarify that many of the experiments you raised are already included in the main manuscript, and the rebuttal primarily provides extended analyses and more detailed explanations. Specifically:
> > >
> > > Q1 (SPFL vs. SFL/PFL comparison): Table 4 presents direct comparisons between SPFL and PFL. In the rebuttal, we additionally provide results for multiple SFL orders.
> > >
> > > Q2 (non-IID performance): Table 2 reports SPFL's performance under non-IID settings on CIFAR-100. In the rebuttal, we provide additional results on Tiny-ImageNet to complement this analysis.
> > >
> > > Q3 (IID vs. non-IID): Figure 4(a)-(b) compares performance under IID (Dirichlet $\alpha$=100) and non-IID ($\alpha$=0.5) distributions. In the rebuttal, we include extended results for $\alpha$=0.1, 0.2, and 1.0.
> > >
> > > Q4 (Communication/Computation cost): Figure 5(c) shows the computational cost comparison, and Table 8 in the appendix presents a theoretical analysis of communication cost. We will move this content into the main paper and clarify the findings in the final version.
> > >
> > > Q5 (Update order sensitivity): Figure 2(c)(d) illustrates 3-client sequential order variations. The rebuttal extends this to 5 clients for completeness.
> > >
> > > We sincerely appreciate your concern and agree that improving the visibility of these experiments will strengthen the clarity and contribution of the paper. According to the NeurIPS policy,**the final version may include one additional page**, which gives us ample space to incorporate all key experimental results presented in the rebuttal into the main manuscript, and we will revise the structure accordingly to enhance readability and rigor.
> > >
> > > Thank you again for your constructive feedback and thoughtful review.We hope this clarification addresses your concerns and that you may consider maintaining your original rating.

---

> > > ### Comment · Reviewer_ARDv · 2025-08-06
> > >
> > > Thank you for your comments.
> > >
> > > Upon reviewing the rebuttals to the comments, I found that most rebuttals extend the results of what’s already in the main paper or appendix. The authors have addressed these concerns well with additional experiments and explanations, which are natural continuations rather than missing essentials, and the authors have committed to including them in the final version.

---

> > > > ### Author Response · Authors · 2025-08-06
> > > >
> > > > Thank you for your understanding and trust. We will incorporate these extended experiments into the final version and plan to update the revised manuscript on arXiv coming soon. We sincerely welcome any further comments or suggestions.

---

### Official Review · Reviewer_BJo3 · 2025-07-04

**Clarity:** 2
**Significance:** 3
**Originality:** 3
**Rating:** 4
**Confidence:** 4

**Summary:**

This paper introduces SPFL (Sequential updates with Parallel aggregation Federated Learning), a novel framework that combines the strengths of Parallel Federated Learning (PFL) and Sequential Federated Learning (SFL). While PFL (e.g., FedAvg) enables parallel client training but suffers from limited data exposure, SFL allows a single model to traverse clients sequentially, enhancing data utilization. However, SFL is vulnerable to update order sensitivity and catastrophic forgetting, especially under Non-IID conditions. SPFL addresses these issues by organizing sequential client updates and periodically performing parallel aggregation, mitigating order bias and improving convergence. The authors also propose GLAM (Global-Local Alignment Module) to reduce forgetting by aligning local predictions with both the global and previous client models. The paper provides rigorous convergence analysis under various convexity settings and demonstrates, through extensive experiments on category and domain shifts, that SPFL significantly improves performance and stability when integrated into existing FL methods like FedAvg, FedProx, and FedDC. Overall, SPFL effectively bridges PFL and SFL, offering a robust solution for federated learning under heterogeneous data conditions.

**Questions:**

In tables like Table 2, only SPFL enhanced versions of FL methods are shown. Including original (non-SPFL) baselines such as FedAvg, FedProx, and FedDC would better quantify SPFL’s contribution.

Current results are based on CIFAR-10 and similar datasets. To demonstrate scalability and robustness, evaluations on more complex datasets (e.g., CIFAR-100, subsets of ImageNet) are recommended.

While the focus is on Non-IID scenarios, the paper should detail the partitioning strategies used. Including comparisons across varying levels of data heterogeneity (IID vs extreme Non-IID) would strengthen the claims.


SPFL introduces additional communication through sequential and parallel updates in line 246. A comparison of communication/computation overhead versus standard methods like FedAvg would help clarify trade offs.

Since update order sensitivity is a central issue, empirical evidence (e.g., accuracy variance across different client sequences) should be presented to support SPFL’s robustness against order changes.

**Ethical Concerns:**

["Major Concern: Improper research involving human subjects"]

**Limitations:**

As addressed above

**Quality:**

3

**Strengths And Weaknesses:**

The paper addresses a key limitation in FL under Non-IID data by combining sequential updates with parallel aggregation. This hybrid strategy (SPFL) effectively balances broader data exposure from Sequential Federated Learning (SFL) and order bias control via periodic parallel aggregation a notable conceptual contribution.
The authors provide convergence analysis for SPFL under strongly convex, convex, and non-convex settings. These theoretical results show SPFL’s stability and improved convergence bounds over FedAvg and pure SFL, adding credibility to the proposed method.
The GLAM module aligns client model predictions with both the previous local model and the global model, helping preserve earlier knowledge and reduce forgetting. This is a practical and validated addition.

Experiments under category and domain shift conditions show consistent improvements when SPFL is applied to baseline methods like FedAvg and FedProx. The results highlight SPFL’s generalizability and effectiveness in challenging IID settings.

---

> ### Author Rebuttal · Authors · 2025-07-30
>
> Dear Reviewer BJo3：
>
> We sincerely thank you for the detailed and constructive feedback. We are encouraged that the reviewer appreciates our contributions in proposing SPFL as a hybrid solution bridging PFL and SFL, our theoretical convergence analysis, and the design of the GLAM module. Below, we address the reviewer’s insightful questions and concerns.
>
> **Note: To quickly converge,the most model below was initialized using a FedAvg pre-trained checkpoint trained for 200 rounds on CIFAR-10,the number of clients is 10.**
> ### Response to Question
> >**Q1**:In tables like Table 2, only SPFL enhanced versions of FL methods are shown. Including original (non-SPFL) baselines such as FedAvg, FedProx, and FedDC would better quantify SPFL’s contribution.
>
> Thank you for your valuable comments.  We compare three update paradigms: PFL, SFL, and our proposed SPFL. Since SFL is decentralized, to facilitate the evaluation，we use the final client’s model as the global model. As shown in the table,  SPFL reintroduces server aggregation and demonstrates strong adaptability across FL algorithms like FedProx, consistently improving performance.
>
> **Tab 1:Comparison of different update strategies (PFL, SFL, and our SPFL) under various FL methods on CIFAR-10 (Dirichlet α = 0.1).**
>
> |Update\Method|FedAvg|FedProx|FedDC|Moon|
> |-|-|-|-|-|
> |PFL|55.60|56.65|55.16|56.33|
> |SFL|68.59|53.75|54.41|55.24|
> |SPFL(ours)|**71.39**|**57.66**|**55.26**|**58.44**|
> >**Q2**:Current results are based on CIFAR-10 and similar datasets. To demonstrate scalability and robustness, evaluations on more complex datasets (e.g., CIFAR-100, subsets of ImageNet) are recommended.
>
> We appreciate your suggestion regarding the evaluation of more complex datasets. We have addressed this by adding results on more complex datasets, including CIFAR-100 and Tiny-ImageNet, as shown in the table below. These results further validate the scalability and robustness of SPFL across diverse FL methods.
>
> **Tab 2:Comparison of SPFL and PFL on Tiny-ImageNet and CIFAR-100 across various FL methods.(Dirichlet α = 0.1)**
>
> |Method|Tiny-ImageNet(PFL)|Tiny-ImageNet(SPFL)|CIFAR-100(PFL)| CIFAR-100(SPFL)|
> |-|-|-|-|-|
> |FedAvg|15.69|**33.82 (+18.13)**| 57.55| **64.33 (+6.78)**|
> |FedProx|15.47|16.65 (+1.18)|57.56| 60.46 (+2.90)|
> |MOON|15.42| 16.47 (+1.05)| 58.44|63.54 (+5.20)|
> |FedDyn|15.22|17.66 (+2.44)|64.11|64.24 (+0.13)|
> |Scaffold|14.58|18.16 (+3.58)|56.15|65.05 (+8.90)|
> |FedDC|15.37|16.45 (+1.08)|**64.44**|**69.88 (+5.44)**|
> |FedNova|15.53|16.45 (+0.98)|57.64|64.47 (+6.83)|
> |FedDisco|**33.55**|**34.50 (+0.95)**|57.50|64.44 (+6.94)|
> >**Q3**:While the focus is on Non-IID scenarios, the paper should detail the partitioning strategies used. Including comparisons across varying levels of data heterogeneity (IID vs extreme Non-IID) would strengthen the claims.
>
> We thank the reviewer for pointing out the importance of evaluating model performance under varying degrees of data heterogeneity. Our main experiments were conducted under the widely adopted Non-IID setting with Dirichlet \$\alpha=0.1\$, which we believe best reflects realistic federated learning scenarios. We also reported results under the IID setting in the main line 211. As suggested, we have included a comprehensive comparison across a range of Dirichlet partitioning parameters ($\alpha$), which control the degree of Non-IID-ness—from highly skewed ($\alpha=0.1$) to nearly IID ($\alpha=100$). The tab 3 below reports the accuracy of SPFL and a representative PFL baseline (FedAvg):
>
> **Tab 3: Accuracy (%) of SPFL and PFL (FedAvg) under Different Dirichlet $\alpha$ Settings**
>
> |Update\Dir|0.1|0.2|0.5|1|100|
> |-|-|-|-|-|-|
> |PFL(FedAvg)|39.25|43.40|42.85|46.03|53.37|
> |SPFL(Our)|47.55|50.96|49.38|49.47|55.27|
>
> We will integrate this analysis into the revised manuscript and provide a clearer description of our partitioning strategy in the appendix for completeness and reproducibility.
>
> >**Q4**:SPFL introduces additional communication through sequential and parallel updates in line 246. A comparison of communication/computation overhead versus standard methods like FedAvg would help clarify trade-offs.
>
> Thank you for pointing this out. We agree that SPFL introduces additional coordination for sequential and parallel updates. We acknowledge that SPFL increases the per-round communication volume by a factor of M/2 and the computational cost by approximately 5M/3 during training. **However, since SPFL converges faster, SPFL effectively reduces the number of training rounds, thereby reducing the overall communication and computation costs, as shown in Tab 4 and 5.**
>
>
> **Tab 4: Efficiency Comparison on CIFAR-10 (Dirichlet α=0.1,Number of clients=10)**
>
> |Method|Accuracy(%)|Rounds|Comm. Cost (GB)|Total Time(seconds)|Compute Cost(GB)|
> |-|-|-|-|-|-|
> |PFL(FedAvg)|79.15|500|116.90|3419.91|175.35|
> |SPFL(Ours)|84.17|15|21.042|678.6|87.675|
>
> **Tab 5: Efficiency Comparison on CIFAR-10 (Dirichlet α=0.1,Number of clients=100)**
>
> |Method|Accuracy(%)|Rounds|Comm. Cost(TB)|Total Time(seconds)|Compute Cost(TB)|
> |-|-|-|-|-|-|
> |PFL(FedAvg)|81.38|2000|18.30|30259.82|27.46|
> |SPFL(Ours)|86.68|10|4.68|14713.13|22.85|
>
>
> >**Q5**:Since update order sensitivity is a central issue, empirical evidence (e.g., accuracy variance across different client sequences) should be presented to support SPFL’s robustness against order changes.
>
> Thank you for highlighting this important point. While we briefly illustrated SFL’s instability in Fig. 2 of the main, we now provide a more comprehensive analysis. As shown in Table 6, after sequential training across five clients, SFL exhibits substantial performance variance due to different update orders—for instance, accuracy ranges from 47.98% to 55.56% when α = 0.1. In contrast, SPFL achieves consistently great performance, closely matching the best SFL case while eliminating sensitivity to update order. This confirms that SPFL robustly mitigates order sensitivity, and we will incorporate this expanded experiment in the final version. Here, SFL(i) denotes a sequential update process where client i initiates the update order.
>
> **Tab 6: Accuracy (%) of PFL, multiple SFL variants, and SPFL under different Dirichlet α settings**
>
> |Method|PFL|SFL(1)|SFL(2)|SFL(3)|SFL(4)|SFL(5)|SPFL(ours)|
> |-|--|-|-|-|-|-|-|
> |Dir(0.1)|41.85|49.56|54.45|49.51|47.98|55.56|55.33|
> |Dir(0.5)|43.60|50.22|47.95|55.18|54.20|49.52|55.16|
>
>
> **We sincerely appreciate your insightful feedback and have addressed your concerns to the best of our ability. Should you have any further questions or recommendations, we would be glad to engage in further discussion.**

---

### Note · Authors · 2025-08-16

Dear NeurIPS 2025 Reviewers, AC, SAC, and PC,

We sincerely thank the AC and all reviewers for their time and valuable feedback. The discussion period has been highly productive and has significantly strengthened our manuscript.

Our paper introduces SPFL, a novel federated learning framework that combines sequential and parallel updates. We provide theoretical convergence guarantees and show that SPFL effectively overcomes the update-order sensitivity issue of SFL under category and domain shifts. In particular, SPFL demonstrates stronger generalization under category shifts, while the proposed GLAM module alleviates challenges under domain shifts. During rebuttal, we further proposed a hybrid SPFL–PFL approach, which directly addresses concerns about scalability to many clients and high communication costs.

Overall, the reviews recognize that our work is sufficiently novel, and we are encouraged by the positive reception. **Reviewer cWLJ** clearly supports acceptance. **Reviewer ARDv**, after reading our rebuttal, explicitly stated a willingness to raise their score, acknowledging the additional clarifications we provided. **Reviewers BJo3 and wzqr** maintained their scores after our responses, and both agreed that the paper is novel, though **Reviewer wzqr** expressed some concerns regarding the experimental setup. We would like to emphasize that our rebuttal did not involve changing the method itself but rather clarifying its scope and providing a practical application plan that better reflects real-world scenarios. We believe this strengthened the paper without altering its original contribution.

We are confident that our paper, now enhanced through this rigorous review process, presents an even stronger and more robust contribution to the field. We hope we have resolved all outstanding issues and thank you for your consideration.

Best regards,

The Authors

---

### Decision · Program_Chairs · 2025-09-17

**Decision:**

Accept (poster)

**Comment:**

This paper proposes SPFL, a novel federated learning framework combining parallel and serial updates, with a solid convergence analysis. It effectively mitigates update-order sensitivity under category and domain shifts, with GLAM addressing the latter more effectively. While one reviewer maintained a low score over presentation choices, most concerns were addressed through additional experiments and clarifications. Therefore, I recommend acceptance.